# TENSOR LEARNING WITH ORTHOGONAL, LORENTZ, AND SYMPLECTIC SYMMETRIES

**Wilson G. Gregory**
Dept. of Applied Mathematics and Statistics
Mathematical Institute for Data Science
Johns Hopkins University
Baltimore, MD 21218
`wgregor4@jhu.edu`

**Josué Tonelli-Cueto**
Department of Quantitative Methods
CUNEF Universidad
Madrid, SPAIN
`josue.tonelli.cueto@bizkaia.eu`

**Nicholas F. Marshall**
Department of Mathematics
Oregon State University
Corvallis, OR 97331
`marsnich@oregonstate.edu`

**Andrew S. Lee**
Department of Mathematics and Computer Science
Adelphi University
Garden City, NY 11530
`alee2@adelphi.edu`

**Soledad Villar**
Dept. of Applied Mathematics and Statistics
Mathematical Institute for Data Science
Johns Hopkins University
Baltimore, MD 21218
Flatiron Institute
New York, NY 10010
`svillar3@jhu.edu`

## ABSTRACT

Tensors are a fundamental data structure for many scientific contexts, such as time series analysis, materials science, and physics, among many others. Improving our ability to produce and handle tensors is essential to efficiently address problems in these domains. In this paper, we show how to exploit the underlying symmetries of functions that map tensors to tensors. More concretely, we develop universally expressive equivariant machine learning architectures on tensors that exploit that, in many cases, these tensor functions are equivariant with respect to the diagonal action of the orthogonal, Lorentz, and/or symplectic groups. We showcase our results on three problems coming from material science, theoretical computer science, and time series analysis. For time series, we combine our method with the increasingly popular path signatures approach, which is also invariant with respect to reparameterizations. Our numerical experiments show that our equivariant models perform better than corresponding non-equivariant baselines.

## 1 INTRODUCTION

Tensors are fundamental mathematical objects that appear in a broad spectrum of domains (Ballard & Kolda, 2025; Landsberg, 2012). In the natural sciences, tensor-valued data are used to express polarizations (Melrose & Stoneham, 1977), permeabilities (Durlofsky, 1991), and stresses (Levitas et al., 2019), for example. Theoretical computer science studies several problems related to tensors, including factorization or decomposition of tensors (Rabanser et al., 2017), and planted tensor models (Hopkins et al., 2016). For time series analysis, the path signature introduced in Chen (1957; 1958) transforms path data into a sequence of tensors, enabling methods to process time series data that are invariant with respect to reparameterizations (Bonnier et al., 2019; Pfeffer et al., 2019; Tóth, 2025).

In physics, tensors are not just a multidimensional array of numbers. Tensors also have specific transformation properties under changes of coordinates, so any function of a tensor should obey certain transformation rules. These rules can be expressed as invariances or equivariances with respect to group actions. In this work, we show how to parameterize machine learning models that learn tensors and implement these symmetries, and we demonstrate that imposing them improves the learning performance on a variety of problems. The relevant symmetries are given by classical Lie groups acting diagonally on tensors. The groups we study arise naturally in physics and other settings, including the orthogonal group $O(d)$ (which typically appears in coordinate transformations), the indefinite orthogonal group $O(s, k - s)$ (which includes the Lorentz group, a fundamental group for special relativity), and the symplectic group $Sp(d)$ (the underlying group in much of classical and quantum mechanics).

We apply the methods to problems in time series prediction, materials science, and theoretical computer science. Our equivariant learned models outperform prior static methods and non-equivariant learned models in almost all cases.

**Our Contributions.** We provide a generic recipe to define equivariant machine learning models mapping from tensors to tensors. To this end, we give explicit parameterizations for polynomials (Sec. 3) and analytic functions with globally convergent Taylor series (Sec. 4) from tuples of tensor inputs to tensor outputs that are equivariant with respect to the orthogonal (Sec. 3), indefinite orthogonal (which includes Lorentz), and symplectic groups (Sec. 4). This generalizes the existing results of Villar et al. (2021) and leverages the tensor invariant theory (Appleby et al., 1987; Jeffreys, 1973; Roe Goodman, 2009) into a format useful for machine learning frameworks. On first reading and for those primarily interested in practical applications, we suggest focusing on Corollary 1 in Section 3 from which our experiments follow.

In Section 5 we consider three disparate applications. For materials science, we use our machine learning model on tensors to learn the relationship between second-order stress and strain tensors of an $O(d)$-isotropic neo-Hookean hyperelastic material (Garanger et al., 2024). For time series, we focus on their representation via path signatures, which are sequences of tensors of growing order (Chen, 1958; Lyons et al., 2007). These tensors can be used for any downstream learning tasks. Here, we consider the problem of estimating the path signature from only a few sampled points on the path. Finally, from theoretical computer science we consider the problem of sparse vector estimation. Given a vector space $\mathbb{R}^n$ and a random orthonormal basis of a subspace $\mathbb{R}^d$ that contains a sparse vector $v_0$ for $d \ll n$, can we recover $v_0$? The problem has roots in dictionary learning (Spielman et al., 2012) and tensor PCA, it is closely related to the spiked tensor model (Montanari & Richard, 2014), and has several applications (Qu et al., 2020).

**Related work.** Many recent theoretical and applied efforts have focused on the implementation of symmetries and other structural constraints in the design of machine learning models. This is the case of graph neural networks Scarselli et al. (2008); Maron et al. (2019), geometric deep learning Bronstein et al. (2021); Weiler et al. (2021), and AI for science Zhang et al. (2023). The goal is to design a hypothesis class of functions with good *inductive bias* that is aligned with the theoretical framework of the physical, mathematical, or algorithmic objects it aims to represent. This includes respecting coordinate freedoms (Villar et al., 2023a), conservation laws (Alet et al., 2021), or internal symmetries (e.g. in the implicit neural representations framework (Lim et al., 2023)). Symmetries have also been used to provide interpretability to learned data representations (Suau et al., 2023; Gupta et al., 2023). Mathematically, it has been shown that imposing symmetries can improve the generalization error and sample complexity of machine learning models (Elesedy, 2021b; Wang et al., 2021b; Elesedy, 2021a; Bietti et al., 2021; Petrache & Trivedi, 2024; Tahmasebi & Jegelka, 2023; Huang et al., 2024).

A variety of methods can be used for implementing invariances or equivariances, including group convolutions (Cohen & Welling, 2016; 2017; Wang et al., 2021a), irreducible representations (Fuchs et al., 2020; Kondor, 2018; Weiler et al., 2018; Cohen et al., 2018; Weiler & Cesa, 2019; Simeon & De Fabritiis, 2023), constraints on optimization (Finzi et al., 2021), canonicalization (Kaba et al., 2023), and invariant theory (Gripaios et al., 2021; Haddadin, 2022; Villar et al., 2021; Blum-Smith & Villar, 2023; Villar et al., 2023b; Blum-Smith et al., 2025). This work is closer to the line of research that constructs explicit equivariant functions from invariant features.

Closest to us is the work of Kunisky, Moore, and Wein on tensor cumulants (Kunisky et al., 2024); and the works parameterizing equivariant tensor functions using Clebsch–Gordan methods and spherical harmonics, including e3nn (Geiger & Smidt, 2022), escnn (Cesa et al., 2022), and the recent work of Domina et al. (2025). In Kunisky et al. (2024), it is shown that $O(d)$-invariant polynomials on symmetric tensors can be turned into $O(d)$-invariant polynomials over general tensors by symmetrizing over $O(d)$. They do not consider learning applications, tensors of different orders or parities, nor indefinite orthogonal groups or the symplectic group.

The works in Geiger & Smidt (2022); Cesa et al. (2022); Domina et al. (2025) implement equivariant machine learning models on tensors using representation theory. To do so they decompose tensors into irreducible representations and use Schur's lemma to parameterize the equivariant linear maps. If the orders of the intermediate tensor layers are large enough, then these models can parameterize polynomial equivariant functions of arbitrary degree (and are universal in a Stone-Weierstrass sense). The method we introduce here also parameterizes equivariant tensor polynomials of arbitrary degree, but does so using invariant theory results instead of irreducible representations. Our parameterization for the invariant and equivariant functions does not require the computation of the Clebsch–Gordan coefficients. The Clebsch–Gordan–based methods in Geiger & Smidt (2022); Cesa et al. (2022); Domina et al. (2025) are specific for SO($d$) and O($d$) for $d = 2, 3$, whereas our method applies to other groups as well. We remark that those methods are more memory efficient than our general formulation in Theorems 1 and 2, but they are comparable to our Corollaries 1 and 3 (which require the inputs to be vectors but are applicable to O(d), the Lorentz group, the symplectic group). In summary, the computational and approximation power should be equivalent, however, the parameterization is different and the mathematical techniques used to arrive at the parameterization are also different.

Another related work Pearce-Crump (2023) characterizes neural networks that are $O(d)$, $SO(d)$, and $Sp(d)$ equivariant, but only for the case of functions whose input is a tensor power of $\mathbb{R}^n$ and whose output is a tensor power of $\mathbb{R}^n$. Finally, some other works use outer products and contractions of Cartesian tensors to capture higher order interactions, such as HotPP (Wang et al., 2024), GI-Net (Gregory et al., 2025), and Vector Neurons (Deng et al., 2021). These models build higher order tensors in a point cloud or image setting, while our method exploits shortcuts depending on the type of input to build efficient models.

## 2 DEFINITIONS

To simplify the exposition, we start by focusing on the case of the orthogonal group before extending the result to the indefinite orthogonal and symplectic groups. We consider the orthogonal group $O(d)$, the group of isometries of Euclidean space $\mathbb{R}^d$ that fix the origin. It acts on vectors and pseudovectors $v \in \mathbb{R}^d$ in the following way:

$$g \cdot v = \det(M(g))^{\frac{1-p}{2}} M(g) \, v, \tag{1}$$

where $g \in O(d)$, $M(g) \in \mathbb{R}^{d \times d}$ is the standard matrix representation of $g$ (i.e. $M(g)^\top M(g) = \mathbb{I}_d$, where $\mathbb{I}_d$ is the identity matrix), and $p \in \{-1, +1\}$ is the parity of $v$. If $p = +1$ we obtain the standard $O(d)$ action on $\mathbb{R}^d$ *vectors*. If $p = -1$ we obtain the $O(d)$ action on what in physics are known as *pseudovectors*. For a common pseudovector, consider a rotating Ferris wheel with angular velocity whose direction is given by the right-hand rule. A reflection of the wheel, which will have $\det(M(g)) = -1$ in (1), does not change the direction of rotation or angular velocity.

**Definition 1** ($k_{(p)}$-tensors)**.** We define the space of $1_{(p)}$-*tensors* to be $\mathbb{R}^d$ equipped with the action $O(d)$ defined by (1). If $v_i$ is a $1_{(p_i)}$-tensor for $i = 1, \ldots, k$, then $a := v_1 \otimes \ldots \otimes v_k \in (\mathbb{R}^d)^{\otimes k}$ is a *rank-1 $k_{(p)}$-tensor*, where $p = \prod_{i=1}^k p_i$ and the action of $O(d)$ is the diagonal action:

$$g \cdot (v_1 \otimes \ldots \otimes v_k) = (g \cdot v_1) \otimes \ldots \otimes (g \cdot v_k). \tag{2}$$

This definition generalizes to higher rank $k_{(p)}$-tensors by linearity (see (5) below). The space of $k_{(p)}$-tensors in $d$ dimensions is denoted $\mathcal{T}_k(\mathbb{R}^d, p)$. We will write $+$ or $-$ for p when it is clear; for example, $\mathcal{T}_1(\mathbb{R}^d, -)$ is the space of pseudovectors and $\mathcal{T}_1(\mathbb{R}^d, +)$ is the space of vectors.

**Definition 2** (Einstein summation notation)**.** Suppose that $a$ is a $k_{(p)}$-tensor. Let $[a]_{i_1, \ldots, i_k}$ denote the $(i_1, \ldots, i_k)$-th entry of $a$, where $i_1, \ldots, i_k$ range from 1 to $d$. The *Einstein summation notation* is used to represent tensor products[1] where repeated indices are summed over. In each product, a given

---

[1] We will identify vectors with co-vectors in the usual way and will not distinguish lower vs upper scripts.

index can appear either exactly once, in which case it appears in the result, or exactly twice, in which case it is summed over and does not appear in the result.

For example, in Einstein summation notation, the product of two $2_{(+)}$-tensors (i.e., the matrix product $ab$ of two $d \times d$ matrices $a$ and $b$) is written as

$$[a\,b]_{i,j} = [a]_{i,\ell}\,[b]_{\ell,j} := \sum_{\ell=1}^{d} [a]_{i,\ell}\,[b]_{\ell,j}. \tag{3}$$

Using Einstein summation notation, the action of $g \in O(d)$ on rank-1 tensors can be extended to general tensors by linearity by expressing $b \in \mathcal{T}_k(\mathbb{R}^d, p)$ as a linear combination of (rank-1) standard basis tensors $e_{i_1,\ldots,i_k} = e_{i_1} \otimes \cdots \otimes e_{i_k}$, where $[e_i]_i = 1$ and $[e_i]_j = 0$ for $i \neq j$

$$[g \cdot b]_{i_1,\ldots,i_k} = [b]_{j_1,\ldots,j_k}[g \cdot (e_{j_1} \otimes \cdots \otimes e_{j_k})]_{i_1,\ldots,i_k} = [b]_{j_1,\ldots,j_k}[g \cdot e_{j_1}]_{i_1} \cdots [g \cdot e_{j_k}]_{i_k}. \tag{4}$$

Note that the action (1) on a $k_{(p)}$-tensor $b$ can be written as

$$[g \cdot b]_{i_1,\ldots,i_k} = \det(M(g))^{\frac{1-p}{2}}\,[b]_{j_1,\ldots,j_k}\,[M(g)]_{i_1,j_1} \cdots [M(g)]_{i_k,j_k} \tag{5}$$

for all $g \in O(d)$. For example, in this notation a $2_{(+)}$-tensor has the transformation property $[g \cdot b]_{i,j} = [b]_{k,\ell}\,[M(g)]_{i,k}\,[M(g)]_{j,\ell}$, which, in normal matrix notation, is written as $g \cdot b = M(g)\,b\,M(g)^\top$.

When multiple tensors are combined, and all their indices appear in the result, we refer to that as the tensor product or outer product. When indices are summed over, we refer to that as the contraction or scalar product. We will further focus on a specific case of multiple tensor contractions that we will refer to as a $k$-contraction.

**Definition 3** (Outer product of tensors). Given $a \in \mathcal{T}_k(\mathbb{R}^d, p)$ and $b \in \mathcal{T}_{k'}(\mathbb{R}^d, p')$, the *outer product*, denoted $a \otimes b$, is a tensor in $\mathcal{T}_{k+k'}(\mathbb{R}^d, p\,p')$ defined as $[a \otimes b]_{i_1,\ldots,i_{k+k'}} = [a]_{i_1,\ldots,i_k}\,[b]_{i_{k+1},\ldots,i_{k+k'}}$. We write $a^{\otimes k}$ to denote the outer product of $a$ with itself $k$ times and use the convention for $k = 0$ that $a^{\otimes 0} \otimes b = b$.

**Definition 4** ($k$-contraction). Given a tensor $a \in \mathcal{T}_{2k+k'}(\mathbb{R}^d, p)$, the $k$-*contraction* of $a$, denoted $\iota_k(a)$, is the $k'_{(p)}$-tensor that contracts the $2k$ first indices as follows (in Einstein summation):

$$[\iota_k(a)]_{j_1,\ldots,j_{k'}} := [a]_{i_1,\ldots,i_k,i_1,\ldots,i_k,j_1,\ldots,j_{k'}}. \tag{6}$$

For instance, if $a = u \otimes v \otimes x \otimes y \otimes z \in \mathcal{T}_{4+1}(\mathbb{R}^d, p)$ then $\iota_1(a) = \langle u, x \rangle \langle v, y \rangle z$, where $\langle u, x \rangle$ denotes the standard inner product between $u$ and $x$. Since $k_{(p)}$-tensors are elements of the vector space $(\mathbb{R}^d)^{\otimes k}$, tensor addition and scalar multiplication are defined in the usual way. The final operation on tensors is the permutation of the indices.

**Definition 5** (Permutations of tensor indices). Given $a \in \mathcal{T}_k(\mathbb{R}^d, p)$ and permutation $\sigma \in S_k$, the *permutation of tensor indices* of $a$ by $\sigma$, denoted $a^\sigma$, is defined by

$$[a^\sigma]_{i_1,\ldots,i_k} := [a]_{i_{\sigma^{-1}(1)},\ldots,i_{\sigma^{-1}(k)}}. \tag{7}$$

**Definition 6** (Invariant and equivariant functions). We say that $f : \mathcal{T}_k(\mathbb{R}^d, p) \to \mathcal{T}_{k'}(\mathbb{R}^d, p')$ is $O(d)$-invariant if

$$f(g \cdot a) = f(a), \quad \text{for all} \quad g \in O(d). \tag{8}$$

We say that $f : \mathcal{T}_k(\mathbb{R}^d, p) \to \mathcal{T}_{k'}(\mathbb{R}^d, p')$ is $O(d)$-equivariant if

$$f(g \cdot a) = g \cdot f(a), \quad \text{for all} \quad g \in O(d). \tag{9}$$

If $f$ were instead a function with multiple inputs, then the same group element $g$ would act on all inputs simultaneously.

**Definition 7** (Isotropic tensors). We say that a tensor $a \in \mathcal{T}_k(\mathbb{R}^d, p)$ is $O(d)$-isotropic if $g \cdot a = a$, for all $g \in O(d)$.

There are two special tensors, the Kronecker delta, and the Levi-Civita symbol. These tensors are $O(d)$-isotropic and, as we will show in Appendix C, we can construct all $O(d)$-isotropic tensors using only Kronecker deltas and Levi-Civita symbols.

**Definition 8** (Kronecker delta). The *Kronecker delta*, $\delta$, is the $O(d)$-isotropic $2_{(+)}$-tensor satisfying $[\delta]_{ij} = 1$ if $i = j$ and $0$ otherwise. When considered as a matrix, it is the identity matrix $\mathbb{I}_d$.

**Definition 9** (Levi-Civita symbol). The *Levi-Civita symbol*, $\epsilon$, in dimension $d \geq 2$ is the $O(d)$-isotropic $d_{(-)}$-tensor such that $[\epsilon]_{i_1,\ldots,i_d} = 0$ if any two of the $i_1,\ldots,i_d$ are equal, $[\epsilon]_{i_1,\ldots,i_d} = +1$ if $i_1,\ldots,i_d$ is an even permutation of $1,\ldots,d$, and $[\epsilon]_{i_1,\ldots,i_d} = -1$ if $i_1,\ldots,i_d$ is an odd permutation of $1,\ldots,d$. For example, when $d = 2$ this is simply the matrix $\begin{bmatrix} 0 & 1 \\ -1 & 0 \end{bmatrix}$.

## 3 $O(d)$-EQUIVARIANT POLYNOMIAL FUNCTIONS

In this section, we characterize the $O(d)$-equivariant polynomial functions mapping multiple tensor inputs to tensor outputs. On first reading and for those primarily interested in practical applications, we advise focusing on Corollary 1 and Example 1 below.

In what follows we consider functions of $n$ input tensors of orders $k_i$ and parities $p_i$ for $i = 1,\ldots,n$, and fixed dimension $d$. This space is expressed by the cartesian product $\prod_{i=1}^{n} \mathcal{T}_{k_i}(\mathbb{R}^d, p_i) = (\mathcal{T}_{k_1}(\mathbb{R}^d, p_1),\ldots,\mathcal{T}_{k_n}(\mathbb{R}^d, p_n))$. For many practical applications all inputs are of the same type, but we write the theory in this generality to allow for functions that take, for example, $1_{(+)}$-tensor positions and $2_{(+)}$-tensor stresses as inputs.

The theorem below states that every $O(d)$-equivariant polynomial from tensors to tensors can be written as a combination of tensor products of the inputs with isotropic tensors followed by Einstein-summation contractions. Each term in the right-hand-side of (10) should be viewed as combining $r$ of the input tensors with the tensor product, then mapping them to the appropriate output with a linear map. Since a linear map between tensors can always be written as a tensor product followed by a sequence of contractions (Dimitrienko, 2013, Theorem 5.1), the linear map is implemented by the tensor $c_{\ell_1,\ldots,\ell_r}$. However, since the function is also $O(d)$-equivariant, the tensor $c_{\ell_1,\ldots,\ell_r}$ must be $O(d)$-isotropic. The theorem states all the tensor equivariant polynomials can be expressed this way.

**Theorem 1.** Let $f : \prod_{i=1}^{n} \mathcal{T}_{k_i}(\mathbb{R}^d, p_i) \to \mathcal{T}_{k'}(\mathbb{R}^d, p')$ be an $O(d)$-equivariant polynomial function of degree at most $R$. Then we may write $f$ as follows:

$$f(a_1,\ldots,a_n) = \sum_{r=0}^{R} \sum_{1 \leq \ell_1 \leq \cdots \leq \ell_r \leq n} \iota_{k_{\ell_1,\ldots,\ell_r}}(a_{\ell_1} \otimes \ldots \otimes a_{\ell_r} \otimes c_{\ell_1,\ldots,\ell_r}) \tag{10}$$

where $c_{\ell_1,\ldots,\ell_r}$ is an $O(d)$-isotropic $(k_{\ell_1,\ldots,\ell_r} + k')_{(p_{\ell_1,\ldots,\ell_r} p')}$-tensor with order and parity chosen to be consistent with the output's ($k_{\ell_1,\ldots,\ell_r} = \sum_{q=1}^{r} k_{\ell_q}$ and $p_{\ell_1,\ldots,\ell_r} = \prod_{q=1}^{r} p_{\ell_q}$).

The proof of Theorem 1 is given in Appendix B. The result is a clean theoretical characterization of $O(d)$-equivariant polynomial tensor functions with arbitrary order tensor inputs. However, computing large polynomials with all possible $O(d)$-isotropic tensors is impractical. One option is considering low-degree polynomials as in Example 2 in the Appendix. Alternatively, in many applications we only need a function that has $1_{(+)}$-tensors (i.e. vectors) as input and a $k_{(+)}$-tensor as output, and the problem takes on a form more amenable to computation.

The condition that $c_{\ell_1,\ldots,\ell_r}$ in Theorem 1 is $O(d)$-isotropic is quite restrictive. Lemma 3 in Appendix C, taken from Jeffreys (1973), shows that all isotropic tensors can be constructed from the Kronecker delta $\delta$ (Definition 8) and the Levi-Civita symbol (Definition 9).

The following corollary says that when the inputs of the $O(d)$-equivariant function are vectors, and the output is a tensor, we can write the function as a linear combination of tensor products of the input vectors and Kronecker deltas (and permutations of them), and the coefficients are scalar functions that only depend on the pairwise inner products of the input vectors. The proof is in Appendix C.

**Corollary 1.** Let $f : \prod_{i=1}^{n} \mathcal{T}_1(\mathbb{R}^d, +) \to \mathcal{T}_{k'}(\mathbb{R}^d, +)$ be an $O(d)$-equivariant polynomial function. Then, we may write it as

$$f(v_1,\ldots,v_n) = \sum_{t=0}^{\lfloor \frac{k'}{2} \rfloor} \sum_{\sigma \in S_{k'}} \sum_{1 \leq J_1 \leq \ldots \leq J_{k'-2t} \leq n} q_{t,\sigma,J}\left( (\langle v_i, v_j \rangle)_{i,j=1}^{n} \right) \left( v_{J_1} \otimes \ldots \otimes v_{J_{k'-2t}} \otimes \delta^{\otimes t} \right)^{\sigma},$$

$$\tag{11}$$

where $J = (J_1, \ldots, J_{k'-2t})$ are indices of the input tensors, and the function $q_{t,\sigma,J}$ which depends on the tuple $(t, \sigma, J)$ is a polynomial of the inner products of the input vectors.

The second factor is a permutation of the outer product of $t$ Kronecker deltas and $k' - 2t$ of the input vectors $v_1, \ldots, v_n$, possibly with repeats. The first sum is over the possible numbers of Kronecker deltas 0 to $\lfloor \frac{k'}{2} \rfloor$, where $\lfloor \cdot \rfloor$ is the floor function. The second sum is over the possible permutations of the $k'$ axes, which is smaller than $S_{k'}$ when $t > 0$ due to the symmetries of $\delta^{\otimes t}$ as discussed in Appendix D. The third sum is over choosing $k' - 2t$ vectors from $v_1$ to $v_n$, allowing repeated vectors. See Figure 1 for an example. Directly evaluating the function $f(v_1, \ldots, v_n)$ defined in (11) has computational complexity $\mathcal{O}\left(k'! n^{k'} \left(Q d n^2 + d^{k'}\right)\right)$ operations, where $Q$ is the maximum number of operations needed to evaluate the polynomials $q_{t,\sigma,J}$. Thus, evaluating $f$ is only practical for small values of $k'$; however, since $k'$ is the rank of the output tensor, $k' \in \{1, 2, 3, 4\}$ already captures many cases of practical interest.

**Remark 1.** Note that Corollary 1 characterizes polynomial functions, but if we allow the $q_{t,\sigma,J}$ to be more general (e.g. in the class of continuous or smooth functions), then we obtain a parameterization of a larger class of $O(d)$-equivariant functions. In the experiments in Section 5, we set the $q_{t,\sigma,J}$ to be learnable multi-layer perceptrons (MLPs). We are unsure if a characterization of this sort can be stated for all continuous $O(d)$-equivariant functions. However, by the Stone–Weierstrass theorem any continuous function can be approximated by a polynomial function to arbitrary accuracy on any fixed compact set, so constructing an architecture that can represent equivariant polynomial functions is sufficient to approximately represent equivariant continuous functions (see Yarotsky (2022)).

The following example shows how to express a given equivariant polynomial in terms of invariant functions and tensors. A longer example appears in Appendix E.

**Example 1.** Let $f : \mathcal{T}_1(\mathbb{R}^d, +) \to \mathcal{T}_2(\mathbb{R}^d, +)$ be an $O(d)$-equivariant polynomial of degree at most 2. By Theorem 1, we can write $f$ in the form

$$f(a) = \iota_0\left(a^{\otimes 0} \otimes c_0\right) + \iota_1\left(a^{\otimes 1} \otimes c_1\right) + \iota_2\left(a^{\otimes 2} \otimes c_2\right), \tag{12}$$

where $c_r$ is an $O(d)$-isotropic $(r+2)_{(+)}$-tensor for $r = 0, 1, 2$. Lemma 3 characterizes such isotropic tensors, $c_0 = \beta_0 \delta$, $c_1 = 0$ is trivial, and $c_2$ is a linear combination of $(\delta^{\otimes 2})^\sigma$ for $\sigma \in G_4 = \{\sigma_1, \sigma_2, \sigma_3\}$ where $\sigma_1 := (1, 2, 3, 4), \sigma_2 = (1, 3, 2, 4), \sigma_3 = (1, 3, 4, 2)$, See Appendix D for the definition of $G_4$ (109).

Thus the final term $\iota_2\left(a^{\otimes 2} \otimes c_2\right)$ is

$$\iota_2\left(a^{\otimes 2} \otimes \left(\beta_1(\delta^{\otimes 2})^{\sigma_1} + \beta_2(\delta^{\otimes 2})^{\sigma_2} + \beta_3(\delta^{\otimes 2})^{\sigma_3}\right)\right) = \beta_1\langle a, a\rangle\delta + \beta_2 a \otimes a + \beta_3 a \otimes a, \tag{13}$$

where the terms associated with $\beta_2$ and $\beta_3$ are the same due to the symmetry of $a^{\otimes 2}$. We conclude

$$f(a) = \beta_0\delta + \beta_1\langle a, a\rangle\delta + \beta_2 a \otimes a, \tag{14}$$

for some scalars $\beta_0, \beta_1$, and $\beta_2$.

When the input and output are both symmetric $2_{(+)}$-tensors, we get another useful corollary.

**Corollary 2.** Let $f : \mathcal{T}_2^{sym}(\mathbb{R}^d, +) \to \mathcal{T}_2^{sym}(\mathbb{R}^d, +)$ be an $O(d)$-equivariant function. Then there exists a function $\widetilde{f} : \mathbb{R}_{\text{diag}}^{d \times d} \to \mathbb{R}_{\text{diag}}^{d \times d}$ of diagonal matrices that is permutation equivariant such that for all $A \in \mathcal{T}_2^{sym}(\mathbb{R}^d, +)$, $f(A) = Q\left(\widetilde{f}(\Lambda)\right)Q^\top$, where $A = Q\Lambda Q^\top$ is the eigenvalue decomposition.

In other words, an $O(d)$-equivariant function of symmetric matrices can be reduced to a permutation-equivariant function of the eigenvalues. The proof is given in Appendix F.

## 4  GENERALIZATIONS TO OTHER GROUPS

The results regarding $O(d)$-equivariant tensor maps from Section 3 are a particular case of a more general result involving algebraic groups, such as the Lorentz and symplectic groups we cover here. We work the full generalization in Appendix G where we give all the details of the proofs.

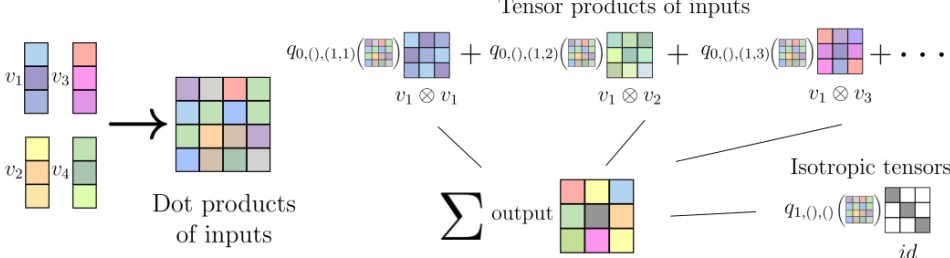

Figure 1: Illustration of the method from Corollary 1 with 4 input vectors in $\mathbb{R}^3$ and a $2_{(+)}$-tensor output. The tensor product of inputs includes all 16 possible tensor products of ordered pairs of input vectors, plus the isotropic Kronecker delta, labeled $id$. The coefficients $q_{t,\sigma,J}$ shown here use $\sigma = ()$, the identity permutation in $S_{k'}$.

Recall that $O(d)$ is the subgroup of linear transformations preserving the Euclidean inner product. However, in some contexts, we might be interested in preserving other bilinear products on $\mathbb{R}^d$, such as the *Minkowski inner product*

$$\langle u, v \rangle_s := u^\top \mathbb{I}_{s,d-s} v,$$

where $\mathbb{I}_{s,d-s} := \begin{pmatrix} \mathbb{I}_s & \\ & -\mathbb{I}_{d-s} \end{pmatrix}$, or, for $d$ even, the *symplectic product*

$$\langle u, v \rangle_{\text{symp}} := u^\top J_d v$$

where $J_d := \begin{pmatrix} & \mathbb{I}_{d/2} \\ -\mathbb{I}_{d/2} & \end{pmatrix}$. The subgroups of linear maps preserving these bilinear products give respectively the *indefinite orthogonal group* (which is the linear part of the *Lorentz group* when $d = 4$ and $s \in \{1, 3\}$) given by

$$O(s, d - s) := \{g \in \text{GL}(\mathbb{R}^d) \mid g^\top \mathbb{I}_{s,d-s} g = \mathbb{I}_{s,d-s}\}, \tag{15}$$

and, when $d$ is even, the *symplectic group* given by

$$Sp(d) := \{g \in \text{GL}(\mathbb{R}^d) \mid g^\top J_d g = J_d\}. \tag{16}$$

For any of these groups $G$, we can consider the modules $\mathcal{T}_k(\mathbb{R}^d, \chi) := (\mathbb{R}^d)^{\otimes k}$, where $\chi : G \to \mathbb{R}^*$ is an algebraic group homomorphism, where the action is given by the linear extension of

$$g \cdot (v_1 \otimes \cdots \otimes v_k) = \chi(g)(g \cdot v_1) \otimes \cdots \otimes (g \cdot v_k). \tag{17}$$

When $G = O(s, d - s)$, with $s \neq 0, d$, we have four possible $\chi$: $\chi_{+,+}$ being always equal to 1, $\chi_{+,-}$ being the sign of the determinant of the bottom-right $(d - s) \times (d - s)$ submatrix, $\chi_{-,+}$ being the sign of the determinant of the top-left $s \times s$ submatrix, and $\chi_{-,-}$ being the determinant of the matrix. Hence, we can represent them by $(p_1, p_2)$, where $p_i \in \{-1, +1\}$. When $G = Sp(d)$, we have that $\chi$ can only be the trivial group-homomorphism. (It follows, for instance, from the representation theory of simple Lie algebras from (Fulton & Harris, 2013, Part III) and a standard abelianization argument).

Additionally, we have $G$-equivariant contractions $\iota_k^G : \mathcal{T}_{2k+k'}(\mathbb{R}^d, \chi) \to \mathcal{T}_{k'}(\mathbb{R}^d, \chi)$ given by

$$\iota_k^{O(s,d-s)}(a) := [a]_{i_1,\ldots,i_k,j_1,\ldots,j_k,\ell_1,\ldots,\ell_{k'}} \, [\mathbb{I}_{s,d-s}]_{i_1,j_1} \cdots [\mathbb{I}_{s,d-s}]_{i_k,j_k} \tag{18}$$

and

$$\iota_k^{Sp(d)}(a) := [a]_{i_1,\ldots,i_k,j_1,\ldots,j_k,\ell_1,\ldots,\ell_{k'}} \, [J_d]_{i_1,j_1} \cdots [J_d]_{i_k,j_k}. \tag{19}$$

Under these notations, we can state the generalization of Theorem 1 as follows. Recall that an *entire function* is a function that is analytic and whose Taylor series converges globally at any point.

**Theorem 2.** Let $G$ be either $O(s, d - s)$ or $Sp(d)$ and $f : \prod_{i=1}^n \mathcal{T}_{k_i}(\mathbb{R}^d, \chi_i) \to \mathcal{T}_{k'}(\mathbb{R}^d, \chi')$ be a $G$-equivariant entire function. Then we may write $f$ as follows:

$$f(a_1, \ldots, a_n) = \sum_{r=0}^{\infty} \sum_{1 \leq \ell_1 \leq \cdots \leq \ell_r \leq n} \iota_{k_{\ell_1,\ldots,\ell_r}}^G \left( a_{\ell_1} \otimes \ldots \otimes a_{\ell_r} \otimes c_{\ell_1,\ldots,\ell_r} \right) \tag{20}$$

| dataset size | MLP baseline | MLP augmented | TFENN | Ours |
|---|---|---|---|---|
| n = 5,000 | 1.586e-4 $\pm$ 2.307e-6 | 2.020e-5 $\pm$ 2.141e-7 | 5.3e-5 | **4.057e-6 $\pm$ 3.458e-7** |
| n = 20,000 | 4.014e-5 $\pm$ 1.476e-6 | 9.365e-6 $\pm$ 1.584e-7 | 3.0e-5 | **7.748e-7 $\pm$ 2.911e-7** |
| n = 40,000 | 2.766e-5 $\pm$ 8.134e-7 | 7.516e-6 +- 1.457e-7 | 1.0e-5 | **3.310e-6 $\pm$ 8.448e-7** |

Table 1: Test error comparison on the TFENN Garanger et al. (2024) dataset, averaged over 5 trials with standard deviation given as $\pm 0.xxx$. The metric is the squared Frobenius norm of the difference of the predicted and target $2_{(+)}$-tensor, so lower values are better. For each row, the best value is **bolded**. The TFENN errors are the results reported in Garanger et al. (2024).

where $c_{\ell_1,\dots,\ell_r} \in \mathcal{T}_{k_{\ell_1,\dots,\ell_r}+k'}(\mathbb{R}^d, \chi_{\ell_1,\dots,\ell_r}\chi')$ is a $G$-isotropic tensor, i.e., a tensor in $\mathcal{T}_{k_{\ell_1,\dots,\ell_r}+k'}(\mathbb{R}^d, \chi_{\ell_1,\dots,\ell_r}\chi')$ invariant under the action of $G$; for $k_{\ell_1,\dots,\ell_r} := \sum_{q=1}^r k_{\ell_q}$ and $\chi_{\ell_1,\dots,\ell_r} = \prod_{q=1}^r \chi_{\ell_q}$.

Using the above theorem and an analogous version of Lemma 3 that characterizes $G$-isotropic tensors (see Proposition 7 in Appendix G or (Roe Goodman, 2009, Theorem 5.3.3)), we can then prove the following corollary, which generalizes Corollary 1.

**Corollary 3.** Let $G$ be either $O(s, d-s)$ or $Sp(d)$ and $f : \prod_{i=1}^n \mathcal{T}_1(\mathbb{R}^d, \chi_0) \to \mathcal{T}_k(\mathbb{R}^d, \chi_0)$, with $\chi_0$ the constant map to 1, be a $G$-equivariant entire function. Then we may write $f$ as follows:

$$f(v_1,\dots,v_n) = \sum_{t=0}^{\lfloor \frac{k}{2}\rfloor} \sum_{\sigma \in S_k} \sum_{1 \le J_1 \le \dots \le J_{k-2t} \le n} q_{t,\sigma,J}\left(\left(\langle v_i, v_j\rangle_G\right)_{i,j=1}^n\right)\left(v_{J_1} \otimes \dots \otimes v_{J_{k-2t}} \otimes \theta_G^{\otimes t}\right)^\sigma$$

(21)

where $\langle \cdot, \cdot \rangle_G = \langle \cdot, \cdot \rangle_s$ and $\theta_G = [\mathbb{I}_{s,d-s}]_{i,j}$ if $G = O(s, d-s)$, and $\langle \cdot, \cdot \rangle_G = \langle \cdot, \cdot \rangle_{\text{symp}}$ and $\theta_G = [J_d]_{i,j}$ if $G = Sp(d)$, and $q_{t,\sigma,J}$ is an entire function that depends on the tuple $(t, \sigma, J)$ and whose inputs are all possible inner products between the input vectors and whose output is a scalar.

## 5 NUMERICAL EXPERIMENTS

With the preceding theory in place, we can build machine learning models to learn equivariant tensor functions. We use Corollaries 1 and 3 which characterize the $O(d)$- and Lorentz-equivariant functions from vectors to tensors, and Corollary 2 which characterizes $O(d)$-equivariant functions from symmetric $2_{(+)}$-tensors to symmetric $2_{(+)}$-tensors.

**Stress-Strain Tensors.** We consider the problem materials science from Garanger et al. (2024). We can model an isotropic neo-Hookean hyperelastic material with the equation

$$W = \frac{\lambda}{2}(\log \det F)^2 - \mu \log \det F + \frac{\mu}{2}\left(\text{tr}\left(F^\top F\right) - 3\right),$$

(22)

where $\lambda, \mu$ are model parameters and $F$ is a random deformation gradient. For the Cauchy-Green strain tensor $C = F^\top F$, the second Piola-Kirchoff stress tensor is given by

$$S = \left(\frac{1}{2}\lambda \log \det C - \mu\right)C^{-1} + \mu \mathbb{I}_d.$$

(23)

Thus $S$ is a function of $C$. Both $S$ and $C$ are $2_{(+)}$-tensors, and the function is $O(d)$-equivariant, so we can parameterize this function by Corollary 2. We enforce permutation equivariance of the function of the eigenvalues by Maron et al. (2019). We compare our model to an MLP baseline, the MLP baseline trained on an augmented dataset with 4 random rotations, and the method from Garanger et al. (2024) which is also equivariant. The results are shown in Table 1. We can see that for all dataset sizes, our equivariant model performs dramatically better than the other models. See Appendix H for further model and training details.

**Path Signature.** Let $x : [0, T] \to \mathbb{R}^d$ be a continuous path of bounded variation. The path signature $S(x)$ is a sequence of tensors $S_0(x), S_1(x), S_2(x), \dots$, where $S_0(x) = 1$ and $S_k(x)$ for $k > 0$ is an $k_{(+)}$-tensor defined as:

$$[S_k(x)]_{i_1,\dots,i_k} = \int_{\vec{t} \in \Delta_k([0,T])} [\dot{x}_{t_1}]_{i_1} \cdots [\dot{x}_{t_k}]_{i_k} dt_1 \cdots dt_k,$$

(24)

| Group | Discrete (24) | MLP (same width) | MLP (same # params) | MLP augmented | Ours |
|-------|--------------|------------------|---------------------|---------------|------|
| $O(d)$ | 1.336 | $0.255 \pm 0.003$ | $0.071 \pm 0.001$ | 0.007 | **0.002** |
| Lorentz | 1.489 | $1.391 \pm 0.005$ | $0.450 \pm 0.002$ | $0.186 \pm 0.002$ | **0.005** |

Table 2: Path signature test performance averaged over 3 trials with standard deviation given as $\pm 0.xxx$ when it is at least 1e-3. The metric for true truncated signature $S_M(x)$ and predicted truncated signature $\hat{S}_M(x(t_1), \ldots, x(t_n))$ is $\ell\left(S_M(x), \hat{S}_M(x(t_1), \ldots, x(t_n))\right) = \frac{1}{M} \sum_{k=1}^{M} \frac{1}{d^k} \left\|S_k(x) - \hat{S}_k(x(t_1), \ldots, x(t_n))\right\|_F^2$ where $\|\cdot\|_F$ is the Frobenius norm, so lower values are better. For each row, the best value is **bolded**.

where $\dot{x}_{t_i} = \frac{dx}{dt}$, $\Delta_k([0, T]) := \{(t_1, \ldots, t_k) \in \mathbb{R}^m : 0 < t_1 < \ldots < t_k < T\}$ is the $k$-dimensional simplex, and the integral is in the sense of a Riemann-Stieljes integral.

The path signature is a useful object when working with path data because it nicely encodes the properties of a path (see Lyons et al. (2007)). For example, if $x, y$ are regular paths, then $S(x) = S(y)$ if and only if $x$ and $y$ are the same up to translation and reparameterization (Chen, 1958). This result generalizes to non-regular paths (Hambly & Lyons, 2010) and it's fundamental in reconstructing paths from signatures (Pfeffer et al., 2019; Rauscher et al., 2025). Furthermore, we can approximate any function on a path by a linear function on its path signature Chevyrev & Oberhauser (2022).

We will consider the problem of approximating the path signature from a small sample of points along the path, which if done well allows us to reconstruct the path (see for example (Pfeffer et al., 2019)). Let $\mathcal{P} = \left\{x : [0, T] \to \mathbb{R}^d\right\}$ be a family of paths. Let $n \in \mathbb{N}$ be fixed and small, and let $0 \le t_1 < \ldots < t_n \le T$ also be fixed. Suppose for $x \in \mathcal{P}$ that we know $x(t_1), \ldots, x(t_n)$, then the problem is to approximate the truncated signature $S_M(x) = \{S_k(x), 1 \le k \le M\}$.

Our baseline for comparison is the discrete version of (24) on the $n$ points. It is not hard to see that $\hat{S}_k(x(t_1), \ldots, x(t_n))$ is an $O(d)$-equivariant function from $n$ input vectors to a $k_{(+)}$-tensor, so we can parameterize it with Corollary 1. All the $q_{t, \sigma J}$ functions are learned as a single, shared MLP. Further, it is also equivariant under the Lorentz and symplectic groups, making it well-behaved in more general physical settings. We compare against three baseline MLP methods, one with the same width as our method, one with the same number of parameters, and one trained on an augmented dataset with 4 random transformations. Further architecture and training details are in Appendix I.2.

The results are shown in Table 2. We generate paths for both the orthogonal group and the Lorentz group, see Appendix I.1. The learned methods perform better than the fixed naive methods, and our method does the best of all. This method would be a viable first step for processing path data for any downstream learning problem.

**Sparse Vector Estimation.** We consider the problem of finding a planted sparse vector in a linear subspace. This problem was introduced in Spielman et al. (2012) in the context of dictionary learning. The works Hopkins et al. (2016) and Mao & Wein (2022) developed state-of-the-art methods with theoretical guarantees based on sum-of-squares (SoS). The results in this section suggest (i) the SoS methods outperform learning-based methods when the (strong) SoS assumptions are met, (ii) equivariant tensor learning can be used to learn models with good performance when the SoS assumptions are not met, and (iii) equivariant tensor learning outperforms standard machine learning models where no structure is imposed.

The problem is defined as follows. Let $v \in \mathbb{R}^n$ be an (approximately) sparse vector of unit length, construct $v_0, \ldots, v_{d-1} \in \mathbb{R}^n$ by adding noise to $v$ according to the procedure described in Appendix J.4, then consider $S$ to be an $n \times d$ matrix whose columns form a random orthonormal basis of span$\{v_0, \ldots, v_{d-1}\}$. The goal is to recover $v$ from $S$. The SoS methods consider explicit maps $h : (\mathbb{R}^d)^n \to \mathcal{S}_d$ that take the rows of $S$ (denoted by $a_1^\top, \ldots a_n^\top \in \mathbb{R}^d$) and output a $d \times d$ symmetric matrix. The estimator for $v$ is the multiplication of $S$ times the top eigenvector of $h(a_1, \ldots, a_n)$:

$$\hat{v} = S \, \lambda_{\text{vec}}(h(a_1, \ldots, a_n)) \,. \tag{25}$$

The SoS methods have theoretical guarantees under strict assumptions described in Appendix J.5. Here we learn a function $h$ from data and we compare the learned model with SoS methods and non-equivariant learned baselines for settings that do and do not satisfy the theoretical assumptions (different sampling methods for $v_0, \ldots, v_n$ described in Appendix J.3). Our method uses Corollary 1

| sampling | $\Sigma$ | SoS | MLP baseline | Ours (Diag) | Ours |
|---|---|---|---|---|---|
| Accept/Reject | Random | $0.610 \pm 0.009$ | $0.241 \pm 0.019$ | $0.493 \pm 0.005$ | $\mathbf{0.938 \pm 0.002}$ |
| | Diagonal | $0.448 \pm 0.012$ | $0.196 \pm 0.011$ | $\mathbf{0.589 \pm 0.026}$ | $0.465 \pm 0.027$ |
| | Identity | $\mathbf{0.606 \pm 0.014}$ | $0.196 \pm 0.008$ | $0.351 \pm 0.065$ | $0.190 \pm 0.008$ |
| Bernoulli-Gaussian | Random | $\mathbf{0.962 \pm 0.002}$ | $0.242 \pm 0.006$ | $0.917 \pm 0.004$ | $0.937 \pm 0.002$ |
| | Diagonal | $\mathbf{0.949 \pm 0.005}$ | $0.205 \pm 0.013$ | $0.914 \pm 0.006$ | $0.463 \pm 0.018$ |
| | Identity | $\mathbf{0.962 \pm 0.002}$ | $0.196 \pm 0.009$ | $0.908 \pm 0.006$ | $0.342 \pm 0.043$ |
| Corrected Bernoulli-Gaussian | Random | $0.412 \pm 0.017$ | $0.239 \pm 0.012$ | $0.372 \pm 0.011$ | $\mathbf{0.935 \pm 0.002}$ |
| | Diagonal | $0.288 \pm 0.018$ | $0.206 \pm 0.003$ | $\mathbf{0.550 \pm 0.026}$ | $0.460 \pm 0.022$ |
| | Identity | $\mathbf{0.412 \pm 0.011}$ | $0.198 \pm 0.005$ | $0.239 \pm 0.025$ | $0.197 \pm 0.011$ |
| Bernoulli-Rademacher | Random | $0.526 \pm 0.020$ | $0.923 \pm 0.004$ | $0.437 \pm 0.034$ | $\mathbf{0.957 \pm 0.001}$ |
| | Diagonal | $0.334 \pm 0.024$ | $0.864 \pm 0.005$ | $0.588 \pm 0.011$ | $\mathbf{0.903 \pm 0.004}$ |
| | Identity | $0.524 \pm 0.010$ | $0.845 \pm 0.006$ | $0.317 \pm 0.046$ | $\mathbf{0.889 \pm 0.003}$ |

Table 3: Test error comparison on synthetic data averaged over 5 trials ($n = 100, d = 5, \epsilon = 0.25$) with the standard deviation given by $\pm 0.xxx$. The metric $\langle v, \hat{v} \rangle^2$ ranges from 0 to 1 with 1 indicating the estimate $\hat{v}$ identical to the true $v$. For each row, the best value is **bolded**. The SoS methods perform best when their assumptions are met, such as identity covariance for the noise vectors, but perform worse than our learned models when using Random or Diagonal covariances. One exception to this trend is the Bernoulli-Gaussian sampling. This is likely because in expectation the BG satisfies the sparsity requirements of SoS by a large margin (see Appendix J.3). By contrast, the Corrected Bernoulli-Gaussian has lower sparsity and the learned models perform better. Finally, we see that in all experiments, the baseline MLP generalizes poorly, despite doing well on the training data (Table 7 in Appendix). This is consistent with the claim that enforcing symmetries improves generalization performance.

to learn $h$ (see Appendix J.2 for an explanation of why this problem is $O(d)$-equivariant). We also include a variant which only takes the norms of each vector as input, instead of all the pairwise cross products. See Appendix J.5 for the details of all models. The results are displayed in Table 3.

## 6 DISCUSSION

This paper provides a full characterization of polynomial functions from multiple tensor inputs to tensor outputs that are equivariant with respect to the diagonal action by classical Lie groups, including the orthogonal group, the symplectic group, and the Lorentz group.

Our main goal is to define equivariant machine learning models. To the best of our knowledge this is the first work that provides a recipe for equivariant machine learning models for tensors at this level of generality. We apply the resulting models to time series data, stress and strain tensors, and sparse vectors. The equivariant models outperform all non-equivariant baseline models, and in the case of the sparse vector problem, the learned models can operate in settings where theoretical guarantees have yet to be developed.

**Acknowledgements** The authors would like to thank Teresa Huang, Ben Blum-Smith, and Daniel Packer for helpful discussions. WGG and JTC are thankful to Jazz G. Suchen for useful suggestions regarding Appendix G. WGG and SV are partially supported by NSF CCF 2212457 and the NSF–Simons Research Collaboration on the Mathematical and Scientific Foundations of Deep Learning (MoDL) (NSF DMS 2031985). SV is also funded by NSF CAREER 2339682 and NSF BSF 2430292. JTC, NFM, ASL and SV are thankful to the 2022 *Mathematics Research Communities* program of the American Mathematical Society, particularly the MRC Conference "Data Science at the Crossroads of Analysis, Geometry, and Topology" that took place from May 29th to June 4th of 2022, for allowing them to start collaborating. JTC thanks SV, WGG and the other people at Hopkins for the wonderful working environment when writing this (and other) work(s).

**Reproducibility** All our code is open source, and it is available at: `https://github.com/WilsonGregory/TensorPolynomials`. All datasets are synthetically generated by our code, or public. We also include all necessary experimental details in appendices H, I, and J.

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

# A    BASIC PROPERTIES OF $O(d)$ ACTIONS ON TENSORS

In this section, we will show that the basic operations are $O(d)$-equivariant and linear by direct computation. We do so explicitly by performing routine computations. However, the universal property of tensor products, which we use in Appendix G, would give immediate proofs of these statements.

**Proposition 1.** The outer product is a $O(d)$-equivariant bilinear map. In other words, for $g \in O(d)$, $a, a' \in \mathcal{T}_k(\mathbb{R}^d, p)$, $b, b' \in \mathcal{T}_{k'}(\mathbb{R}^d, p')$ and $\alpha, \beta \in \mathbb{R}$, we have $g \cdot (a \otimes b) = (g \cdot a) \otimes (g \cdot b)$, $(\alpha a + \beta a') \otimes b = \alpha(a \otimes b) + \beta(a' \otimes b)$, and $a \otimes (\alpha b + \beta b') = \alpha(a \otimes b) + \beta(a \otimes b')$. In particular, if $c \in \mathcal{T}_{k'}(\mathbb{R}^d, p')$ is an $O(d)$-isotropic tensor, then the function mapping

$$\mathcal{T}_k(\mathbb{R}^d, p) \to \mathcal{T}_{k+k'}(\mathbb{R}^d, pp') \quad \text{by} \quad a \mapsto a \otimes c \tag{26}$$

is an $O(d)$-equivariant linear map.

**Proposition 2.** The $k$-contraction $\iota_k : \mathcal{T}_{2k+k'}(\mathbb{R}^d, p) \to \mathcal{T}_{k'}(\mathbb{R}^d, p)$ (def. 4) is an $O(d)$-equivariant linear map.

**Proposition 3.** For fixed $\sigma \in S_k$, the tensor index permutation mapping $\mathcal{T}_k(\mathbb{R}^d, p) \to \mathcal{T}_k(\mathbb{R}^d, p)$ by $a \mapsto a^\sigma$ is an $O(d)$-equivariant linear map.

*Proof of Proposition 1.* First, we establish equivariance. Let $a \in \mathcal{T}_k(\mathbb{R}^d, p)$, $b \in \mathcal{T}_{k'}(\mathbb{R}^d, p')$, and $g \in O(d)$. We have

$$[g \cdot (a \otimes b)]_{j_1, \ldots, j_{k+k'}}$$

$$= \det(M(g))^{\frac{1-p\,p'}{2}} [(a \otimes b)]_{i_1, \ldots, i_{k+k'}} [M(g)]_{j_1, i_1} \cdots [M(g)]_{j_{k+k'}, i_{k+k'}}$$

$$= \det(M(g))^{\frac{1-p}{2}} \det(M(g))^{\frac{1-p'}{2}} [a]_{i_1, \ldots, i_k} [b]_{i_{k+1}, \ldots, i_{k+k'}} [M(g)]_{j_1, i_1} \cdots$$
$$\cdots [M(g)]_{j_k, i_k} [M(g)]_{j_{k+1}, i_{k+1}} \cdots [M(g)]_{j_{k+k'}, i_{k+k'}}$$

$$= \left( \det(M(g))^{\frac{1-p}{2}} [a]_{i_1, \ldots, i_k} [M(g)]_{j_1, i_1} \cdots [M(g)]_{j_k, i_k} \right)$$
$$\left( \det(M(g))^{\frac{1-p'}{2}} [b]_{i_{k+1}, \ldots, i_{k+k'}} [M(g)]_{j_{k+1}, i_{k+1}} \cdots [M(g)]_{j_{k+k'}, i_{k+k'}} \right)$$

$$= [g \cdot a]_{j_1, \ldots, j_k} [g \cdot b]_{j_{k+1}, \ldots, j_{k+k'}}$$

$$= [g \cdot a \otimes g \cdot b]_{j_1, \ldots, j_{k+k'}} \, ,$$

where the second equality uses the fact that

$$\det(M(g))^{\frac{1-p\,p'}{2}} = \det(M(g))^{\frac{1-p}{2}} \det(M(g))^{\frac{1-p'}{2}},$$

which is straightforward to verify via a case analysis over possible parameter values (i.e., $p, p' \in \{+1, -1\}$ and $\det(M(g)) \in \{+1, -1\}$).

Next, we verify linearity. Let $a, a' \in \mathcal{T}_k(\mathbb{R}^d, p)$, $b \in \mathcal{T}_{k'}(\mathbb{R}^d, p')$, and $\alpha, \beta \in \mathbb{R}$. Then,

$$[(\alpha a + \beta a') \otimes b]_{i_1, \ldots, i_{k+k'}} = [(\alpha a + \beta a')]_{i_1, \ldots, i_k} [b]_{i_{k+1}, \ldots, i_{k+k'}} \tag{27}$$

$$= \alpha [a]_{i_1, \ldots, i_k} [b]_{i_{k+1}, \ldots, i_{k+k'}} + \beta [a']_{i_1, \ldots, i_k} [b]_{i_{k+1}, \ldots, i_{k+k'}} \tag{28}$$

$$= \alpha [a \otimes b]_{i_1, \ldots, i_{k+k'}} + \beta [a' \otimes b]_{i_1, \ldots, i_{k+k'}}. \tag{29}$$

The linearity in the second argument follows in the same manner.

Finally, (26) follows immediately from the fact that the bilinear $O(d)$-equivariance. ☐

*Proof of Proposition 2.* To establish equivariance, let $a \in \mathcal{T}_{2k+k'}(\mathbb{R}^d, p)$ and let $g \in O(d)$. Then,

$$[g \cdot \iota_k(a)]_{j_1,\ldots,j_{k'}} \tag{30}$$

$$= \det(M(g))^{\frac{1-p}{2}} [a]_{\ell_1,\ldots,\ell_k,\ell_1,\ldots,\ell_k,i_1,\ldots,i_{k'}} [M(g)]_{j_1,i_1} \cdots [M(g)]_{j_{k'},i_{k'}} \tag{31}$$

$$= \det(M(g))^{\frac{1-p}{2}} [a]_{\ell_1,\ldots,\ell_{2k},i_1,\ldots,i_{k'}} [\delta]_{\ell_1,\ell_{k+1}} \cdots [\delta]_{\ell_k,\ell_{2k}} [M(g)]_{j_1,i_1} \cdots [M(g)]_{j_{k'},i_{k'}} \tag{32}$$

$$= \det(M(g))^{\frac{1-p}{2}} [a]_{\ell_1,\ldots,\ell_{2k},i_1,\ldots,i_{k'}} [M(g)]_{\ell_1,m_1} [M(g)]_{\ell_{k+1},m_1} \cdots \tag{33}$$

$$\cdots [M(g)]_{\ell_k,m_k} [M(g)]_{\ell_{2k},m_k} [M(g)]_{j_1,i_1} \cdots [M(g)]_{j_{k'},i_{k'}} \tag{34}$$

$$= [g \cdot a]_{m_1,\ldots,m_k,m_1,\ldots,m_k,j_1,\ldots,j_{k'}} \tag{35}$$

$$= [\iota_k(g \cdot a)]_{j_1,\ldots,j_{k'}}, \tag{36}$$

where the third equality uses the fact that $\delta = M(g) M(g)^\top$.

Next, to establish linearity, let $a, b \in \mathcal{T}_{2k+k'}(\mathbb{R}^d, p)$ and let $\alpha, \beta \in \mathbb{R}$. Then,

$$[\iota_k(\alpha a + \beta b)]_{j_1,\ldots,j_{k'}} = [\alpha a + \beta b]_{i_1,\ldots,i_k,i_1,\ldots,i_k,j_1,\ldots,j_{k'}} \tag{37}$$

$$= \alpha [a]_{i_1,\ldots,i_k,i_1,\ldots,i_k,j_1,\ldots,j_{k'}} + \beta [b]_{i_1,\ldots,i_k,i_1,\ldots,i_k,j_1,\ldots,j_{k'}} \tag{38}$$

$$= \alpha [\iota_k(a)]_{j_1,\ldots,j_{k'}} + \beta [\iota_k(b)]_{j_1,\ldots,j_{k'}}. \tag{39}$$

This completes the proof. $\qquad\square$

*Proof of Proposition 3.* Fix $\sigma \in S_k$. To establish equivariance, let $a \in \mathcal{T}_k(\mathbb{R}^d, p)$ and $g \in O(d)$. Then,

$$[g \cdot (a^\sigma)]_{j_1,\ldots,j_k} = \det(M(g))^{\frac{1-p}{2}} [a^\sigma]_{i_1,\ldots,i_k} [M(g)]_{j_1,i_1} \cdots [M(g)]_{j_k,i_k} \tag{40}$$

$$= \det(M(g))^{\frac{1-p}{2}} [a]_{i_{\sigma^{-1}(1)},\ldots,i_{\sigma^{-1}(k)}} [M(g)]_{j_1,i_1} \cdots [M(g)]_{j_k,i_k} \tag{41}$$

$$= \det(M(g))^{\frac{1-p}{2}} [a]_{i_{\sigma^{-1}(1)},\ldots,i_{\sigma^{-1}(k)}} [M(g)]_{j_{\sigma^{-1}(1)},i_{\sigma^{-1}(1)}} \cdots [M(g)]_{j_{\sigma^{-1}(k)},i_{\sigma^{-1}(k)}} \tag{42}$$

$$= [g \cdot a]_{j_{\sigma^{-1}(1)},\ldots,j_{\sigma^{-1}(k)}} \tag{43}$$

$$= [(g \cdot a)^\sigma]_{j_1,\ldots,j_k}, \tag{44}$$

$$\tag{45}$$

where the third equality holds since we are merely reordering the $M(g)$ components—which is allowed because they are scalars.

To show linearity, let $a, b \in \mathcal{T}_k(\mathbb{R}^d, p)$ and $\alpha, \beta \in \mathbb{R}$. We have

$$[(\alpha a + \beta b)^\sigma]_{i_1,\ldots,i_k} = [\alpha a + \beta b]_{i_{\sigma^{-1}(1)},\ldots,i_{\sigma^{-1}(k)}} \tag{46}$$

$$= \alpha [a]_{i_{\sigma^{-1}(1)},\ldots,i_{\sigma^{-1}(k)}} + \beta [b]_{i_{\sigma^{-1}(1)},\ldots,i_{\sigma^{-1}(k)}} \tag{47}$$

$$= \alpha [a^\sigma]_{i_1,\ldots,i_k} + \beta [b^\sigma]_{i_1,\ldots,i_k}. \tag{48}$$

$$\square$$

## B  PROOF OF THEOREM 1

The main idea of the proof of Theorem 1 is to write out the polynomial $f$ in a way that takes advantage of the tensor operations of Section 2, then show that each term must be $O(d)$-equivariant (Lemmas 1 and 2). We then use a group averaging argument to show that $c_{\ell_1,\ldots,\ell_r}$ can be written as an $O(d)$-isotropic tensor. We state the lemmas, prove the theorem, then prove the lemmas.

**Lemma 1.** Let $f : \prod_{i=1}^n \mathcal{T}_{k_i}(\mathbb{R}^d, p_i) \to \mathcal{T}_{k'}(\mathbb{R}^d, p')$ be a polynomial map of degree $R$, and write

$$f(a_1, \ldots, a_n) = \sum_{r=0}^R f_r(a_1, \ldots, a_n),$$

where $f_r : \prod_{i=1}^n \mathcal{T}_{k_i}(\mathbb{R}^d, p_i) \to \mathcal{T}_{k'}(\mathbb{R}^d, p')$ is homogeneous degree $r$ polynomial. If $f$ is $O(d)$-equivariant, then each $f_r$ is $O(d)$-equivariant.

**Lemma 2.** Let $f_r : \prod_{i=1}^n \mathcal{T}_{k_i}(\mathbb{R}^d, p_i) \to \mathcal{T}_{k'}(\mathbb{R}^d, p')$ be a homogeneous polynomial of degree $r$. Then, we can write $f_r$ as

$$f_r(a_1, \ldots, a_n) = \sum_{1 \le \ell_1 \le \ldots \le \ell_r \le n} f_{\ell_1, \ldots, \ell_r}(a_{\ell_1}, \ldots, a_{\ell_r}), \tag{49}$$

where $f_{\ell_1, \ldots, \ell_r} : \prod_{i=1}^r \mathcal{T}_{k_{\ell_i}}(\mathbb{R}^d, p_{\ell_i}) \to \mathcal{T}_{k'}(\mathbb{R}^d, p')$ is the composition of the map

$$\prod_{i=1}^r \mathcal{T}_{k_{\ell_i}}(\mathbb{R}^d, p_{\ell_i}) \to \mathcal{T}_{\sum_{i=1}^r k_{\ell_i}}\left(\mathbb{R}^d, \prod_{i=1}^r p_{\ell_i}\right) \tag{50}$$

$$(a_{\ell_1}, \ldots, a_{\ell_r}) \mapsto a_{\ell_1} \otimes \ldots \otimes a_{\ell_r} \tag{51}$$

with a linear map $\mathcal{T}_{\sum_{i=1}^r k_{\ell_i}}(\mathbb{R}^d, \prod_{i=1}^r p_{\ell_i}) \to \mathcal{T}_{k'}(\mathbb{R}^d, p')$.

Moreover, if $f_r$ is $O(d)$-equivariant, then so are the $f_{\ell_1, \ldots, \ell_r}$.

**Remark 2.** Note that Lemma 2 is nothing more than the decomposition of $f_r$ as a sum of multihomogeneous maps in the inputs $a_1, \ldots, a_n$.

*Proof of Theorem 1.* Combining Lemmas 1 and 2, we can write $f$ as follows:

$$f(a_1, \ldots, a_n) = \sum_{r=0}^R \sum_{1 \le \ell_1 \le \ldots \le \ell_r \le n} f_{\ell_1, \ldots, \ell_r}(a_{\ell_1}, \ldots, a_{\ell_r}), \tag{52}$$

where the $f_{\ell_1, \ldots, \ell_r}$ is the composition of a linear map $\mathcal{T}_{k_{\ell_1, \ldots, \ell_r}}(\mathbb{R}^d, p_{\ell_1, \ldots, \ell_r}) \to \mathcal{T}_{k'}(\mathbb{R}^d, p')$ with the map $(a_1, \ldots, a_\ell) \mapsto a_{\ell_1} \otimes \cdots \otimes a_{\ell_r}$. Recall that $k_{\ell_1, \ldots, \ell_r} = \sum_{q=1}^r k_{\ell_q}$ and $p_{\ell_1, \ldots, \ell_r} = \prod_{q=1}^r p_{\ell_q}$. Moreover, by the lemmas, each $f_{\ell_1, \ldots, \ell_r}$ is $O(d)$-equivariant. Hence, without loss of generality, it is enough to prove the theorem in the special case

$$f(a_1, \ldots, a_n) = \lambda(a_{\ell_1} \otimes \cdots \otimes a_{\ell_r}), \tag{53}$$

where $\lambda : \mathcal{T}_{\sum_{i=1}^r k_{\ell_i}}(\mathbb{R}^d, \prod_{i=1}^r p_{\ell_i}) \to \mathcal{T}_{k'}(\mathbb{R}^d, p')$ is linear.

Now, in coordinates, we can write this map as

$$[f(a_1, \ldots, a_n)]_{j_1, \ldots, j_{k'}} = \lambda_{i_1, \ldots, i_{k_{\ell_1, \ldots, \ell_r}}, j_1, \ldots, j_{k'}} [a_{\ell_1} \otimes \cdots \otimes a_{\ell_r}]_{i_1, \ldots, i_{k_{\ell_1, \ldots, \ell_r}}}. \tag{54}$$

Consider now the tensor $c \in \mathcal{T}_{k_{\ell_1, \ldots, \ell_r} + k'}(\mathbb{R}^d, p_{\ell_1, \ldots, \ell_r} p')$ given by

$$[c]_{i_1, \ldots, i_{k_{\ell_1, \ldots, \ell_r} + k'}} = \lambda_{i_1, \ldots, i_{k_{\ell_1, \ldots, \ell_r}}, i_{k_{\ell_1, \ldots, \ell_r} + 1}, \ldots, i_{k_{\ell_1, \ldots, \ell_r} + k'}} \tag{55}$$

Then we have that

$$[f(a_1, \ldots, a_n)]_{j_1, \ldots, j_{k'}} = [c]_{i_1, \ldots, i_{k_{\ell_1, \ldots, \ell_r}}, j_1, \ldots, j_{k'}} [a_{\ell_1} \otimes \cdots \otimes a_{\ell_r}]_{i_1, \ldots, i_{k_{\ell_1, \ldots, \ell_r}}} \tag{56}$$

$$= [a_{\ell_1} \otimes \cdots \otimes a_{\ell_r} \otimes c]_{i_1, \ldots, i_{k_{\ell_1, \ldots, \ell_r}}, i_1, \ldots, i_{k_{\ell_1, \ldots, \ell_r}}, j_1, \ldots, j_{k'}} \tag{57}$$

$$= [\iota_{k_{\ell_1, \ldots, \ell_r}}(a_{\ell_1} \otimes \cdots \otimes a_{\ell_r} \otimes c)]_{j_1, \ldots, j_{k'}}, \tag{58}$$

after using the definition of $k$-contraction. Hence

$$f(a_1, \ldots, a_n) = \iota_{k_{\ell_1, \ldots, \ell_r}}(a_{\ell_1} \otimes \cdots \otimes a_{\ell_r} \otimes c). \tag{59}$$

Since $f$ is $O(d)$-equivariant, we have that for all $g \in O(d)$,

$$f(a_1, \ldots, a_n) = \iota_{k_{\ell_1, \ldots, \ell_r}}(a_{\ell_1} \otimes \cdots \otimes a_{\ell_r} \otimes g \cdot c). \tag{60}$$

To see this, we argue as follows:

$$f(a_1, \ldots, a_n) \tag{61}$$

$$= f(g \cdot (g^{-1} \cdot a_1), \ldots, g \cdot (g^{-1} \cdot a_n)) \tag{62}$$

$$= g \cdot f((g^{-1} \cdot a_1), \ldots, (g^{-1} \cdot a_n)) \qquad (f \ O(d)\text{-equivariant}) \tag{63}$$

$$= g \cdot \iota_{k_{\ell_1, \ldots, \ell_r}}((g^{-1} \cdot a_{\ell_1}) \otimes \cdots \otimes (g^{-1} \cdot a_{\ell_r}) \otimes c) \tag{64}$$

$$= \iota_{k_{\ell_1, \ldots, \ell_r}}(g \cdot (g^{-1} \cdot a_{\ell_1}) \otimes \cdots \otimes g \cdot (g^{-1} \cdot a_{\ell_r}) \otimes (g \cdot c)) \quad (\iota_{k_{\ell_1, \ldots, \ell_r}} \ O(d)\text{-equivariant}) \tag{65}$$

$$= \iota_{k_{\ell_1, \ldots, \ell_r}}(a_{\ell_1} \otimes \cdots \otimes a_{\ell_r} \otimes (g \cdot c)). \tag{66}$$

Hence, by taking the expectation with respect to the Haar probability measure of $O(d)$ and linearity of contractions, we have that

$$f(a_1, \ldots, a_n) = \iota_{k_{\ell_1, \ldots, \ell_r}} \left( a_{\ell_1} \otimes \cdots \otimes a_{\ell_r} \otimes \left( \underset{\mathfrak{g} \in O(d)}{\mathbb{E}} \, \mathfrak{g} \cdot c \right) \right), \tag{67}$$

where $\mathbb{E}_{\mathfrak{g} \in O(d)}$ is the expectation with respect the Haar probability measure of $O(d)$. This holds because

$$f(a_1, \ldots, a_n) = \underset{\mathfrak{g} \in O(d)}{\mathbb{E}} f(a_1, \ldots, a_n) \tag{68}$$

$$= \underset{\mathfrak{g} \in O(d)}{\mathbb{E}} \iota_{k_{\ell_1, \ldots, \ell_r}} (a_{\ell_1} \otimes \cdots \otimes a_{\ell_r} \otimes (\mathfrak{g} \cdot c)) \tag{69}$$

$$= \iota_{k_{\ell_1, \ldots, \ell_r}} \left( a_{\ell_1} \otimes \cdots \otimes a_{\ell_r} \otimes \left( \underset{\mathfrak{g} \in O(d)}{\mathbb{E}} \, \mathfrak{g} \cdot c \right) \right). \tag{70}$$

Now, $\mathbb{E}_{\mathfrak{g} \in O(d)} \, \mathfrak{g} \cdot c$ is an $O(d)$-isotropic tensor. Hence, we have shown that we can write $f$ in the desired form. $\qquad\square$

*Proof of Lemma 1.* Let $t \in \mathbb{R}$, since each $f_r$ is homogeneous of degree $r$, we have

$$f(t\, a_1, \ldots, t\, a_n) = \sum_{r=0}^{R} f_r(t\, a_1, \ldots, t\, a_n) = \sum_{r=1}^{R} t^r f_r(a_1, \ldots, a_n).$$

Let now $g \in O(d)$, then, by equivariance of $f$, we have

$$\sum_{r=0}^{R} t^r \, f_r(g \cdot a_1, \ldots, g \cdot a_n) = \sum_{r=0}^{R} t^r \, g \cdot f_r(a_1, \ldots, a_n), \tag{71}$$

since

$$\sum_{r=0}^{R} t^r \, f_r(g \cdot a_1, \ldots, g \cdot a_n) = f(t\,(g \cdot a_1), \ldots, t\,(g \cdot a_n)) \tag{72}$$

$$= f(g \cdot t\, a_1, \ldots, g \cdot t\, a_n) \tag{73}$$

$$= g \cdot f(t\, a_1, \ldots, t\, a_n) \tag{74}$$

$$= g \cdot \sum_{r=0}^{R} t^r f_r(a_1, \ldots, a_n) \tag{75}$$

$$= \sum_{r=0}^{R} t^r \, g \cdot f_r(a_1, \ldots, a_n). \tag{76}$$

Hence, for all $g \in O(d)$, $t \in \mathbb{R}$ and $(a_1, \ldots, a_n) \in \prod_{i=1}^{n} \mathcal{T}_{k_i}(\mathbb{R}^d, p_i)$, we have that

$$0 = \sum_{r=0}^{R} t^r \, (g \cdot f_r(a_1, \ldots, a_n) - f_r(g \cdot a_1, \ldots, g \cdot a_n)). \tag{77}$$

Now, the only way in which the univariate polynomial in $t$ of degree $R$ is identically zero is if it is the zero polynomial (cf. (Cox et al., 2015, Chapter 1 §1 Proposition 5)). Therefore for all $r \in \mathbb{N}$, $g \in O(d)$ and $(a_1, \ldots, a_n) \in \prod_{i=1}^{n} \mathcal{T}_{k_i}(\mathbb{R}^d, p_i)$,

$$f_r(g \cdot a_1, \ldots, g \cdot a_n) = g \cdot f_r(a_1, \ldots, a_n), \tag{78}$$

i.e., for each $r$, $f_r$ is $O(d)$-equivariant, as we wanted to show. $\qquad\square$

*Proof of Lemma 2.* First, we will show that if the decomposition exists, each summand is equivariant. Then, we will show that the decomposition exists.

Let $t_1, \ldots, t_n \in \mathbb{R}$. Then, by the linearity, we have that

$$f_r(t_1 a_1, \ldots, t_n a_n) = \sum_{1 \leq \ell_1 \leq \ldots \leq \ell_r \leq n} t_{\ell_1} \cdots t_{\ell_r} f_{\ell_1, \ldots, \ell_r}(a_{\ell_1}, \ldots, a_{\ell_r}), \tag{79}$$

since

$$f_r(t_1 a_1, \ldots, t_n a_n) = \sum_{1 \leq \ell_1 \leq \ldots \leq \ell_r \leq n} f_{\ell_1, \ldots, \ell_r}(t_{\ell_1} a_{\ell_1}, \ldots, t_{\ell_r} a_{\ell_r}) \tag{80}$$

$$= \sum_{1 \leq \ell_1 \leq \ldots \leq \ell_r \leq n} t_{\ell_1} \cdots t_{\ell_r} f_{\ell_1, \ldots, \ell_r}(a_{\ell_1}, \ldots, a_{\ell_r}). \tag{81}$$

Now, let $g \in O(d)$. Then, by the equivariance of $f_r$, we have

$$\sum_{1 \leq \ell_1 \leq \ldots \leq \ell_r \leq n} t_{\ell_1} \cdots t_{\ell_r} f_{\ell_1, \ldots, \ell_r}(g \cdot a_{\ell_1}, \ldots, g \cdot a_{\ell_r}) = \sum_{1 \leq \ell_1 \leq \ldots \leq \ell_r \leq n} t_{\ell_1} \cdots t_{\ell_r} g \cdot f_{\ell_1, \ldots, \ell_r}(a_{\ell_1}, \ldots, a_{\ell_r}),$$
$$\tag{82}$$

since

$$\sum_{1 \leq \ell_1 \leq \ldots \leq \ell_r \leq n} t_{\ell_1} \cdots t_{\ell_r} f_{\ell_1, \ldots, \ell_r}(g \cdot a_{\ell_1}, \ldots, g \cdot a_{\ell_r}) \tag{83}$$

$$= f_r(t_1 (g \cdot a_1), \ldots, t_n (g \cdot a_n)) \tag{84}$$
$$= f_r(g \cdot t_1 a_1, \ldots, g \cdot t_n a_n) \tag{85}$$
$$= g \cdot f_r(t_1 a_1, \ldots, t_n a_n) \tag{86}$$

$$= g \cdot \left( \sum_{1 \leq \ell_1 \leq \ldots \leq \ell_r \leq n} t_{\ell_1} \cdots t_{\ell_r} f_{\ell_1, \ldots, \ell_r}(a_{\ell_1}, \ldots, a_{\ell_r}) \right) \tag{87}$$

$$= \sum_{1 \leq \ell_1 \leq \ldots \leq \ell_r \leq n} t_{\ell_1} \cdots t_{\ell_r} g \cdot f_{\ell_1, \ldots, \ell_r}(a_{\ell_1}, \ldots, a_{\ell_r}). \tag{88}$$

Hence, for all $g \in O(d)$, $t \in \mathbb{R}$ and $(a_1, \ldots, a_n) \in \prod_{i=1}^n \mathcal{T}_{k_i}(\mathbb{R}^d, p_i)$, we have that

$$0 = \sum_{1 \leq \ell_1 \leq \ldots \leq \ell_r \leq n} t_{\ell_1} \cdots t_{\ell_r} \left[ g \cdot f_{\ell_1, \ldots, \ell_r}(a_{\ell_1}, \ldots, a_{\ell_r}) - f_{\ell_1, \ldots, \ell_r}(g \cdot a_{\ell_1}, \ldots, g \cdot a_{\ell_r}) \right]_{i_1, \ldots, i_{k'}}.$$
$$\tag{89}$$

Now, each of these is a polynomial in $t_1, \ldots, t_n$ that vanishes on $\mathbb{R}^n$. Moreover, note that no two $t_{\ell_1} \cdots t_{\ell_r}$ give the same monomial. Hence, by (Cox et al., 2015, Chapter 1 §1 Proposition 5), all these polynomials are the zero polynomial, i.e., their coefficients are zero. In this way, we conclude that for each $f_{\ell_1, \ldots, \ell_r}$, and all $g \in O(d)$, $t \in \mathbb{R}$ and $(a_1, \ldots, a_n) \in \prod_{i=1}^n \mathcal{T}_{k_i}(\mathbb{R}^d, p_i)$,

$$f_{\ell_1, \ldots, \ell_r}(g \cdot a_{\ell_1}, \ldots, g \cdot a_{\ell_r}) = g \cdot f_{\ell_1, \ldots, \ell_r}(a_{\ell_1}, \ldots, a_{\ell_r}), \tag{90}$$

i.e., each $f_{\ell_1, \ldots, \ell_r}$ is $O(d)$-equivariant.

Now, we show how to obtain the decomposition. Recall that $f_r$ is homogeneous of degree $r$. Therefore each entry of $f_r(a_1, \ldots, a_n)$ is an homogeneous polynomial of degree $r$ in the $[a_i]_{j_1, \ldots, j_{k_i}}$, i.e., a linear combination of products of the form

$$\prod_{q=1}^r [a_{\ell_q}]_{j_{q,1}, \ldots, j_{q, k_{\ell_q}}},$$

where, without loss of generality, we can assume that $\ell_1 \leq \cdots \leq \ell_q$. Hence, in coordinates, we have

$$[f_r(a_1, \ldots, a_n)]_{i_1, \ldots, i_{k'}}$$

$$= \sum_{1 \leq \ell_1 \leq \cdots \leq \ell_r \leq n} \lambda_{\ell_1, \ldots, \ell_r; i_1, \ldots, i_{k'}; j_{1,1}, \ldots, j_{1,k_{\ell_1}}, \ldots, j_{r,1}, \ldots, j_{r,k_{\ell_r}}} \prod_{q=1}^r [a_{\ell_q}]_{j_{q,1}, \ldots, j_{q,k_{\ell_q}}} \tag{91}$$

And so, we can consider the map $f_{\ell_1,\dots,\ell_r}$ given in coordinates by

$$[f_{\ell_1,\dots,\ell_r}(a_{\ell_1},\dots,a_{\ell_r})]_{i_1,\dots,i_{k'}} := \lambda_{\ell_1,\dots,\ell_r;i_1,\dots,i_{k'};j_{1,1},\dots,j_{1,k_{\ell_1}},\dots,j_{r,1},\dots,j_{r,k_{\ell_r}}} \prod_{q=1}^{r}[a_{\ell_q}]_{j_{q,1},\dots,j_{q,k_{\ell_q}}},$$

(92)

which, by construction, is the composition of the linear map given by

$$b \mapsto \lambda_{\ell_1,\dots,\ell_r;i_1,\dots,i_{k'};j_1,\dots,j_{\sum_{q=1}^r k_{\ell_q}}}[b]_{j_1,\dots,j_{\sum_{q=1}^r k_{\ell_q}}},$$

in coordinates, and $(a_{\ell_1},\dots,a_{\ell_r}) \mapsto a_{\ell_1} \otimes \cdots \otimes a_{\ell_r}$. Hence the desired decomposition of $f_r$ has been obtained. $\qquad\square$

## C  Proof of Corollary 1

In this section, we will prove Corollary 1 using the following lemma, which we prove afterward. This lemma, originally from Pastori, follows from Jeffreys (1973).

**Lemma 3** (Characterization of $O(d)$-isotropic $k_{(p)}$-tensors)**.** Suppose $c \in \mathcal{T}_k(\mathbb{R}^d, p)$ is $O(d)$-isotropic. Then the following holds:

*Case $p = +1$:* Assume $p = +1$. If $k$ is even, then $c$ can be written in the form

$$c = \sum_{\sigma \in S_k} \alpha_\sigma \left(\delta^{\otimes \frac{k}{2}}\right)^\sigma, \quad \text{for some } \alpha_\sigma \in \mathbb{R}.$$

(93)

Otherwise, if $k$ is odd, then $c = 0$ is the zero tensor.

*Case $p = -1$:* Assume $p = -1$. If $k - d$ is even and $k \geq d$, then $c$ can be written in the form

$$c = \sum_{\sigma \in S_k} \beta_\sigma \left(\delta^{\otimes \frac{k-d}{2}} \otimes \epsilon\right)^\sigma$$

(94)

for some $\beta_\sigma \in \mathbb{R}$. Otherwise, if $k - d$ is odd or $k < d$, then $c = 0$ is the zero tensor.

Note that in both cases only a subset of the permutations $\sigma \in S_k$ will yield different isotropic tensors (see Appendix D for a discussion).

*Proof of Corollary 1.* By Theorem 1 and Lemma 3, we can assume, without loss of generality, it suffices to consider the special case where $f$ consists of a single term

$$f(v_1,\dots,v_n) = \iota_r\left(v_{\ell_1} \otimes \cdots \otimes v_{\ell_r} \otimes \left(\delta^{\otimes \frac{r+k'}{2}}\right)^\sigma\right),$$

(95)

for some $\sigma \in S_{r+k'}$ and $r + k'$ even. To simplify notation, set $t := \frac{r+k'}{2}$.

Now, note that

$$\delta = e_i \otimes e_i,$$

where $\{e_1,\dots,e_d\}$ is the canonical basis of $\mathbb{R}^d$. Hence we get

$$f(v_1,\dots,v_n) = \iota_r(v_{\ell_1} \otimes \cdots \otimes v_{\ell_r} \otimes (e_{i_1} \otimes e_{i_1} \otimes \cdots \otimes e_{i_t} \otimes e_{i_t})^\sigma)$$

(96)

Let's write

$$(e_{i_1} \otimes e_{i_1} \otimes \cdots \otimes e_{i_t} \otimes e_{i_t})^\sigma = e_{j_1} \otimes \cdots \otimes e_{j_{2t}}$$

where $(j_1,\dots,j_{2t})$ is some permutation of $(i_1,i_1,\dots,i_t,i_t)$ and so, by Einstein notation, we are still adding over repeated indexes. Then we have that

$$f(v_1,\dots,v_n) = \langle v_{\ell_1}, e_{j_1}\rangle \cdots \langle v_{\ell_r}, e_{j_r}\rangle e_{j_{r+1}} \otimes \cdots \otimes e_{j_{2t}}.$$

(97)

Now, we can freely rearrange the $\langle v_\ell, e_j\rangle$ as they are scalars. There are three cases for each of the original indices $i_q$: (a) $e_{i_q}$ appears in an inner product twice, (b) $e_{i_q}$ appears in an inner product once, or (c) $e_{i_q}$ does not appear in an inner product.

In the case (a), we will get

$$\langle v_\ell, e_i\rangle\langle v_{\ell'}, e_i\rangle = \langle v_\ell, v_{\ell'}\rangle.$$

In the case (b), we will get

$$\langle v_\ell, e_i \rangle e_i = v_\ell.$$

And, in the case (c), we will get

$$e_i \otimes e_i = \delta.$$

Now, assume that we have $\alpha$ of the case (a), $\beta$ of the case (b) and $\gamma$ of the case (c). By permuting the $i_q$, which does not change the result, we can write for some permutation $\tilde\sigma \in S_{\beta+\gamma}$ and some permutation $J_1, \ldots, J_r$ some permutation of $\ell_1, \ldots, \ell_r$ that

$$\begin{aligned}
f(v_1, \ldots, v_n) &= \langle v_{J_1}, e_{i_1} \rangle \langle v_{J_2}, e_{i_1} \rangle \cdots \langle v_{J_{2\alpha-1}}, e_{i_\alpha} \rangle \langle v_{J_{2\alpha}}, e_{i_\alpha} \rangle \\
&\quad \left( \langle v_{J_{2\alpha+1}}, e_{i_{\alpha+1}} \rangle e_{i_{\alpha+1}} \otimes \cdots \otimes \langle v_{J_{2\alpha+\beta}}, e_{i_{\alpha+\beta}} \rangle e_{i_{\alpha+\beta}} \right. \\
&\quad \left. \otimes e_{i_{\alpha+\beta+1}} \otimes e_{i_{\alpha+\beta+1}} \otimes \cdots \otimes e_{i_{\alpha+\beta+\gamma}} \otimes e_{i_{\alpha+\beta+\gamma}} \right)^{\tilde\sigma} \\
&= \langle v_{J_1}, v_{J_2} \rangle \cdots \langle v_{J_{2\alpha-1}}, v_{J_{2\alpha}} \rangle \left( v_{J_{2\alpha+1}} \otimes \cdots \otimes v_{2\alpha+\beta} \otimes \delta^{\otimes\gamma} \right)^{\tilde\sigma}.
\end{aligned}$$

Hence, the desired claim follows, and we finish the proof. $\qquad\square$

*Proof of Lemma 3.* We will prove each case separately. However, note that no matter the value of $p$, an $O(d)$-isotropic tensor is always an $SO(d)$-isotropic tensor since $\det(M(g)) = 1$ for all $g \in SO(d)$. Now, by (Jeffreys, 1973, Theorem §2) (cf. (Appleby et al., 1987, Eq. (4.10))), any $SO(d)$-isotropic tensor $z$ can be written as a linear combination of the form

$$z = \sum_{\sigma \in S_k} \alpha_\sigma \left( \delta^{\otimes \frac{k}{2}} \right)^\sigma + \beta_\sigma \left( \delta^{\otimes \frac{k-d}{2}} \otimes \epsilon \right)^\sigma, \tag{98}$$

where $\delta$ is the Kronecker delta (Definition 8), and $\epsilon$ is the Levi-Civita symbol (Definition 9), with the convention that the coefficients $\alpha_\sigma$ and $\beta_\sigma$ are zero when the expressions $\delta^{\otimes \frac{k}{2}}$ and $\delta^{\otimes \frac{k-d}{2}}$ do not make sense. More precisely, the $\alpha_\sigma = 0$ if $k$ is odd, and the $\beta_\sigma = 0$ if $k - d$ is odd.

Note that under the $SO(d)$-action, we don't need to worry about the parity, and so both $\delta$ and $\epsilon$ are $SO(d)$-invariant. However, for the $O(d)$-action, the parity matters. Suppose $\gamma \in O(d)$ is a hyperplane reflection, and let $T$ be an $O(d)$-isotropic $k_{(-)}$-tensor. If $\hat{T}$ is a $k_{(+)}$-tensor whose components equal $T$, then

$$\gamma \cdot \hat{T} = -\hat{T}. \tag{99}$$

Likewise, if $T$ is an $O(d)$-isotropic $k_{(+)}$-tensor and $\hat{T}$ is a $k_{(-)}$-tensor whose components equal $T$, then

$$\gamma \cdot \hat{T} = -\hat{T}. \tag{100}$$

Note that being isotropic depends on the parity because it affects the considered action.

*Case $p = +1$:* Let $z \in \mathcal{T}_k(\mathbb{R}^d, +)$ be $O(d)$-isotropic. In particular, $z$ is also $SO(d)$-isotropic, and so we can write it using (98).

Recall that $O(d)$ is generated by all the (hyperplane) reflections. Hence, to show that $z$ is an $O(d)$-isotropic, we need only to show that for every (hyperplane) reflection $\gamma \in O(d)$,

$$\gamma \cdot z = z.$$

Now by (99),

$$\gamma \cdot \delta^{\otimes \frac{k-d}{2}} \otimes \epsilon = -\delta^{\otimes \frac{k-d}{2}} \otimes \epsilon, \tag{101}$$

since $\delta^{\otimes \frac{k-d}{2}} \otimes \epsilon$ is an $O(d)$-isotropic $k_{(-)}$-tensor. Hence

$$\gamma \cdot z = \gamma \cdot \sum_{\sigma \in S_k} \alpha_\sigma \left( \delta^{\otimes \frac{k}{2}} \right)^\sigma + \beta_\sigma \left( \delta^{\otimes \frac{k-d}{2}} \otimes \epsilon \right)^\sigma \tag{102}$$

$$= \sum_{\sigma \in S_k} \alpha_\sigma \left( (\gamma \cdot \delta)^{\otimes \frac{k}{2}} \right)^\sigma + \beta_\sigma \left( (\gamma \cdot \delta)^{\otimes \frac{k-d}{2}} \otimes \gamma \cdot \epsilon \right)^\sigma \tag{103}$$

$$= \sum_{\sigma \in S_k} \alpha_\sigma \left( \delta^{\otimes \frac{k}{2}} \right)^\sigma - \beta_\sigma \left( \delta^{\otimes \frac{k-d}{2}} \otimes \epsilon \right)^\sigma \tag{104}$$

$$= z - 2 \sum_{\sigma \in S_k} \beta_\sigma \left( \delta^{\otimes \frac{k-d}{2}} \otimes \epsilon \right)^\sigma. \tag{105}$$

By assumption, $z$ is $O(d)$-isotropic, so we conclude that $\sum_{\sigma \in S_k} \beta_\sigma \left( \delta^{\otimes \frac{k-d}{2}} \otimes \epsilon \right)^\sigma = 0$ and $z$ has the desired form.

*Case $p = -1$:* We argue as above, but using that for (hyperplane) reflection $\gamma \in O(d)$, we have, inside $\mathcal{T}_k(\mathbb{R}^d, -)$,

$$\gamma \cdot \delta^{\otimes \frac{k}{2}} = -\delta^{\otimes \frac{k}{2}}, \tag{106}$$

since $\delta^{\otimes \frac{k}{2}}$ is an $O(d)$-isotropic $k_{(+)}$-tensor. Hence, using a similar argument to the previous case, we conclude that $\sum_{\sigma \in S_k} \alpha_\sigma \left( \delta^{\otimes \frac{k}{2}} \right)^\sigma = 0$, so $z$ has the desired form. $\quad\square$

## D  SMALLER PARAMETERIZATION OF $O(d)$-ISOTROPIC TENSORS

In Lemma 3, the sum does not have to be over all permutations. The reason for this is that the tensors

$$\delta^{\otimes \frac{k}{2}} \quad \text{and} \quad \delta^{\otimes \frac{k-d}{2}} \otimes \epsilon$$

do not have a trivial stabilizer under the action of $S_k$. One can easily see the following proposition. Recall that the *stabilizer of a $k$-tensor $\pm T$ in $S_k$* is the following subgroup:

$$\mathrm{Stab}_{S_k}(\pm T) := \{ \sigma \in S_k \mid T^\sigma = \pm T \}, \tag{107}$$

where $T^\sigma = \pm T$ means that either $T^\sigma = T$ or $T^\sigma = -T$. Note that the laxity in the signs comes from the fact that positive summands and their negative counterparts can be combined.

**Proposition 4.** Consider the cases:

1. If $k$ is even, $\mathrm{Stab}_{S_k}\left( \pm \delta^{\otimes \frac{k}{2}} \right)$ is generated by the transpositions $(1, 2), (3, 4), \ldots, (k-1, k)$ and all permutations of the form $(i, j)(i+1, j+1)$ with $i, j < k$ odd. In particular, $\# \mathrm{Stab}_{S_k}\left( \pm \delta^{\otimes \frac{k}{2}} \right) = (k/2)! \, 2^{k/2}$.

2. If $k - d$ is even, $\mathrm{Stab}_{S_k}\left( \pm \delta^{\otimes \frac{k-d}{2}} \otimes \epsilon \right)$ is generated by the transpositions $(1, 2), (3, 4), \ldots, (k-d-1, k-d)$, all permutations of the form $(i, j)(i+1, j+1)$ with $i, j < k-d$ odd, and all transpositions of the form $(i, j)$ with $k - d < i, j$. In particular, $\# \mathrm{Stab}_{S_k}\left( \pm \delta^{\otimes \frac{k-d}{2}} \otimes \epsilon \right) = ((k-d)/2)! \, 2^{(k-d)/2} d!$.

*Proof.* This follows from (Roe Goodman, 2009, Theorem 5.3.4). $\quad\square$

Using this proposition, we can write any $O(d)$-isotropic $k_{(+)}$-tensor as

$$\sum_{\sigma \in G_k} \alpha_\sigma \left( \delta^{\otimes \frac{k}{2}} \right)^\sigma \tag{108}$$

with the $\alpha_\sigma$ real and

$$G_k = \left\{ \sigma \in S_k : \sigma(1) < \sigma(3) < \cdots < \sigma(k-1) \text{ and for all } i \leq \frac{k}{2}, \, \sigma(2i-1) < \sigma(2i) \right\} \tag{109}$$

of size $\frac{k!}{(k/2)! 2^{k/2}}$; and any $O(d)$-isotropic $k_{(-)}$-tensor as

$$\sum_{\sigma \in H_k} \beta_\sigma \left( \delta^{\otimes \frac{k-d}{2}} \otimes \epsilon \right)^\sigma \tag{110}$$

with the $\beta_\sigma$ real and

$$H_k = \Big\{ \sigma \in S_k : \sigma(1) < \sigma(3) < \cdots < \sigma(k-d-1), \text{ for all } i \leq \frac{k-d}{2}, \, \sigma(2i-1) < \sigma(2i)$$
$$\text{and for all } j > k - d, \sigma(j) < \sigma(j+1) \Big\}.$$

of size $\dfrac{k!}{\left( \frac{k-d}{2} \right)! 2^{\frac{k-d}{2}} d!}$.

# E  EXAMPLE OF THEOREM 1

In this section, we give a second example of Theorem 1.

**Example 2.** Let $f : \mathcal{T}_1(\mathbb{R}^d, +) \times \mathcal{T}_2(\mathbb{R}^d, +) \to \mathcal{T}_2(\mathbb{R}^d, +)$ be $O(d)$-equivariant polynomial of degree at most 2. By Theorem 1 we can write $f$ in the form

$$f(a_1, a_2) = \sum_{r=0}^{2} \sum_{1 \le \ell_1 \le \cdots \le \ell_r \le 2} \iota_{k_{\ell_1, \dots, \ell_r}} (a_{\ell_1} \otimes \dots \otimes a_{\ell_r} \otimes c_{\ell_1, \dots, \ell_r}) , \qquad (111)$$

where $c_{\ell_1, \dots, \ell_r}$ is an $O(d)$-isotropic $(k_{\ell_1, \dots, \ell_r} + 2)_{(+)}$-tensor. By Lemma 3, $c_{\ell_1, \dots, \ell_r}$ is nontrivial only when $k_{\ell_1, \dots, \ell_r} + 2$ is even. Recall that $k_{\ell_1, \dots, \ell_r} = \sum_{q=1}^{r} k_{\ell_q}$. The inputs are a $1_{(+)}$-tensor and $2_{(+)}$-tensor. The even combinations of 1 and 2 with at most 2 terms are $\emptyset, 2, 1+1, 2+2$ so we have

$$f(a_1, a_2) = \beta_0 \delta + \iota_2(a_2 \otimes c_2) + \iota_2(a_1 \otimes a_1 \otimes c_2') + \iota_4(a_2 \otimes a_2 \otimes c_3) , \qquad (112)$$

where $c_2, c_2'$ are $O(d)$-isotropic $4_{(+)}$-tensors and $c_3$ is an $O(d)$-isotropic $6_{(+)}$-tensor. By similar reasoning to Example 1, we can write

$$\iota_2(a_2 \otimes c_2) = \beta_1 \operatorname{tr}(a_2)\delta + \beta_2 a_2 + \beta_3 a_2^\top \qquad (113)$$

for constants $\beta_1, \beta_2, \beta_3$ and

$$\iota_2(a_1 \otimes a_1 \otimes c_2') = \beta_4 \langle a_1, a_1 \rangle \delta + \beta_5 a_1 \otimes a_1 , \qquad (114)$$

for constants $\beta_4, \beta_5$ (there are only two terms due to the symmetry of $a_1 \otimes a_1$). It remains to consider $\iota_4(a_2 \otimes a_2 \otimes c_3)$. By Lemma 3, we can write

$$c_3 = \sum_{\sigma \in G_6} \beta_\sigma (\delta^{\otimes 3})^\sigma , \qquad (115)$$

where $|G_6| = 6!/(3! 2^3) = 15$. In particular, we have

$$\begin{aligned} G_6 = \big\{ &(1,2,3,4,5,6), (1,2,3,5,4,6), (1,2,3,5,6,4), (1,3,2,4,5,6), (1,3,2,5,4,6), \\ &(1,3,2,5,6,4), (1,3,4,2,5,6), (1,3,4,5,2,6), (1,3,4,5,6,2), (1,3,5,2,4,6), \\ &(1,3,5,2,6,4), (1,3,5,4,2,6), (1,3,5,4,6,2), (1,3,5,6,2,4), (1,3,5,6,4,2) \big\} . \end{aligned} \qquad (116)$$

However, due to the symmetry of $a_2 \otimes a_2$, when we compute $\iota_4\big(a_2 \otimes a_2 (\delta^{\otimes 3})^\sigma\big)$ for $\sigma \in G_6$, there are only 7 distinct terms

$$\begin{aligned} \iota_4(a_2 \otimes a_2 \otimes c_3) = \\ \beta_6 \operatorname{tr}(a_2)^2 \delta + \beta_7 \operatorname{tr}(a_2)a_2 + \beta_8 \operatorname{tr}(a_2)a_2^\top + \beta_9 a_2^\top a_2 + \beta_{10} a_2 a_2^\top + \beta_{11} a_2 a_2 + \beta_{12} a_2^\top a_2^\top . \end{aligned} \qquad (117)$$

In summary,

$$\begin{aligned} f(a_1, a_2) = \beta_0 \delta + \beta_1 \operatorname{tr}(a_2)\delta + \beta_2 a_2 + \beta_3 a_2^\top + \beta_3 \langle a_1, a_1 \rangle \delta + \beta_4 a_1 \otimes a_1 + \beta_5 \operatorname{tr}(a_2)^2 \delta \\ + \beta_6 \operatorname{tr}(a_2)a_2 + \beta_7 \operatorname{tr}(a_2)a_2^\top + \beta_8 a_2^\top a_2 + \beta_9 a_2 a_2^\top + \beta_{10} a_2 a_2 + \beta_{11} a_2^\top a_2^\top , \end{aligned} \qquad (118)$$

for some coefficients $\beta_0, \beta_1, \dots, \beta_{11}$.

# F  PROOF OF COROLLARY 2

Before proving Corollary 2, we will state and prove an important lemma. Since this is the setting of symmetric matrices, we will use familiar notation and concepts from that setting rather than relying on tensor algebra.

**Lemma 4.** Let $f : \mathbb{R}_{sym}^{d \times d} \to \mathbb{R}_{sym}^{d \times d}$ be an $O(d)$-equivariant function. Let $A \in \mathbb{R}_{sym}^{d \times d}$, then $f(A) A = A f(A)$ so they are simultaneously diagonalizable.

*Proof.* Let $f : \mathbb{R}_{sym}^{d \times d} \to \mathbb{R}_{sym}^{d \times d}$ be an $O(d)$-equivariant function. First we will show that for all diagonal matrices $D, f(D)$ is also diagonal. Let $D \in \mathbb{R}_{sym}^{d \times d}$ be a diagonal matrix. Let $Q_i$ for $i = 1, \ldots, d$ be the orthogonal matrices that are equal to the identity matrix except $[Q_i]_{ii} = -1$. Since $Q_i \, D \, Q_i^\top = D$, we have $f(D) = f(Q_i \, D \, Q_i^\top) = Q_i \, f(D) \, Q_i^\top$. Now we note that the $i^{th}$ row and column of $Q_i \, f(D) \, Q_i^\top$ is equal to the negative $i^{th}$ row and column of $f(D)$, except for the diagonal elements which are equal: $\left[ Q_i \, f(D) \, Q_i^\top \right]_{ii} = [f(D)]_{ii}$. However, $f(D) = Q_i \, f(D) \, Q_i^\top$. Thus we conclude that the off diagonal elements of $f(D)$ must be equal to 0 since this is true for $i = 1, \ldots, d$. Hence, $f(D)$ is diagonal when $D$ is diagonal.

Now let $A \in \mathbb{R}_{sym}^{d \times d}$, so $A$ is diagonalizable, $A = Q \, \Lambda \, Q^T$. Thus,

$$f(A) \, A = f(Q \, \Lambda \, Q^\top) Q \, \Lambda \, Q^\top = Q \, f(\Lambda) \, \Lambda \, Q^\top = Q \, \Lambda \, f(\Lambda) \, Q^\top = Q \, \Lambda \, Q^\top \, f(Q \, \Lambda \, Q^\top) = A \, f(A) \,. \tag{119}$$

This concludes the proof. $\qquad \square$

Now we will prove the corollary. Note that since $\widetilde{f}$ is a function of diagonal matrices, we could equivalently think of it as the composition of converting the diagonal matrix to a vector, performing the function on the vector, then converting back to a diagonal matrix. We will follow this implementation in the code to avoid storing a large diagonal matrix.

**Corollary.** Let $f : \mathbb{R}_{sym}^{d \times d} \to \mathbb{R}_{sym}^{d \times d}$ be an $O(d)$-equivariant function. Then there exists a function $\widetilde{f} : \mathbb{R}_{\text{diag}}^{d \times d} \to \mathbb{R}_{\text{diag}}^{d \times d}$ of diagonal matrices that is permutation equivariant such that for all $A \in \mathcal{T}_2^{sym}(\mathbb{R}^d, +)$, $f(A) = Q \, \widetilde{f}(\Lambda) \, Q^\top$, where $A = Q \, \Lambda \, Q^\top$ is the eigenvalue decomposition.

*Proof.* Let $f : \mathbb{R}_{sym}^{d \times d} \to \mathbb{R}_{sym}^{d \times d}$ be an $O(d)$-equivariant function. Let $A \in \mathbb{R}_{sym}^{d \times d}$ with eigenvalue decomposition $A = Q \, \Lambda \, Q^\top$. Then by Lemma 4, $Q$ also diagonalizes $f(A)$, that is $Q^\top \, f(A) \, Q$ is diagonal. Hence $Q^\top \, f(A) \, Q = f(Q^\top \, A \, Q) = f(\Lambda)$ is also diagonal. Since the input and output are both diagonal, we can define the function $\widetilde{f}$ that is the same as $f$ but restricted to diagonal matrices, and is therefore $\widetilde{f} : \mathbb{R}_{\text{diag}}^{d \times d} \to \mathbb{R}_{\text{diag}}^{d \times d}$. Thus $f(\Lambda) = \widetilde{f}(\Lambda)$. Therefore $Q^\top \, f(A) \, Q = \widetilde{f}(\Lambda)$ implies that $f(A) = Q \, \widetilde{f}(\Lambda) \, Q^\top$.

Finally, we must show that $\widetilde{f}$ is permutation equivariant. Let $\Lambda$ be any diagonal matrix. Since the permutations of $d$ elements is a subgroup of $O(d)$, we have for any permutation matrix $P$:

$$f(P \, \Lambda \, P^\top) = P \, f(\Lambda) \, P^\top \tag{120}$$

$$\widetilde{f}(P \, \Lambda \, P^\top) = P \, \widetilde{f}(\Lambda) \, P^\top \,. \tag{121}$$

Thus $\widetilde{f}$ is permutation equivariant for any input, which completes the proof. $\qquad \square$

## G  GENERALIZATION TO OTHER LINEAR ALGEBRAIC GROUPS

In this section, we will show how Theorem 1 and Corollary 1 can be extended to the indefinite orthogonal and the symplectic group as Theorem 2 and Corollary 3.

The main idea to extend Theorem 1 to other groups is to use some form of averaging. On $O(d)$, the compactness guarantees the existence of a Haar probability measure. However, to apply the same trick over non-compact groups such as $O(s, d - s)$ and $Sp(d)$, we need to use technical machinery to imitate the averaging strategy.

First, we introduce some definitions and examples regarding complex and real linear algebraic groups. The main point will be to establish how to get a compact subgroup over which to average. Basically the results will generalize to real linear algebraic groups such that their complexifications have a Zariski-dense compact subgroup. For instance reductive connected complex algebraic groups satisfy this assumption. Second, we prove a generalization of Theorem 1 for complex linear algebraic groups with a Zariski-dense compact subgroup acting on complex tensors. Third, we prove a generalization of Theorem 1 for real linear algebraic groups that are compact or such that their complexification has a Zariski-dense compact subgroup. Finally, we prove Corollary 3.

## G.1 REDUCTIVE COMPLEX AND REAL LINEAR ALGEBRAIC GROUPS

Recall that a *complex linear algebraic group* is a subgroup $G$ of $\mathrm{GL}(V)$, where $V$ is a finite-dimensional complex vector space, such that $G$ is the zero set of some set of complex polynomial functions over $End(V)$, the set of (complex) linear maps $V \to V$. Recall also that a *rational G-module* of $G$ is a vector space $U$ together with a linear action of $G$ on $U$ such that the map $G \times U \ni (g, x) \mapsto g \cdot x \in U$ is polynomial[2], and that a $G$-submodule $U_0$ of $U$ is a vector subspace $U_0 \subseteq U$ such that for all $g \in G$, $g \cdot U_0 \subseteq U_0$.

**Definition 10.** (Roe Goodman, 2009, Def. 3.3.1) A *reductive complex linear algebraic group* is a complex linear algebraic group $G \subset GL(V)$ such that every rational $G$-module $U$ is *completely reducible*, i.e., for every $G$-submodule $U_0$ of $U$, there is a $G$-submodule $U_1$ such that $U = U_0 + U_1$ and $U_0 \cap U_1 = 0$.

**Example 3.** Given any finite-dimensional vector space, the classical complex groups $GL(V)$ and $SL(V)$ are reductive complex linear algebraic groups.

**Example 4.** Given any finite-dimensional vector space $V$ together with a symmetric non-degenerate bilinear form[3] $\langle \cdot, \cdot \rangle : V \times V \to \mathbb{C}$, the (complex) orthogonal group

$$O(V, \langle \cdot, \cdot \rangle) := \{g \in GL(V) \mid \text{for all } v, w \in V, \langle g \cdot v, g \cdot w \rangle = \langle v, w \rangle\} \tag{122}$$

is a reductive complex linear algebraic group. We will pay special attention to the following family of complex orthogonal groups:

$$O^{\mathbb{C}}(s, d - s) := \{g \in GL(\mathbb{C}^d) \mid g^\top \mathbb{I}_{s,d-s} g = \mathbb{I}_{s,d-s}\} = O(\mathbb{C}^d, \langle \cdot, \cdot \rangle_s) \tag{123}$$

where $\langle u, v \rangle_s := u^\top \mathbb{I}_{s,d-s} v$. Note that all these groups are isomorphic, satisfying that

$$O^{\mathbb{C}}(s, d - s) = \begin{pmatrix} \mathbb{I}_s & \\ & i\mathbb{I}_{d-s} \end{pmatrix} O^{\mathbb{C}}(d, 0) \begin{pmatrix} \mathbb{I}_s & \\ & i\mathbb{I}_{d-s} \end{pmatrix}^{-1}.$$

Moreover, this is true in general: any two complex orthogonal groups are isomorphic if they are of the same order—this follows from the fact that all symmetric non-degenerate bilinear forms are equivalent over the complex numbers.

**Example 5.** Given any finite-dimensional vector space $V$ together with an anti-symmetric non-degenerate bilinear form $\langle \cdot, \cdot \rangle : V \times V \to \mathbb{C}$, the (complex) symplectic group

$$Sym(V, \langle \cdot, \cdot \rangle) := \{g \in GL(V) \mid \text{for all } v, w \in V, \langle g \cdot v, g \cdot w \rangle = \langle v, w \rangle\} \tag{124}$$

is a reductive complex linear algebraic group. We will pay special attention to the following special case:

$$Sp^{\mathbb{C}}(d) := \{g \in GL(\mathbb{C}^d) \mid g^\top J_d g = J_d\} = Sp(\mathbb{C}^d, \langle \cdot, \cdot \rangle_{\text{symp}}) \tag{125}$$

where $\langle u, v \rangle_{\text{symp}} := u^\top J_d v$. Note that any symplectic group of order $d$ is isomorphic to $Sp^{\mathbb{C}}(d)$ because any two antisymmetric non-degenerate bilinear forms are equivalent over the complex numbers.

**Example 6.** The complex linear algebraic group

$$H = \left\{ \begin{pmatrix} 1 & t \\ & 1 \end{pmatrix} \mid t \in \mathbb{C} \right\}$$

is not reductive, since $\mathbb{C}^2$ is an $H$-module that is not completely reducible. Note that $\mathbb{C} \times 0$ is the only $H$-submodule of $\mathbb{C}^2$, so we cannot find a complementary $H$-submodule.

Recall that a subset $X$ of a set $\tilde{X}$ is *Zariski-dense* in $\tilde{X}$ if every polynomial function that vanishes in $X$ vanishes in $\tilde{X}$, i.e. if every polynomial function that does not vanish on $\tilde{X}$ does not vanish in $X$. The following theorem allows us to use the power of averaging for reductive connected complex linear algebraic groups.

---

[2]To be precise, we mean that the map $G \times U \to U$ is a morphism of algebraic varieties. Choose basis for $U$ and $V$, so that we can identify $V$ with $\mathbb{C}^d$ and $U$ with $\mathbb{C}^n$. Then, being a morphism between algebraic varieties, just means that the map $G \times \mathbb{C}^n \to \mathbb{C}^n$ is the restriction of a map $\mathbb{C}^{d \times d} \times \mathbb{C}^n \to \mathbb{C}^n$ that can be written componentwise as $(p_l((g_{i,j})_{i,j}, (u_k)_k)/(\det g)^{a_l})_l$ where each $p_l$ is a polynomial in the $g_{i,j}$ and $u_k$ and each $a_l$ an integer.

[3]Recall that this means that for all $u, v, w \in V$ and $t, s \in \mathbb{C}$: (a) $\langle u, v \rangle = \langle v, u \rangle$, (b) for all $x \in V$, $\langle u, x \rangle = 0$ if and only if $u = 0$, and (c) $\langle tu + sv, w \rangle = t\langle u, w \rangle + s\langle v, w \rangle$.

**Theorem 3.** (Roe Goodman, 2009, Theorem 11.5.1) Let $G$ be a reductive connected complex algebraic group. Then there exists a Zariski-dense compact subgroup $K$. More precisely, there is a subgroup $U(G)$ of $G$ that is Zariski-dense in $G$ and that, with respect to the usual topology[4], is compact.

**Remark 3.** Note that using this compact subgroup $K$, we can consider expressions of the form

$$\mathop{\mathbb{E}}_{\mathfrak{u} \in U(G)} \mathfrak{u} \cdot T$$

by taking the expectation with respect to the unique Haar probability measure of $K$. Now, since $U(G)$ is Zariski-dense in $G$, we have that the fact that for all $u \in U(G)$, $u \cdot \left(\mathbb{E}_{\mathfrak{u} \in U(G)} \mathfrak{u} \cdot T\right) = \mathbb{E}_{\mathfrak{u} \in U(G)} \mathfrak{u} \cdot T$ implies that for all $g \in G$,

$$g \cdot \left(\mathop{\mathbb{E}}_{\mathfrak{u} \in U(G)} \mathfrak{u} \cdot T\right) = \mathop{\mathbb{E}}_{\mathfrak{u} \in U(G)} \mathfrak{u} \cdot T.$$

Note that $U(G)$ is not necessarily unique.

**Example 7.** In $GL(\mathbb{C}^d)$, the Zariski-dense compact subgroup is the group of unitary matrices:

$$U(\mathbb{C}^d) := \{g \in GL(\mathbb{C}^d) \mid g^* g = \mathbb{I}_d\}$$

where $^*$ denotes the conjugate transpose. In $SL(\mathbb{C}^d)$, it is the group of special unitary transformations:

$$SU(\mathbb{C}^d) := \{g \in U(\mathbb{C}^d) \mid \det g = 1\}.$$

**Example 8.** In $O^{\mathbb{C}}(s, d - s)$, the Zariski-dense compact subgroup is

$$\begin{pmatrix} \mathbb{I}_s & \\ & i\mathbb{I}_{d-s} \end{pmatrix} O(d) \begin{pmatrix} \mathbb{I}_s & \\ & i\mathbb{I}_{d-s} \end{pmatrix}^{-1}.$$

Note that when $s = 0$ or $s = d$, this is the orthogonal group over the reals. Moreover, this does not follow from Theorem 3 as $O^{\mathbb{C}}(s, d - s)$ is not connected.

**Example 9.** In $Sp^{\mathbb{C}}(d)$, the Zariski-dense compact subgroup is the so-called compact symplectic group:

$$USp(d) := Sp^{\mathbb{C}}(d) \cap U(\mathbb{C}^d).$$

Recall that a *real linear algebraic group* is a subgroup $G$ of $GL(V)$, where $V$ is a finite-dimensional real vector space, such that $G$ is the zero set of some set of real polynomial functions over $\mathbb{R}^{d \times d}$. Similarly, as we did with complex linear algebraic groups, we can talk about *rational modules* and about *reductive real linear algebraic groups*.

However, given a reductive real linear algebraic group we cannot necessarily guarantee the existence of a Zariski-dense compact subgroup. This means that we cannot apply the averaging trick directly, but we can do so by passing to the Zariski-dense compact subgroup of the complexification of the real linear algebraic group.

**Definition 11.** Let $G \subset GL(V)$ be a real linear algebraic group. The *complexification* $G^{\mathbb{C}}$ of $G$ is the complex linear algebraic group given by

$$G^{\mathbb{C}} := \{g \in GL(V^{\mathbb{C}}) \mid \text{for every polynomial } f \text{ such that } f(G) = 0, \ f(g) = 0\} \tag{126}$$

where $V^{\mathbb{C}} := V \otimes_{\mathbb{R}} \mathbb{C}$ is the complexification of $V$, i.e., the complex vector space obtained from $V$ by extending scalars.

**Remark 4.** In essence, we complexify the underlying real algebraic variety. Group multiplication preserves its structure as a complex variety as it is given by polynomial functions of the matrix entries.

**Definition 12.** A real linear algebraic group $G$ is *complexly averageable* if it's Zariski-dense in its complexification and its complexification admits a Zariski-dense compact subgroup closed under complex conjugation.

**Remark 5.** Recall that the complexification of $\mathbb{R}^d$ is naturally isomorphic to $\mathbb{C}^d$.

**Example 10.** The complexification of $GL(\mathbb{R}^d)$ is $GL(\mathbb{C}^d)$, and the complexification of $SL(\mathbb{R}^d)$ is $SL(\mathbb{C}^d)$.

---

[4] The topology inherited from the Euclidean topology of $GL(\mathbb{C}^d)$.

**Example 11.** We have that
$$O(s, d-s)^{\mathbb{C}} = O^{\mathbb{C}}(s, d-s)$$
and that
$$Sp(d)^{\mathbb{C}} = Sp^{\mathbb{C}}(d).$$
Hence, both the indefinite orthogonal group and symplectic group are complexly averageable. The symplectic group is connected but the indefinite orthogonal group is not connected. However, it does have a Zariski-dense compact subgroup (see Example 8).

The following proposition shows that complexly averageable real linear algebraic groups are common.

**Proposition 5.** Let $G \subset GL(V)$ be a real linear algebraic group. Then: (1) $G$ is Zariski-dense in $G^{\mathbb{C}}$. (2) If the complexification $G^{\mathbb{C}}$ of $G$ is connected and reductive, then $G$ is complexly averageable.

*Proof.* (1) Let $f$ be a complex polynomial vanishing on $G$. Then, we can write this polynomial as $f = f_r + i f_i$ for some polynomials $f_r$ and $f_i$ with real coefficients. Now, since $f$ vanishes on $G$, then $f_r$ and $f_i$ vanish also on $G$—as otherwise there would be $g \in G$ such that either $f_r(g) \neq 0$ or $f_i(g) \neq 0$, contradicting $f(g) = 0$. But then, by definition of $G^{\mathbb{C}}$, $f_r$ and $f_i$ vanish on $G$ and so $f = f_r + i f_i$ vanishes on $G^{\mathbb{C}}$. Hence we have just proven that a complex polynomial vanishes on $G$ if and only if vanishes on $G^{\mathbb{C}}$, i.e., we have proven that $G$ is Zariski-dense in $G^{\mathbb{C}}$.

(2) This follows from Theorem 3. $\qquad\square$

**Example 12.** Observe that the Zariski-dense compact subgroups of $O^{\mathbb{C}}(s, d-s)$ and $Sp^{\mathbb{C}}(d)$ that have been given satisfy that they are closed under the complex conjugation.

### G.2 COMPLEX EQUIVARIANT TENSOR MAPS

We will consider vector spaces on which a non-degenerate bilinear form has been chosen.

**Definition 13.** A *self-paired vector space* $(V, \langle \cdot, \cdot \rangle)$ is a finite-dimensional vector space $V$ together with a non-degenerate bilinear form $\langle \cdot, \cdot \rangle : V \times V \to \mathbb{C}$.

Recall the *universal property* of tensor products of vector spaces, by which multilinear maps $V_1 \times \cdots \times V_k \to W$ can be lifted to linear maps $V_1 \otimes \cdots \otimes V_k \to W$. Using the universal property, we can see that from a self-paired vector space $(V, \langle \cdot, \cdot \rangle)$, we get the family
$$(V^{\otimes k}, \langle \cdot, \cdot \rangle)$$
of self-paired spaces of tensors, by extending by linearity the expression
$$\langle v_1 \otimes \cdots \otimes v_k, \tilde{v}_1 \otimes \cdots \otimes \tilde{v}_k \rangle = \langle v_1, \tilde{v}_1 \rangle \cdots \langle v_k, \tilde{v}_k \rangle. \tag{127}$$
And again, by the universal property, we get a *k-contraction*
$$\iota_k : V^{\otimes(2k+k')} \cong V^{\otimes k} \otimes V^{\otimes k} \otimes V^{\otimes k'} \to V^{\otimes k'} \tag{128}$$
by extending by linearity, the expression
$$a \otimes b \otimes c \mapsto \langle a, b \rangle c. \tag{129}$$

Now, in the above setting, let $G$ be a group acting in a structure-preserving way on $(V, \langle \cdot, \cdot \rangle)$, meaning that the action is linear and preserves $\langle \cdot, \cdot \rangle$, i.e., for all $g \in G, v, \tilde{v} \in V, \langle v, \tilde{v} \rangle = \langle g \cdot v, g \cdot \tilde{v} \rangle$. Then, by the universal property, we get that $G$ acts also on $(V^{\otimes k}, \langle \cdot, \cdot \rangle)$ by extending linearly the expression
$$g(v_1 \otimes \cdots \otimes v_k) = \chi(g)(gv_1) \otimes \cdots \otimes (gv_k). \tag{130}$$
Moreover, by considering all (rational)[5] unidimensional representations $\chi : G \to \mathbb{C}^*$ of $G$, we get the following family of self-paired (rational) $G$-modules:
$$\mathcal{T}_k(V, \chi) := (V^{\otimes k}, \langle \cdot, \cdot \rangle) \tag{131}$$
where the action by $G$ is given by
$$g \cdot T := \chi(g) M(g) \cdot T \tag{132}$$

---

[5]Recall that rational means that the homomorphism is given by polynomials.

in a structure-preserving way. For the sake of distinction, we will denote the $k$-contraction as

$$\iota_k^G : \mathcal{T}_{2k+k'}(V, \chi) \to \mathcal{T}_{k'}(V, \chi) \tag{133}$$

in this setting to emphasize the dependence on the group $G$, as we will be choosing the original $\langle \cdot , \cdot \rangle$ in terms of the group. Using the universal property, we can easily see the following:

**Proposition 6.** The following statements hold:

(a) The outer product map

$$\mathcal{T}_k(V, \chi) \times \mathcal{T}_{k'}(V, \chi') \to \mathcal{T}_{k+k'}(V, \chi\chi')$$

is a $G$-equivariant bilinear map.

(b) The $k$-contraction $\iota_k^G : \mathcal{T}_{2k+k'}(V, \chi) \to \mathcal{T}_{k'}(V, \chi)$ is a $G$-equivariant linear map.

(c) For any $\sigma \in S_k$, the tensor index permutation by $\sigma$, $\mathcal{T}_k(V, \chi) \to \mathcal{T}_k(V, \chi)$ given by $v_1 \otimes \cdots \otimes v_k \mapsto v_{\sigma^{-1}(1)} \otimes \cdots \otimes v_{\sigma^{-1}(k)}$, is a $G$-equivariant linear map.

Finally, recall that a $G$-isotropic tensor of $\mathcal{T}_\ell(V, \chi)$ is a $G$-invariant tensor in $\mathcal{T}_\ell(V, \chi)$. Further, recall that an *entire* function is an analytic function whose Taylor series at any point has an infinite radius of convergence. We can now state the theorem.

**Theorem 4.** Let $G \subset GL(V)$ be a reductive connected complex linear algebraic group (or more generally, a complex linear algebraic group with a Zariski-dense compact subgroup) acting rationally on an structure-preserving way on a self-paired complex vector space $(V, \langle \cdot , \cdot \rangle)$ and $f : \prod_{i=1}^n \mathcal{T}_{k_i}(V, \chi_i) \to \mathcal{T}_{k'}(V, \chi')$ a $G$-equivariant entire function. Then we may write $f$ as follows:

$$f(a_1, \ldots, a_n) = \sum_{r=0}^\infty \sum_{1 \leq \ell_1 \leq \cdots \leq \ell_r \leq n} \iota_{k_{\ell_1, \ldots, \ell_r}}^G \left( a_{\ell_1} \otimes \ldots \otimes a_{\ell_r} \otimes c_{\ell_1, \ldots, \ell_r} \right) \tag{134}$$

where $c_{\ell_1, \ldots, \ell_r} \in \mathcal{T}_{k_{\ell_1, \ldots, \ell_r} + k'}(\mathbb{R}^d, \chi_{\ell_1, \ldots, \ell_r} \chi')$ is a $G$-isotropic tensor for $k_{\ell_1, \ldots, \ell_r} := \sum_{q=1}^r k_{\ell_q}$ and $\chi_{\ell_1, \ldots, \ell_r} = \prod_{q=1}^r \chi_{\ell_q}$.

To prove this, we proceed as in the orthogonal case: we reduce to the multihomogeneous case and then prove the result using averaging over the Zariski-dense compact subgroup.

**Lemma 5.** Let $G \in GL(V)$ be any subgroup acting linearly on a self-paired complex vector space $(V, \langle \cdot , \cdot \rangle)$ and $f : \prod_{i=1}^n \mathcal{T}_{k_i}(V, \chi_i) \to \mathcal{T}_{k'}(V, \chi')$ an entire function. Then, we can write $f$ as

$$f_r(a_1, \ldots, a_n) = \sum_{r=0}^\infty \sum_{1 \leq \ell_1 \leq \ldots \leq \ell_r \leq n} f_{\ell_1, \ldots, \ell_r}(a_{\ell_1}, \ldots, a_{\ell_r}), \tag{135}$$

where $f_{\ell_1, \ldots, \ell_r} : \prod_{i=1}^r \mathcal{T}_{k_{\ell_i}}(V, \chi_i) \to \mathcal{T}_{k'}(V, \chi')$ is the composition of the map

$$\prod_{i=1}^r \mathcal{T}_{k_{\ell_i}}(V, \chi_{\ell_i}) \to \mathcal{T}_{\sum_{i=1}^r k_{\ell_i}} \left( V, \prod_{i=1}^r \chi_{\ell_i} \right) \tag{136}$$

$$(a_{\ell_1}, \ldots, a_{\ell_r}) \mapsto a_{\ell_1} \otimes \ldots \otimes a_{\ell_r} \tag{137}$$

with a linear map $\mathcal{T}_{\sum_{i=1}^r k_{\ell_i}}(V, \prod_{i=1}^r \chi_{\ell_i}) \to \mathcal{T}_{k'}(\mathbb{R}^d, \chi')$.

Moreover, for the above decomposition, if $f$ is $G$-equivariant, then so are the $f_{\ell_1, \ldots, \ell_r}$.

**Remark 6.** Note that we don't need to assume anything about $G$ in the above lemma.

*Proof of Theorem 4.* By Lemma 5, we can assume without loss of generality that $f$ is of the form

$$f(a_1, \ldots, a_n) = \lambda(a_{\ell_1} \otimes \cdots \otimes a_{\ell_r})$$

for some non-negative integer $r$, $1 \leq \ell_1 \leq \cdots \leq \ell_r \leq r$ and $\lambda : \mathcal{T}_{\sum_{i=1}^r k_{\ell_i}}(V, \prod_{i=1}^r \chi_{\ell_i}) \to \mathcal{T}_{k'}(V, \chi')$ is linear.

The above map can be written as a linear combination of maps of the form

$$(a_1, \ldots, a_n) \mapsto \left( \prod_{i=1}^{r} \lambda_i(a_{\ell_i}) \right) v_{j_1} \otimes \cdots \otimes v_{j_{k'}}$$

where the $\lambda_i$ are linear and $v_j \in V$, due to the universal property—the factor $(\prod_{i=1}^{r} \lambda_i(a_{\ell_i}))$ just corresponds to a linear map $\mathcal{T}_{\sum_{i=1}^{r} k_{\ell_i}}(V, \prod_{i=1}^{r} \chi_{\ell_i}) \to \mathbb{C}$. Moreover,

$$\left( \prod_{i=1}^{r} \lambda_i(a_{\ell_i}) \right) v_{j_1} \otimes \cdots \otimes v_{j_{k'}} = \iota_{\sum_{i=1}^{r} k_{\ell_i} + k'}^{G} \left( a_{\ell_1} \otimes \cdots \otimes a_{\ell_r} \otimes c_1 \otimes \cdots \otimes c_r \otimes v_{j_1} \otimes \cdots \otimes v_{j_{k'}} \right)$$

where the $c_i \in \mathcal{T}_{k_i}(V, \chi_i)$ are the unique tensors such that for all $a_{\ell_i} \in \mathcal{T}_{k_i}(V, \chi_i)$,

$$\lambda_i(a_{\ell_i}) = \langle a_{\ell_i}, c_i \rangle.$$

These $c_i$ exist, because $\langle \cdot, \cdot \rangle$ is non-degenerate. Hence for some $c \in \mathcal{T}_{\sum_{i=1}^{r} k_{\ell_i} + k'}(V, \prod_{i=1}^{r} \chi_i \chi')$, we have

$$f(a_1, \ldots, a_n) = \iota_{\sum_{i=1}^{r} k_{\ell_i} + k'}^{G}(a_{\ell_1} \otimes \cdots \otimes a_{\ell_r} \otimes c). \tag{138}$$

Since $f$ and $\iota_{k_\ell + k'}(\cdot)$ are $G$-equivariant, we have that for all $g \in G$ and $a \in \prod_{i=1}^{n} \mathcal{T}_{k_i}(V, \chi_i)$,

$$\iota_{k_\ell + k'}(a_{\ell_1} \otimes \cdots \otimes a_{\ell_r} \otimes c_\ell) = f(a_1, \ldots, a_n) \tag{139}$$

$$= f(g \cdot (g^{-1} \cdot a_1), \ldots, g \cdot (g^{-1} \cdot a_n)) \tag{140}$$

$$= g \cdot f((g^{-1} \cdot a_1), \ldots, (g^{-1} \cdot a_n)) \tag{141}$$

$$= g \cdot \iota_{k_\ell + k'} \left( (g^{-1} \cdot a_{\ell_1}) \otimes \cdots \otimes (g^{-1} \cdot a_{\ell_r}) \otimes c_\ell \right) \tag{142}$$

$$= \iota_{k_\ell + k'}(a_{\ell_1} \otimes \cdots \otimes a_{\ell_r} \otimes (g \cdot c_\ell)). \tag{143}$$

Finally, $G$ has a Zariski-dense compact subgroup $U(G)$. Hence, averaging over $U(G)$, we can substitute $c$ by the $U(G)$-isotropic tensor

$$\mathbb{E}_{\mathfrak{u} \in U(G)} \mathfrak{u} \cdot c$$

where the expectation is taken with respect the unique Haar probability measure of $U(G)$. But, since $U(G)$ is Zariski-dense in $G$ and the action rational, $\mathbb{E}_{\mathfrak{u} \in U(G)} \mathfrak{u} \cdot c$ is also $G$-isotropic, as we wanted to show. □

*Proof of Lemma 5.* Recall that, since $f$ in entire, we have, by Taylor's theorem, that

$$f(a) = \sum_{r=0}^{\infty} \frac{1}{r!} D_0^r f(a, \ldots, a) \tag{144}$$

where $a = (a_1, \ldots, a_n) \in \prod_{i=1}^{n} \mathcal{T}_{k_i}(V, \chi_i)$ and $D_0^k f : \left( \prod_{i=1}^{n} \mathcal{T}_{k_i}(V, \chi_i) \right)^k \to \mathcal{T}_{k'}(V, \chi')$ is the $k$-multilinear map given by $k$th order partial derivatives of $f$ at 0. Now, write $a = a_1 + \cdots + a_n$ as an abuse of notation for

$$a = (a_1, 0, \ldots, 0) + \cdots + (0, \ldots, 0, a_n).$$

We will further use this abuse of notation to write $a_i$ instead of $(0, \ldots, 0, a_i, 0, \ldots, 0)$. Now, since $D_0^k f$ is $k$-multilinear and symmetric, we have that

$$\frac{1}{r!} D_0^r f(a, \ldots, a) = \sum_{1 \leq \ell_1 \leq \cdots \leq \ell_r \leq n} \frac{1}{\alpha_{\ell_1, \ldots, \ell_r}!} D_0^r f(a_{\ell_1}, \ldots, a_{\ell_r}) \tag{145}$$

where $\alpha_{\ell_1, \ldots, \ell_r} \in \mathbb{N}^r$ is the vector given by $(\alpha_{\ell_1, \ldots, \ell_r})_i := \#\{j \mid \ell_j = i\}$ and $\alpha! := \alpha_1! \cdots \alpha_r!$. Note that this terms appears when we reorder $(a_{\ell_1}, \ldots, a_{\ell_r})$ so that the subindices are in order.

Summing up, we can write $f$ as (135), with

$$f_{\ell_1, \ldots, \ell_r}(a_1, \ldots, a_n) = \frac{1}{\alpha_{\ell_1, \ldots, \ell_r}!} D_0^r f(a_{\ell_1}, \ldots, a_{\ell_r}), \tag{146}$$

where this has the desired form by the universal property of tensor products. Now, observe that for $t_1, \ldots, t_n \in \mathbb{C}$ and $(a_1, \ldots, a_n) \in \prod_{i=1}^{n} \mathcal{T}_{k_i}(V, \chi_i)$,

$$f_{\ell_1, \ldots, \ell_r}(t_1 a_1, \ldots, t_n a_n) = t^{\alpha_{\ell_1, \ldots, \ell_r}} f_{\ell_1, \ldots, \ell_r}(a_1, \ldots, a_n) \tag{147}$$

where $t^{\alpha_{\ell_1, \ldots, \ell_r}} := t_1^{\alpha_1} \cdots t_n^{\alpha_n}$. Hence, arguing as in Lemma 2, we have that for any $g \in G$ and all $(a_1, \ldots, a_n) \in \prod_{i=1}^{n} \mathcal{T}_{k_i}(V, \chi_i)$,

$$\sum_{r=0}^{\infty} \sum_{1 \leq \ell_1 \leq \cdots \leq \ell_r \leq n} t^{\alpha_{\ell_1, \ldots, \ell_r}} g \cdot f_{\ell_1, \ldots, \ell_r}(a_1, \ldots, a_n)$$

$$= \sum_{r=0}^{\infty} \sum_{1 \leq \ell_1 \leq \cdots \leq \ell_r \leq n} t^{\alpha_{\ell_1, \ldots, \ell_r}} f_{\ell_1, \ldots, \ell_r}(g \cdot a_1, \ldots, g \cdot a_n). \tag{148}$$

Hence, by the uniqueness of coefficients for entire functions functions that are equal[6], we conclude that for any $g \in G$ and all $(a_1, \ldots, a_n) \in \prod_{i=1}^{n} \mathcal{T}_{k_i}(V, \chi_i)$,

$$g \cdot f_{\ell_1, \ldots, \ell_r}(a_1, \ldots, a_n) = f_{\ell_1, \ldots, \ell_r}(g \cdot a_1, \ldots, g \cdot a_n), \tag{149}$$

and so that the $f_{\ell_1, \ldots, \ell_r}$ are $G$-equivariant. $\qquad \square$

### G.3 REAL EQUIVARIANT TENSOR MAPS (AND PROOF OF THEOREM 2)

All the definitions in the previous subsection can be specialized to the real case. Hence we will have a self-paired real vector space $(V, \langle \cdot, \cdot \rangle)$ on which a group $G$ acts (rationally) in a structure-preserving way. Then we get the family of (rational) $G$-modules:

$$\mathcal{T}_k(V, \chi) := (V^{\otimes k}, \langle \cdot, \cdot \rangle)$$

where $\chi : G \to \mathbb{R}^*$ is a one-dimensional (rational) group-homomorphism of $G$. Together with this family, we have the $k$-contractions given by

$$\iota_k^G : \mathcal{T}_{2k+k'}(V, \chi) \to \mathcal{T}_{k'}(V, \chi), \tag{150}$$

which are $G$-equivariant linear maps. Then we get a very similar theorem to Theorem 4 from which Theorem 2 follows.

**Theorem 5.** Let $G \subset GL(V)$ be either a compact or a complexly averagable real linear algebraic group acting rationally in a structure-preserving way on a self-paired vector space $(V, \langle \cdot, \cdot \rangle)$ and $f : \prod_{i=1}^{n} \mathcal{T}_{k_i}(V, \chi_i) \to \mathcal{T}_{k'}(V, \chi')$ a $G$-equivariant entire function. Then we may write $f$ as follows:

$$f(a_1, \ldots, a_n) = \sum_{r=0}^{\infty} \sum_{1 \leq \ell_1 \leq \cdots \leq \ell_r \leq n} \iota_{k_{\ell_1, \ldots, \ell_r}}^G (a_{\ell_1} \otimes \ldots \otimes a_{\ell_r} \otimes c_{\ell_1, \ldots, \ell_r}) \tag{151}$$

where $c_{\ell_1, \ldots, \ell_r} \in \mathcal{T}_{k_{\ell_1, \ldots, \ell_r} + k'}(\mathbb{R}^d, \chi_{\ell_1, \ldots, \ell_r} \chi')$ is a $G$-isotropic tensor for $k_{\ell_1, \ldots, \ell_r} := \sum_{q=1}^{r} k_{\ell_q}$ and $\chi_{\ell_1, \ldots, \ell_r} = \prod_{q=1}^{r} \chi_{\ell_q}$.

*Proof of Theorem 2.* This is just a particular case of Theorem 5 as both $O(s, d-s)$ and $Sp(d)$ are both real linear algebraic groups and their complexifications have a Zariski-dense compact subgroup. $\quad \square$

*Proof of Theorem 5.* When $G$ is compact, we can just repeat the proof for the orthogonal group. When $G$ is a linear algebraic group such that its complexification has a Zariski-dense compact subgroup, we can extend, using the same analytic expression evaluated in the complex tensors, the $G$-equivariant map $f : \prod_{i=1}^{n} \mathcal{T}_{k_i}(V, \chi_i) \to \mathcal{T}_{k'}(V, \chi')$ to a complex $G^{\mathbb{C}}$-equivariant map $f^{\mathbb{C}} : \prod_{i=1}^{n} \mathcal{T}_{k_i}(V^{\mathbb{C}}, \chi_i) \to \mathcal{T}_{k'}(V^{\mathbb{C}}, \chi')$. The map becomes $G^{\mathbb{C}}$-equivariant, because $G$ is Zariski-dense inside $G_{\mathbb{C}}$ by Proposition 5.

But for $a \in \prod_{i=1}^{n} \mathcal{T}_{k_i}(V, \chi_i)$, we have that

$$f(a) = \frac{1}{2} f^{\mathbb{C}}(a) + \frac{1}{2} \overline{f^{\mathbb{C}}(a)}, \tag{152}$$

by reality of the input and output. Hence, by linearity, we can change the not necessarily real $c_{\ell_1, \ldots, \ell_r}$ by the still $G$-isotropic and real $\frac{1}{2} c_{\ell_1, \ldots, \ell_r} + \frac{1}{2} \overline{c_{\ell_1, \ldots, \ell_r}}$. The latter is $G$-isotropic, finishing the proof. $\quad \square$

---

[6]The statement is qualitatively different from (Cox et al., 2015, Chapter 1 §1 Proposition 5), but its proof is similar. We only need to use that a univariate entire function which vanishes in an infinite set with an accumulation point has to vanish everywhere.

## G.4    Proof of Corollary 3

The following proposition is needed to prove the above corollary.

**Proposition 7.** (Roe Goodman, 2009, Theorem 5.3.3) Let $G$ be either $O(s, k-s)$ or $Sp(d)$ and $\langle \cdot, \cdot \rangle$ be the corresponding non-degenerate bilinear form fixed by the usual action of $G$ on $\mathbb{R}^d$, i.e., $\langle \cdot, \cdot \rangle_s$ for $O(s, k-s)$ and $\langle \cdot, \cdot \rangle_{\text{symp}}$ for $Sp(d)$. The subspace of $G$-isotropic tensors in $\mathcal{T}_k(\mathbb{R}^d, \chi_0)$, where $\chi_0$ the constant map to 1, consist only of the zero tensor if $k$ is odd, and it is of the form

$$\sum_{\sigma \in S_k} \alpha_\sigma \left( \theta_G^{\otimes k/2} \right)^\sigma \tag{153}$$

with the $\alpha_\sigma \in \mathbb{R}$ and $\theta_G \in (\mathbb{R}^d)^{\otimes 2}$ the only tensor such that for all $v \in \mathbb{R}^d$, $\iota_1^G(v \otimes \theta_G) = v$, if $k$ is even.

**Remark 7.** Recall that $\theta_G = [\mathbb{I}_{s,d-s}]_{i,j}$ if $G = O(s, d-s)$ and $\theta_G = [J_d]_{i,j}$ if $G = Sp(d)$.

**Remark 8.** Note that the above sum can be written with less summands using the methods of Appendix D.

*Proof of Corollary 3.* By Theorem 2, Proposition 7 and linearity, we can assume, without loss of generality, that

$$f(v_1, \ldots, v_n) = \iota_{r+k} \left( v_{\ell_1} \otimes \cdots \otimes v_{\ell_r} \otimes \theta_G^{\frac{r+k'}{2}} \right)$$

with $1 \le \ell_1 \le \cdots \le \ell_r \le n$ and $r + k'$ even.

Now, the proof is very similar to that of Corollary 1. However, note that now, we write

$$\theta = e_i \otimes \tilde{e}_i,$$

where $\{e_i \mid i \in [d]\}$ and $\{\tilde{e}_i \mid i \in [d]\}$ are dual basis to each other, i.e., for all $i, j$, $\langle e_i, \tilde{e}_j \rangle = \delta_{i,j}$. The reason we have to pick a couple of bases is that the bilinear form is not necessarily an inner product.

Now, the proof becomes the same as that of Corollary 1, but we have to be careful regarding the $e_i$ and the $\tilde{e}_i$. However, after making the pairings for contraction, we get four cases:

1. $\langle v, e_j \rangle \langle w, \tilde{e}_j \rangle = \pm \langle v, w \rangle$, where the sign depends on whether $\langle \cdot, \cdot \rangle$ is symmetric or antisymmetric.

2. $\langle v, \tilde{e}_j \rangle e_j = v$.

3. $\langle v, e_j \rangle \tilde{e}_j = \pm v$, where the sign depends on whether $\langle \cdot, \cdot \rangle$ is symmetric or antisymmetric.

4. $e_j \otimes \tilde{e}_j = \pm \theta$, where the sign depends on whether $\langle \cdot, \cdot \rangle$ is symmetric or antisymmetric, or $\sum_j \tilde{e}_j \otimes e_j = \theta_G$.

Now, putting these back together as we did in the proof of Corollary 1 gives the desired statement. $\square$

# H    Stress-Strain Tensor

## H.1    Data

We use the Neo-Hookean material dataset from Garanger et al. (2024) which can be found here: `https://github.com/kgaranger/TFENN-examples`. We used training sets of $5\,000$, $20\,000$, and $40\,000$ samples, and validation and test datasets of $4\,000$ samples each. Similar to Garanger et al. (2024), we normalize the data for each model individually during training, then compare the outputs under the same scaling for an apples-to-apples comparison. For the baseline model, we shift and scale the data so that the mean and standard deviation of the components are 0 and 1 respectively. For our equivariant model, we scale the matrices so that the mean and standard deviation of their eigenvalues are 0 and 1 respectively.

## H.2 MODELS AND TRAINING

Our equivariant model takes the input matrix, performs the eigenvalue decomposition, sends the eigenvalues through a permutation equivariant network, then takes the output and uses the eigenvectors the reconstruct a matrix. The permutation equivariant network has three hidden layers following Maron et al. (2019) with a width of 23. We compare against a baseline MLP model which also uses 3 hidden layers, but has a width of 32 to result in roughly the same number of parameters. We also compare against the baseline MLP trained on an augmented dataset. For each data point in the original training dataset, we sample four elements uniformly from $O(d)$ and use them to rotate the input and output tensors. Thus the resulting dataset is four times the size of the original. Each model uses the GELU Hendrycks & Gimpel (2023) non-linearity, which we found significantly improves the accuracy.

Each model is trained with the AdamW Loshchilov & Hutter (2019) optimizer using a cosine annealing learning rate schedule Smith & Topin (2019) for 1500 epochs and batch size of 256. The baseline models were trained with learning rate `3e-3` while the equivariant model used `2e-3` when the dataset had $5\,000$ samples and `1e-3` otherwise. The experiments were run on a single RTX 6000 Ada GPU.

| model | parameter count | layer width | learning rate |
|---|---|---|---|
| MLP Baseline | 2 729 | 32 | 3e-3 |
| MLP augmented | 2 729 | 32 | 3e-3 |
| TFENN | 2 278 | 23 | |
| Ours | 2 278 | 23 | 1e-3 |

Table 4: Parameter count and learning rate for each model. All models use three hidden layers. Numbers for TFENN are taken from Garanger et al. (2024).

## I PATH SIGNATURES

### I.1 DATA

For the orthogonal group experiments, we generate paths in $d = 3$ dimensions using degree 5 polynomials for each coordinate in the domain $u \in [-1, 1]$. The coefficients of the polynomial are sampled uniformly from $[-1, 1]$. We calculate the signature tensors numerically using Signax Tong (2023) with 1000 points from the path and select $n = 10$ evenly spaced points for the input data. We truncate the signature to first, second, and third order tensors. For the Lorentz group, we use 3 spatial dimensions and 1 time dimension for $d = 4$ total dimensions. The train, validation, and test data sets each have 1024 trajectories. We normalize the data for each model individually during training, then compare the outputs under the same scaling for an apples-to-apples comparison.

### I.2 MODELS AND TRAINING

Our model calculates the inner products of all pairs of vectors which are then input into an MLP with three hidden layers with a width of 32 and GeLU nonlinearities Hendrycks & Gimpel (2023). The output of the model is the coefficients of a linear combination of basis elements specified by Corollary 1. The baseline models merely input all the vectors into an MLP and directly calculate the path signature tensors. The baseline models also have three hidden layers and GeLU nonlinearities, and the same width model has width 32 while the same params model has width 128. For the Lorentz data set, the same params model has width 116. This information is summarized in Table 5.

We also compare against the baseline MLP trained on an augmented dataset where each data point has been replaced by four transformed copies of that data point. For the orthogonal group, we can sample four elements uniformly from the Haar measure of $O(d)$ and use them to rotate the input and output tensors. Since the Lorentz group is not compact, we instead sample elements from a compact subgroup in the following way. First we sample a boost vector $\vec{\beta} \in \mathbb{R}^3$ whose entries $\beta_i$ are from a truncated normal distribution with mean 0, variance 1, lower bound $\frac{-1}{\sqrt{3}}$, and upper bound $\frac{1}{\sqrt{3}}$, which

ensures $\left\|\vec{\beta}\right\| \leq 1$. We then construct the pure boost transformation matrix:

$$\Lambda(\beta) = \begin{bmatrix} \gamma & -\gamma\beta_x & -\gamma\beta_y & -\gamma\beta_z \\ -\gamma\beta_x & 1+(\gamma-1)\frac{\beta_x^2}{\|\vec{\beta}\|^2} & (\gamma-1)\frac{\beta_x\beta_y}{\|\vec{\beta}\|^2} & (\gamma-1)\frac{\beta_x\beta_z}{\|\vec{\beta}\|^2} \\ -\gamma\beta_y & (\gamma-1)\frac{\beta_x\beta_y}{\|\vec{\beta}\|^2} & 1+(\gamma-1)\frac{\beta_y^2}{\|\vec{\beta}\|^2} & (\gamma-1)\frac{\beta_y\beta_z}{\|\vec{\beta}\|^2} \\ -\gamma\beta_z & (\gamma-1)\frac{\beta_x\beta_z}{\|\vec{\beta}\|^2} & (\gamma-1)\frac{\beta_y\beta_z}{\|\vec{\beta}\|^2} & 1+(\gamma-1)\frac{\beta_z^2}{\|\vec{\beta}\|^2} \end{bmatrix} \tag{154}$$

$$= \begin{bmatrix} \gamma & -\gamma\vec{\beta}^\top \\ -\gamma\vec{\beta} & \mathbb{I}_3 + (\gamma-1)\frac{\beta\beta^\top}{\|\vec{\beta}\|^2} \end{bmatrix}, \tag{155}$$

where $\gamma = \frac{1}{\sqrt{1-\|\vec{\beta}\|^2}}$. See for example Mansuripur (2020) for a derivation of this matrix. Next we sample an orthogonal matrix $Q \sim O(3)$ and construct the matrix $R(Q) = \begin{bmatrix} 1 & 0 \\ 0 & Q \end{bmatrix}$. Finally we sample $B = +1$ or $-1$ from a Bernoulli random variable to make the time inversion matrix $T(B) = \begin{bmatrix} B & 0 \\ 0 & \mathbb{I}_3 \end{bmatrix}$. Our Lorentz transformation is the product of all these matrices, $L = T(B)\Lambda(\beta)R(Q)$. For the orthogonal group and the Lorentz group, the resulting dataset is four times the size of the original.

| data set | model | parameter count | layer width | learning rate |
|---|---|---|---|---|
| O(d) | Baseline Same Width | 4 391 | 32 | 5e−3 |
| | Baseline Same Params | 42 023 | 128 | 1e−3 |
| | Baseline Augmented | 42 023 | 128 | 1e−3 |
| | Ours | 41 557 | 32 | 5e−4 |
| Lorentz | Baseline Same Width | 6 196 | 32 | 5e−3 |
| | Baseline Same Params | 41 728 | 116 | 1e−3 |
| | Baseline Augmented | 41 728 | 116 | 1e−3 |
| | Ours | 41 557 | 32 | 5e−4 |

Table 5: Parameter count and learning rate for each model. Since all models have the same number of hidden layers of the same width, the difference in the number of parameters is driven by the different inputs and outputs of each model.

For training we use the AdamW optimizer Loshchilov & Hutter (2019) with a cosine annealing learning rate schedule Smith & Topin (2019). To determine the peak learning rate, we held all other hyperparameters fixed and varied the learning rate. We used a batch size of 32 and trained for 500 epochs. The experiments were run on a single RTX 6000 Ada GPU and took under 10 minutes to train per model, per trial.

## J  SPARSE VECTOR ESTIMATION DETAILS

### J.1  PROBLEM SETTING

The methods developed in Hopkins et al. (2016) and Mao & Wein (2022) each derive an $h$ function and prove using sum-of-squares methods that with high probability that the solutions are accurate. Their proofs hold under particular assumptions, including that for sparsity parameter $\varepsilon \leq 1/3$, $\|v\|_4^4 \geq \frac{1}{\varepsilon n}$ and that $v_0 = v$ and $v_1, \ldots, v_{d-1} \sim \mathcal{N}\left(\mathbf{0}_n, \frac{1}{n}\mathbb{I}_n\right)$.

In our experiments we violate some or all of these assumptions and use data to find the best $h$ function for these new, unexplored settings. We sample the sparse vector using four different methods: Accept/Reject (AR), Bernoulli-Gaussian (BG), Corrected Bernoulli-Gaussian (CBG), and Bernoulli-Rademacher (BR). Only the Accept/Reject method explicitly satisfies $\|v\|_4^4 \geq \frac{1}{\varepsilon n}$, the others only satisfy this condition in expectation. Additionally, we sample $v_1, \ldots, v_{d-1} \sim \mathcal{N}(\mathbf{0}_n, \Sigma)$ where $\Sigma$ can be the identity, a non-identity diagonal covariance, or a random covariance from a Wishart distribution.

## J.2 EQUIVARIANCE IN SPARSE VECTOR RECOVERY

In this section, we show a sufficient condition for the $O(d)$-invariance of sparse vector estimation. We start with a lemma on the equivariance of finding an eigenvector.

**Lemma 6.** Let $b$ be a $2_{(+)}$-tensor and let $g \in O(d)$. If $u$ is an eigenvector for eigenvalue $\lambda$ of $M(g) \, b \, M(g)^\top$, then $M(g)^\top u$ is an eigenvector for eigenvalue $\lambda$ of $b$.

*Proof.* Let $b$ be a $2_{(+)}$-tensor, let $g \in O(d)$, and let $\lambda, u$ be an eigenvalue, eigenvector pair of $M(g) \, b \, M(g)^\top$.

$$(M(g) \, b \, M(g)^\top) \, u = \lambda u \Rightarrow b(M(g)^\top u) = \lambda(M(g)^\top u) \,.$$

Thus $M(g)^\top u$ is an eigenvector for eigenvalue $\lambda$ of $b$. $\qquad\square$

**Proposition 8.** Let $S \in \mathbb{R}^{n \times d}$ with rows $a_i^\top \in \mathbb{R}^d$ so that $a_i$ are column vectors. We define the action of $O(d)$ on $S$ for all $g \in O(d)$ as $S \, M(g)$, and therefore $M(g)^\top a_i$ for the rows. Let $f : \mathbb{R}^{n \times d} \to \mathbb{R}^n, h : \left(\mathbb{R}^d\right)^n \to \mathbb{R}^{d \times d}$ symmetric such that $f(S) = S \, \lambda_{\text{vec}}(h(a_1, \ldots, a_n))$ where $\lambda_{\text{vec}}(\cdot)$ returns a normalized eigenvector for the top eigenvalue of the input symmetric matrix. If $h$ is $O(d)$-equivariant, then $f$ is $O(d)$-invariant.

*Proof.* Let $S, h$, and $f$ be defined as above. Suppose that $h$ is $O(d)$-equivariant. Suppose $\lambda_{\text{vec}}\left(M(g)^\top h(a_1, \ldots, a_n) \, M(g)\right) = u$, then by lemma 6, up to a sign flip, we have:

$$\lambda_{\text{vec}}\left(M(g)^\top h(a_1, \ldots, a_n) \, M(g)\right) = u = M(g)^\top M(g) \, u = M(g)^\top \lambda_{\text{vec}}(h(a_1, \ldots, a_n)) \quad (156)$$

Thus,

$$f(g \cdot S) = (g \cdot S) \, \lambda_{\text{vec}}\left(h\left(g^{-1} \cdot a_1, \ldots, g^{-1} \cdot a_n\right)\right) \tag{157}$$

$$= (g \cdot S) \, \lambda_{\text{vec}}\left(g^{-1} \cdot h(a_1, \ldots, a_n)\right) \tag{158}$$

$$= S \, M(g) \, \lambda_{\text{vec}}\left(M(g)^\top h(a_1, \ldots, a_n) \, M(g)\right) \tag{159}$$

$$= S \, M(g) \, M(g)^\top \lambda_{\text{vec}}(h(a_1, \ldots, a_n)) \tag{160}$$

$$= S \, \lambda_{\text{vec}}(h(a_1, \ldots, a_n)) \tag{161}$$

$$= f(S) \,. \tag{162}$$

This completes the proof. $\qquad\square$

## J.3 DATA

First we generate the sparse vectors with one of the following sampling procedures for sparsity $\varepsilon \le 1/3$.

**Accept/Reject (A/R).** A random vector $v_0 \sim \mathcal{N}(\mathbf{0}_n, \mathbb{I}_n)$ is sampled and normalized to unit $\ell_2$ length. We accept it if $\|v_0\|_4^4 \ge \frac{1}{\varepsilon n}$ and otherwise reject it. Note that the sparsity of $v_0$ is not explicitly imposed, but the 4-norm condition suggests that $v_0$ is approximately sparse. The 4-norm condition of sparsity is used in Hopkins et al. (2016).

**Bernoulli-Gaussian (BG)** This sampling procedure, considered in Mao & Wein (2022), defines $v_0$ as

$$\begin{cases} [v_0]_i = 0 & \text{with probability } 1 - \varepsilon \\ [v_0]_i \sim \mathcal{N}\left(0, \frac{1}{\varepsilon n}\right) & \text{with probability } \varepsilon. \end{cases} \tag{163}$$

Note that under this sampling procedure $\mathbb{E}\|v_0\|_4^4 = \frac{3}{\varepsilon n}$.

**Corrected Bernoulli-Gaussian (CBG)** We consider a modified version of the Bernoulli-Gaussian that replaces the values set to exactly 0 in the Bernoulli-Gaussian distribution with values sampled from a Gaussian with small variance. Under this distribution we have $\mathbb{E}\|v_0\|_2 = 1$ and $\mathbb{E}\|v_0\|_4^4 = \frac{1}{\varepsilon n}$.

$$\begin{cases} [v_0]_i \sim \mathcal{N}\left(0, \frac{1 - \varepsilon - \sqrt{\frac{1}{3}(1-\varepsilon)(1-3\varepsilon)}}{(1-\varepsilon)n}\right) & \text{with probability } 1 - \varepsilon \\ [v_0]_i \sim \mathcal{N}\left(0, \frac{\varepsilon + \sqrt{\frac{1}{3}(1-\varepsilon)(1-3\varepsilon)}}{\varepsilon n}\right) & \text{with probability } \varepsilon. \end{cases} \tag{164}$$

**Bernoulli-Rademacher (BR)** This sampling procedure, studied in Mao & Wein (2022), defines $v_0$ as

$$[v_0]_i = \begin{cases} 0 & \text{with probability } 1 - \varepsilon \\ \frac{1}{\sqrt{\varepsilon n}} & \text{with probability } \frac{\varepsilon}{2} \\ \frac{-1}{\sqrt{\varepsilon n}} & \text{with probability } \frac{\varepsilon}{2}. \end{cases} \tag{165}$$

Under this distribution we have $\mathbb{E}\|v_0\|_2 = 1$ and $\mathbb{E}\|v_0\|_4^4 \geq \frac{1}{\varepsilon n}$.

Since the BG, CBG, and BR distributions have $\mathbb{E}\|v_0\|_2 = 1$, we also normalize these vectors to unit $\ell_2$ length after generating them.

**Proposition 9.** Let $v_0$ be a Bernoulli-Gaussian vector. Then $\mathbb{E}\left[\|v_0\|_2^2\right] = 1$ and $\mathbb{E}\left[\|v_0\|_4^4\right] = \frac{3}{\varepsilon n}$.

*Proof.* Let $\varepsilon \in (0, 1]$ and let $v_0$ be a Bernoulli-Gaussian sparse vector. Thus

$$\mathbb{E}\left[\|v_0\|_2^2\right] = \mathbb{E}\left[\sum_{i=1}^n [v_0]_i^2\right] = \sum_{i=1}^n \mathbb{E}\left[[v_0]_i^2\right]. \tag{166}$$

Thus, we need to find the 2nd moment of an entry of $[v_0]_i$, which we will do by first calculating its moment generating function. If $Z$ is a Bernoulli-Gaussian random variable, then $Z = XY$ where $X$ and $Y$ are random variables with $X \sim \text{Bern}(\varepsilon)$ and $Y \sim \mathcal{N}\left(0, \frac{1}{\varepsilon n}\right)$. Then

$$\mathbb{E}[\exp\{tXY\}] = \mathbb{E}[\mathbb{E}[\exp\{tXY\}|X]] \tag{167}$$
$$= \mathbb{E}[\exp\{tXY\}|X = 0]P(X = 0) + \mathbb{E}[\exp\{tXY\}|X = 1]P(X = 1) \tag{168}$$
$$= \mathbb{E}[\exp\{0\}](1 - \varepsilon) + \varepsilon\mathbb{E}[\exp\{tY\}] \tag{169}$$
$$= (1 - \varepsilon) + \varepsilon\mathbb{E}[\exp\{tY\}]. \tag{170}$$

Since $\mathbb{E}[\exp\{tY\}]$ is the moment generating function of $Y$, a Gaussian random variable, we can see that the 2nd moment of $Z$ is the 2nd moment of $Y$ multiplied by $\varepsilon$. Then

$$\sum_{i=1}^n \mathbb{E}\left[[v_0]_i^2\right] = \sum_{i=1}^n \varepsilon\left(\frac{1}{\varepsilon n}\right) = \sum_{i=1}^n \frac{1}{n} = 1. \tag{171}$$

Now, for the sparsity condition, we have

$$\mathbb{E}\left[\|v_0\|_4^4\right] = \sum_{i=1}^n \mathbb{E}\left[[v_0]_i^4\right] = \sum_{i=1}^n \varepsilon\left(3\left(\frac{1}{\varepsilon n}\right)^2\right) = \sum_{i=1}^n \frac{3}{\varepsilon n^2} = \frac{3}{\varepsilon n}. \tag{172}$$

This follows because our previous analysis shows that the 4th moment of an entry of $[v_0]_i$ is $3\sigma^4 = 3\left(\frac{1}{\varepsilon n}\right)^2$. This completes the proof. □

**Proposition 10.** Let $v_0$ be a Corrected Bernoulli-Gaussian vector. Then $\mathbb{E}\left[\|v_0\|_2^2\right] = 1$ and $\mathbb{E}\left[\|v_0\|_4^4\right] = \frac{1}{\varepsilon n}$.

*Proof.* Let $\varepsilon \in \left(0, \frac{1}{3}\right]$ and let $v_0 \in \mathbb{R}^n$ be a Corrected Bernoulli-Gaussian sparse vector. Thus

$$\mathbb{E}\left[\|v_0\|_2^2\right] = \mathbb{E}\left[\sum_{i=1}^n [v_0]_i^2\right] = \sum_{i=1}^n \mathbb{E}\left[[v_0]_i^2\right]. \tag{173}$$

Thus, we need to find the 2nd moment of an entry of $[v_0]_i$, which we will do by first calculating its moment-generating function. If $Z$ is a Corrected Bernoulli-Gaussian random variable, then $Z = XY + (1-X)W$ where $X, Y,$ and $W$ are random variables with $X \sim \text{Bern}(\varepsilon)$, $Y \sim \mathcal{N}\left(0, \frac{\varepsilon+q}{\varepsilon n}\right)$, and $W \sim \mathcal{N}\left(0, \frac{1-\varepsilon-q}{n(1-\varepsilon)}\right)$ where $q = \sqrt{\frac{1}{3}(1-\varepsilon)(1-3\varepsilon)}$. Then

$$\mathbb{E}[\exp\{t(XY + (1-X)W)\}] = \mathbb{E}[\mathbb{E}[\exp\{t(XY + (1-X)W)\}|X]] \tag{174}$$
$$= \mathbb{E}[\exp\{tW\}|X = 0]P(X = 0) + \mathbb{E}[\exp\{tY\}|X = 1]P(X = 1) \tag{175}$$
$$= (1-\varepsilon)\mathbb{E}[\exp\{tW\}] + \varepsilon\mathbb{E}[\exp\{tY\}] \tag{176}$$

Since $\mathbb{E}[\exp\{tW\}]$ is the moment generating function of $W$ and $\mathbb{E}[\exp\{tY\}]$ is the moment generating function of $Y$, we can immediately get the moments of $Z$. Then

$$\sum_{i=1}^{n} \mathbb{E}\left[[v_0]_i^2\right] = \sum_{i=1}^{n} \varepsilon\left(\frac{\varepsilon+q}{n\varepsilon}\right) + (1-\varepsilon)\left(\frac{1-\varepsilon-q}{n(1-\varepsilon)}\right) = \sum_{i=1}^{n} \frac{\varepsilon+q+1-\varepsilon-q}{n} = 1 . \quad (177)$$

For the sparsity condition, we use the same result above but now for the 4th moment

$$\mathbb{E}\left[\|v_0\|_4^4\right] = \sum_{i=1}^{n} \mathbb{E}\left[[v_0]_i^4\right] \quad (178)$$

$$= \sum_{i=1}^{n} \varepsilon\left(3\left(\frac{\varepsilon+q}{n\varepsilon}\right)^2\right) + (1-\varepsilon)\left(3\left(\frac{1-\varepsilon-q}{n(1-\varepsilon)}\right)^2\right) \quad (179)$$

$$= \sum_{i=1}^{n} \left(\frac{3\varepsilon + (1-3\varepsilon)}{n^2\varepsilon}\right) \quad (180)$$

$$= \frac{1}{n\varepsilon} . \quad (181)$$

This completes the proof. $\qquad\square$

**Proposition 11.** Let $v_0$ be a Bernoulli-Rademacher vector. Then $\mathbb{E}\left[\|v_0\|_2^2\right] = 1$ and $\mathbb{E}\left[\|v_0\|_4^4\right] = \frac{1}{\varepsilon n}$.

*Proof.* Let $\epsilon \in (0,1]$ and let $v_0$ be a Bernoulli-Rademacher sparse vector. Thus

$$\mathbb{E}\left[\|v_0\|_2^2\right] = \mathbb{E}\left[\sum_{i=1}^{n} [v_0]_i^2\right] \quad (182)$$

$$= \sum_{i=1}^{n} \mathbb{E}\left[[v_0]_i^2\right] \quad (183)$$

$$= \sum_{i=1}^{n} (1-\epsilon)(0)^2 + \frac{\epsilon}{2}\left(\frac{1}{\sqrt{\epsilon n}}\right)^2 + \frac{\epsilon}{2}\left(\frac{-1}{\sqrt{\epsilon n}}\right)^2 \quad (184)$$

$$= \sum_{i=1}^{n} \frac{\epsilon}{\epsilon n} \quad (185)$$

$$= 1 . \quad (186)$$

We also have

$$\mathbb{E}\left[\|v_0\|_4^4\right] = \mathbb{E}\left[\sum_{i=1}^{n} [v_0]_i^4\right] \quad (187)$$

$$= \sum_{i=1}^{n} \mathbb{E}\left[[v_0]_i^4\right] \quad (188)$$

$$= \sum_{i=1}^{n} (1-\epsilon)(0)^4 + \frac{\epsilon}{2}\left(\frac{1}{\sqrt{\epsilon n}}\right)^4 + \frac{\epsilon}{2}\left(\frac{-1}{\sqrt{\epsilon n}}\right)^4 \quad (189)$$

$$= \sum_{i=1}^{n} \frac{\epsilon}{\epsilon^2 n^2} \quad (190)$$

$$= \frac{1}{\epsilon n} . \quad (191)$$

This completes the proof. $\qquad\square$

### J.4 CONSTRUCTION OF OBSERVED SUBSPACE

Next, we generate the noise vectors that determine the rest of the subspace. For a particular experiment trial, we generate a covariance matrix that is either the identity matrix, a diagonal matrix, or a random symmetric positive definite matrix. The diagonal matrix has diagonal entries $[\Sigma]_{i,i} \sim \text{Unif}(\frac{1}{2}, \frac{3}{2})$. For the random covariance matrix, first we generate an $n \times n$ matrix $M$ with entries $[M]_{i,j} \sim \mathcal{N}(0, 1)$, then set $\Sigma = M M^\top + 0.00001 \, \mathbb{I}_n$. We then generate the noise vectors $v_1, \dots, v_{d-1} \sim \mathcal{N}(0, \Sigma)$.

Finally, we use the following algorithm to get a random orthonormal basis of $\text{span}\{v_0, \dots, v_{d-1}\}$. Let $B$ be the matrix with columns $v_0, \dots, v_{d-1}$, and sample an orthogonal matrix $O$ from $O(d)$. Multiply $B\,O$ and take the Q-R factorization, then $Q$ is a random orthonormal basis of $\text{span}\{v_0, \dots, v_{d-1}\}$. We prove this below with the additional assumption that the $v_0, \dots, v_{d-1}$ are linearly independent, which is reasonable given that $d \ll n$ and we are generating these vectors randomly.

**Proposition 12.** Let $n \geq d$, and let $B$ be the $n \times d$ matrix with $v_0, \dots, v_{d-1}$ as the columns. Assume that $v_0, \dots, v_{d-1}$ are linearly independent, so rank $B = d$. Let $O$ be a $d \times d$ orthogonal matrix and $Q\,R = B\,O$ be a Q-R factorization of $B\,O$. Then the columns of $Q$ form an orthonormal basis of $\text{span}\{v_0, \dots, v_{d-1}\}$.

*Proof.* The Q-R factorization gives us that the columns of $Q$ are orthonormal. Thus we just have to show that $\text{span}\{v_0, \dots, v_{d-1}\} = \text{span}\{Q_0, \dots, Q_{d-1}\}$

Let $a \in \text{span}\{v_0, \dots, v_{d-1}\}$, so for some $\alpha_0, \dots, \alpha_{d-1}$ we have $a = \alpha_0 v_0 + \dots + \alpha_{d-1} v_{d-1}$. Let $\alpha \in \mathbb{R}^d$ be the vector of these coefficients, and then we have,

$$a = B\,\alpha = B\,O\,O^\top \alpha = Q\,R\,O^\top \alpha = Q\hat{\alpha} \,. \tag{192}$$

Thus $\hat{\alpha} \in \mathbb{R}^d$ is a vector of coefficients, so $a \in \text{span}\{Q_0, \dots, Q_{d-1}\}$. Therefore, $\text{span}\{v_0, \dots, v_{d-1}\} \subseteq \text{span}\{Q_0, \dots, Q_{d-1}\}$.

Now let $b \in \text{span}\{Q_0, \dots, Q_{d-1}\}$, so for some $\beta_0, \dots, \beta_{d-1}$ we have $b = \beta_0 Q_0 + \dots + \beta_{d-1} Q_{d-1}$. Let $\beta \in \mathbb{R}^d$ be a vector of the coefficients $\beta_0, \dots, \beta_{d-1}$. Now, since rank $B = d$, rank $B\,O = d$, so in the Q-R factorization, the upper triangular $R$ has positive diagonal entries, so it is invertible [Horn & Johnson (1990), Theorem 2.1.14]. Then we have,

$$b = Q\,\beta = Q\,R\,R^{-1}\,\beta = B\,O\,R^{-1}\,\beta = B\hat{\beta} \,. \tag{193}$$

Thus $\hat{\beta} \in \mathbb{R}^d$ is a vector of coefficients, so $b \in \text{span}\{v_0, \dots, v_{d-1}\}$. Therefore, $\text{span}\{Q_0, \dots, Q_{d-1}\} \subseteq \text{span}\{v_0, \dots, v_{d-1}\}$ which completes the proof.

$\square$

### J.5 MODELS

The $h$ function given in Hopkins et al. (2016) is

$$h(a_1, \dots, a_n) := \sum_{i=1}^n \left( \|a_i\|_2^2 - \frac{d}{n} \right) a_i a_i^\top \,, \tag{194}$$

and given in Mao & Wein (2022) is

$$h(a_1, \dots, a_n) := \sum_{i=1}^n \left( \|a_i\|_2^2 - \frac{d-1}{n} \right) a_i a_i^\top - \frac{3}{n}\mathbb{I}_n \,. \tag{195}$$

Note that equations (194) and (195) are $O(d)$-equivariant and a special case of Corollary 1 since they define a sum of outer products of the inputs with coefficients that are polynomial functions of inner products of the inputs.

In comparison to these fixed methods, we propose two machine learning-based models defined using the results of Section 3. The first model parametrizes

$$h(a_1, \dots, a_n) = \left[ \sum_{i=1}^n \sum_{j=i}^n q_{i,j}\Big( (\langle a_\ell, a_m \rangle)_{\ell,m=1}^n \Big) \frac{1}{2} \big( a_i a_j^\top + a_j a_i^\top \big) \right] + q_{\mathbb{I}}\Big( (\langle a_\ell, a_m \rangle)_{\ell,m=1}^n \Big)\mathbb{I}_d \,. \tag{196}$$

The Diag variant of our model only uses the norms of $a_i$ as input and $a_i a_i^\top$ as the basis elements to be more comparable to (194) and (195):

$$h(a_1, \ldots, a_n) = \left[ \sum_i^n q_i \left( \left( \|a_\ell\|_2^2 \right)_{\ell=1}^n \right) a_i a_i^\top \right] + q_{\mathbb{I}} \left( \left( \|a_\ell\|_2^2 \right)_{\ell=1}^n \right) \mathbb{I}_d \,, \tag{197}$$

where $q_{i,j}$, $q_i$, and $q_{\mathbb{I}}$ are $O(d)$-invariant scalar functions. The form of equation (196) follows from Corollary 1 as shown in Appendix J.6. By averaging the general form of a matrix valued $O(d)$-equivariant function with its transpose, we obtain the form of any $O(d)$-equivariant polynomial function that outputs a symmetric matrix. Equation (197) follows the scheme of (196) but only includes inner and outer products of the same vectors to be more directly comparable to (194) and (195). Corollary 1 specifies that $q_{i,j}, q_i$, and $q_{\mathbb{I}}$ should be polynomials, but we will approximate them with dense neural networks. The networks themselves are multi-layer perceptrons (MLP) with 2 hidden layers, width of 128, and ReLU activation functions.

To demonstrate the benefits of equivariance, we also implement a non-equivariant baseline model (BL) which takes as input the $nd$ components of $S$ and outputs the $d + \binom{d}{2}$ components of a symmetric $d \times d$ matrix. This is implemented as a multi-layer perceptron with 2 hidden layers, width of 128, and ReLU activation functions.

### J.6 DERIVATION OF (196)

In the following, we derive the general form of an $O(d)$-equivariant function $h : (\mathbb{R}^d)^n \to S_d$ stated in (196) from Corollary 1.

First, we use Corollary 1 to write the arbitrary form of an $O(d)$-equivariant function $g : (\mathbb{R}^d)^n \to \mathbb{R}^{d \times d}$ that takes values in the space of $d \times d$ matrices that are not necessarily symmetric. Given the general form of $g$, it follows that

$$h = \frac{1}{2}(g + g^\top) \tag{198}$$

is the general form of an $O(d)$-equivariant function $h : (\mathbb{R}^d)^n \to S_d$.

In the notation of Corollary 1, we seek an $O(d)$-equivariant function $g : \left( \mathcal{T}_1(\mathbb{R}^d, +) \right)^n \to \mathcal{T}_2(\mathbb{R}^d, +)$. From Corollary 1 with $k' = 2$, it follows that $g$ can be written in the form

$$g(v_1, \ldots, v_n) = \sum_{t=0}^{1} \sum_{\sigma \in S_2} \sum_{1 \leq J_1 \leq \cdots \leq J_{2-2t} \leq n} q_{t,\sigma,J} \left( (\langle v_i, v_j \rangle)_{i,j=1}^n \right) \left( v_{J_1} \otimes \cdots \otimes v_{J_{2-2t}} \otimes \delta^{\otimes t} \right)^\sigma \,. \tag{199}$$

Expanding the sum of the $t = 0$ and $t = 1$ terms, we have

$$g(v_1, \ldots, v_n) = \left( \sum_{\sigma \in S_2} \sum_{1 \leq J_1 \leq J_2 \leq n} q_{0,\sigma,J} \left( (\langle v_i, v_j \rangle)_{i,j=1}^n \right) \left( v_{J_1} \otimes v_{J_2} \right)^\sigma \right)$$
$$+ \sum_{\sigma \in S_2} q_{1,\sigma} \left( (\langle v_i, v_j \rangle)_{i,j=1}^n \right) \delta^\sigma. \tag{200}$$

The set of permutation $S_2$ consists of $(1, 2)$ and $(2, 1)$. Using the fact that $(u \otimes v)^{(1,2)} = u \otimes v$, $(u \otimes v)^{(2,1)} = v \otimes u$, and $\delta^\sigma = \delta$ for all $\sigma \in S_2$, we can write the above expression as

$$g(v_1, \ldots, v_n) = \sum_{J_1=1}^{n} \sum_{J_2=1}^{n} q_{0,J} \left( (\langle v_i, v_j \rangle)_{i,j=1}^n \right) \left( v_{J_1} \otimes v_{J_2} \right) + q_1 \left( (\langle v_i, v_j \rangle)_{i,j=1}^n \right) \delta \,, \tag{201}$$

where the double sum over $J_1$ and $J_2$ accounts for both the sum over $J_1 \leq J_2$ and the sum over the permutations in $S_2$. Next, we swap to standard matrix and vector notation as well as more simple indices to make the equations clearer for readers who are primarily interested in the application. Thus $u \otimes v \Rightarrow uv^\top, \delta \Rightarrow \mathbb{I}_d$ and $J_1, J_2, i, j$ become $i, j, \ell, m$, and we have

$$g(v_1, \ldots, v_n) = \sum_{i=1}^{n} \sum_{j=1}^{n} q_{i,j} \left( (\langle a_\ell, a_m \rangle)_{\ell,m=1}^n \right) a_i a_j^\top + q_{\mathbb{I}} \left( (\langle a_\ell, a_m \rangle)_{\ell,m=1}^n \right) \mathbb{I}_d \,. \tag{202}$$

Finally, setting $h = \frac{1}{2}(g + g^\top)$ gives the desired form of $h$ stated in (196).

## J.7 TRAINING DETAILS

The train dataset had $5\,000$ vectors and the validation and test datasets had $500$ vectors. For training our models, we used $1 - \langle \hat{v}, v_0 \rangle^2$ as the loss function. We used the Adam optimizer Kingma & Ba (2017) with an exponential decay rate of $0.999$. We used a batch size of $100$ and trained until the validation error had not improved for 20 epochs. See Table 6 for the learning rate and number of parameters for each model. We did a small exploration to find these hyper-parameters. These hyper-parameters seemed to work well, but it is always possible that better ones could be found with more exploration.

| model | parameter count | learning rate |
|---|---|---|
| Baseline | $99\,087$ | 1e-3 |
| Ours (Diag) | $58\,981$ | 5e-4 |
| Ours | $1\,331\,131$ | 3e-4 |

Table 6: Parameter count and learning rate for each model. Since all models have the same number of hidden layers of the same width, the difference in the number of parameters is driven by the different inputs and outputs of each model.

The experiments were run on a single RTX 6000 Ada GPU and took 18 hours.

| sampling | $\Sigma$ | SOS-I | SOS-II | MLP | SVH-Diag | SVH |
|---|---|---|---|---|---|---|
| A/R | Random | $0.610 \pm 0.011$ | $0.610 \pm 0.011$ | $0.647 \pm 0.177$ | $0.768 \pm 0.045$ | $\mathbf{0.966 \pm 0.001}$ |
| | Diagonal | $0.444 \pm 0.012$ | $0.444 \pm 0.012$ | $0.561 \pm 0.262$ | $0.698 \pm 0.034$ | $\mathbf{0.755 \pm 0.057}$ |
| | Identity | $0.611 \pm 0.002$ | $0.611 \pm 0.002$ | $0.494 \pm 0.285$ | $0.622 \pm 0.201$ | $\mathbf{0.647 \pm 0.289}$ |
| BG | Random | $0.963 \pm 0.001$ | $0.963 \pm 0.001$ | $0.783 \pm 0.090$ | $\mathbf{0.970 \pm 0.003}$ | $0.965 \pm 0.002$ |
| | Diagonal | $0.949 \pm 0.002$ | $0.949 \pm 0.002$ | $0.672 \pm 0.260$ | $\mathbf{0.974 \pm 0.004}$ | $0.775 \pm 0.078$ |
| | Identity | $0.963 \pm 0.000$ | $0.963 \pm 0.000$ | $0.681 \pm 0.241$ | $0.966 \pm 0.004$ | $\mathbf{0.999 \pm 0.001}$ |
| CBG | Random | $0.409 \pm 0.005$ | $0.409 \pm 0.005$ | $0.836 \pm 0.149$ | $0.490 \pm 0.089$ | $\mathbf{0.965 \pm 0.002}$ |
| | Diagonal | $0.292 \pm 0.005$ | $0.292 \pm 0.005$ | $\mathbf{0.835 \pm 0.150}$ | $0.597 \pm 0.027$ | $0.722 \pm 0.013$ |
| | Identity | $0.418 \pm 0.006$ | $0.418 \pm 0.006$ | $0.558 \pm 0.216$ | $0.368 \pm 0.119$ | $\mathbf{0.750 \pm 0.288}$ |
| BR | Random | $0.523 \pm 0.006$ | $0.523 \pm 0.006$ | $\mathbf{0.975 \pm 0.005}$ | $0.669 \pm 0.150$ | $0.970 \pm 0.002$ |
| | Diagonal | $0.340 \pm 0.010$ | $0.340 \pm 0.010$ | $\mathbf{0.943 \pm 0.008}$ | $0.701 \pm 0.041$ | $0.913 \pm 0.002$ |
| | Identity | $0.526 \pm 0.005$ | $0.526 \pm 0.005$ | $\mathbf{0.949 \pm 0.006}$ | $0.570 \pm 0.199$ | $0.898 \pm 0.001$ |

Table 7: Train error comparison of different methods under different sampling schemes for $v_0$ and different covariances for $v_1, \ldots, v_{d-1}$. The metric is $\langle v_0, \hat{v} \rangle^2$, which ranges from 0 to 1 with values closer to 1, meaning that the vectors are closer. For each row, the best value is **bolded**. For these experiments, $n = 100, d = 5, \epsilon = 0.25$, and the results were averaged over 5 trials with the standard deviation given by $\pm 0.xxx$.

