# OpenReview forum: "Tensor learning with orthogonal, Lorentz, and symplectic symmetries"
_ICLR.cc/2026/Conference — ICLR 2026 Poster_

### Official Review · Reviewer_ikq1 · 2025-10-17

**Soundness:** 4
**Presentation:** 4
**Contribution:** 4
**Rating:** 10
**Confidence:** 4

**Summary:**

The paper claims to provide a full characterisation of G-equivariant polynomial functions from tuples of tensors to tensor outputs, where the group G is a classical Lie group. The authors apply their characterisation in a number of numerical experiments where the input data is a tuple of vectors.

**Strengths:**

- This is a very strong paper, both in its theoretical contributions and practical results. I commend the authors for such a well-written, densely packed presentation. I felt like I learned a lot as I got into the weeds while reading it.
- To the best of my understanding, I can see that the authors have characterised O(d)-equivariant polynomial functions from tuples of tensors to a tensor output, and have also considered the particular case where the input is a tuple of tensors instead (so that this can be used in their experiments for practical reasons, as the authors state in the paper). I can also see that they have extended this to O(s, d-s) and Sp(d) equivariance for entire functions too. (I say best of my understanding as I haven't gone through every single line of the proofs but I have gone through a decent amount of it + with my background in this area I can see that it is clearly correct.)
- I particularly enjoyed the authors explaining complex equations in words immediately after they are introduced (e.g after Corollary 1). I will be stealing this idea for my own papers in the future.
- The results are clearly significant and deserve top billing at ICLR.

**Weaknesses:**

It is hard to come up with too many weaknesses, but I think the authors should consider the following:

- I think the claim that the authors make in lines 49-50 is too strong (this is repeated in lines 475-476). "General" sounds like it could be used for any group, which I don't think is what the authors are claiming.
- The theory looks at the generic O(d) group action, allowing p to be +1 or -1, which is extended to other classical Lie groups. However the practical experiments only consider the O(d) case for the +1 case, which (to the uninitiated) might make the theory somewhat "overkill". Are there any practical uses of their theory in the -1 case?
- Whilst the related work is pretty comprehensive, I think they are missing a reference to the potential related work [1] which looks at characterising O(d) and Sp(d) equivariant functions between tensors, exactly of the form they have considered when the input tuple is of length 1. Could the authors perhaps comment on the differences between the two papers beyond this?

[1] Pearce-Crump, E. - Brauer's Group Equivariant Neural Networks (ICML 2023).

**Questions:**

Beyond the questions asked in the Weaknesses section:

- The authors note that their Theorem 1 is impractical for computing purposes. Could they clarify further why that is, and if they have thought of any ways to mitigate this so that the theorem in all its glory could be employed in neural networks?
- I might have missed this but where are the non-linearities included in the characterisation/paper?


Other minor questions/points:
- Should there be a standard deviation reported for their model in Table 2 for O(d)?
- I think in their Example 1 equation (13), there should be a delta after beta_1<a, a> - do the authors agree?
- In line 253 I think they should refer directly to (107) instead of Appendix D for the definition of G_4 - I originally thought it was a typo for S_4.
- I think the authors in their Definition section should introduce the plus/minus notation where they write e.g 2_(+), just so that it is 100% clear to everyone. There is a lot going on and since it is used a lot (in place of p = +1, -1), I think the notation should be introduced explicitly.
- Anything else they can do (e.g in the Appendix) to make their examples even clearer (say, by providing more steps) would be useful in helping the reader to understand all of the concepts that are introduced in Section 2.

---

> ### Author Response · Authors · 2025-11-25
>
> > It is hard to come up with too many weaknesses, but I think the authors should consider the following:
> I think the claim that the authors make in lines 49-50 is too strong (this is repeated in lines 475-476). "General" sounds like it could be used for any group, which I don't think is what the authors are claiming.
>
> Thank you for this observation. You are right that the claim is too strong. We removed the first claim and rephrased the second one.
>
> > The theory looks at the generic O(d) group action, allowing p to be +1 or -1, which is extended to other classical Lie groups. However the practical experiments only consider the O(d) case for the +1 case, which (to the uninitiated) might make the theory somewhat "overkill". Are there any practical uses of their theory in the -1 case?
>
> In physics there are many examples of relevant p=-1 objects including angular velocity (pseudovector) and vorticity (pseudovector) [1]. A magnetic field is a pseudovector field which is relevant for magnetohydrodynamics which is often studied in astrophysics [2]. In chemistry, molecules can have chirality where a reflection of the molecule changes its properties.
>
> > Whilst the related work is pretty comprehensive, I think they are missing a reference to the potential related work [1] which looks at characterising O(d) and Sp(d) equivariant functions between tensors, exactly of the form they have considered when the input tuple is of length 1. Could the authors perhaps comment on the differences between the two papers beyond this?
> [1] Pearce-Crump, E. - Brauer's Group Equivariant Neural Networks (ICML 2023).
>
> Thank you for pointing out this reference, which we now cite, that considers the special case of equivariant functions whose inputs and outputs are tensor products of R^n.
>
> > The authors note that their Theorem 1 is impractical for computing purposes. Could they clarify further why that is, and if they have thought of any ways to mitigate this so that the theorem in all its glory could be employed in neural networks?
>
> When the input is higher order tensors, the polynomials do not break down nicely into inner products. For example, if the input is matrices, any product of the matrices is O(d)-equivariant, but they do not all reduce to scalars. If you do not consider all possible contractions but only the full inner product of tensors, you would get an equivariant architecture, but it would not be fully expressive. Further expressive, equivariant architectures would require additional theory work.
>
> > I might have missed this but where are the non-linearities included in the characterisation/paper?
>
> The non-linearities appear in $q$ functions in Corollary 1, and the $\tilde f$ function in Corollary 2.
>
> > Should there be a standard deviation reported for their model in Table 2 for O(d)?
>
> We don’t report them because they are smaller than 1e-3 (see caption).
>
> > I think in their Example 1 equation (13), there should be a delta after beta_1<a, a> - do the authors agree?
>
> Yes, fixed!
>
> > In line 253 I think they should refer directly to (107) instead of Appendix D for the definition of G_4 - I originally thought it was a typo for S_4.
>
> Thank you, we have updated the reference.
>
> > I think the authors in their Definition section should introduce the plus/minus notation where they write e.g 2_(+), just so that it is 100% clear to everyone. There is a lot going on and since it is used a lot (in place of p = +1, -1), I think the notation should be introduced explicitly.
>
> We have added an explicit introduction of this notation and, in particular, an example of a 2_(+) tensor below the definition of a 1_(p) tensor.
>
> [1] Jayesh K. Gupta and Johannes Brandstetter. Towards multi-spatiotemporal-scale generalized pde modeling, 2022.
>
> [2] Ruben Ohana, Michael McCabe, Lucas Thibaut Meyer, Rudy Morel, Fruzsina Julia Agocs, Miguel Beneitez, Marsha Berger, Blakesley Burkhart, Stuart B. Dalziel, Drummond Buschman Fielding, Daniel Fortunato, Jared A. Goldberg, Keiya Hirashima, Yan-Fei Jiang, Rich Kerswell, Suryanarayana Maddu, Jonah M. Miller, Payel Mukhopadhyay, Stefan S. Nixon, Jeff Shen, Romain Watteaux, Bruno R´egaldo-Saint Blancard, Fran¸cois Rozet, Liam Holden Parker, Miles Cranmer, and Shirley Ho. The well: a large-scale collection of diverse physics simulations for machine learning. In The Thirty-eight Conference on Neural Information Processing Systems Datasets and Benchmarks Track, 2024.

---

> ### Comment · Reviewer_ikq1 · 2025-11-26
>
> Thank you for your response. I have read all of the reviews and the additional comments that have been made. I maintain my original statement that this work should receive top billing at ICLR. Hence I maintain my score.

---

### Official Review · Reviewer_jDLE · 2025-10-29

**Soundness:** 4
**Presentation:** 3
**Contribution:** 4
**Rating:** 4
**Confidence:** 3

**Summary:**

This paper develops equivariant neural network architectures for tensor-valued data under orthogonal, Lorentz, and symplecticsymmetries. The method constructs outputs as linear combinations of outer products of inputs and isotropic tensors, with coefficients given by learnable scalar functions $q_{t,\sigma,J}$. These scalar functions are implemented as MLPs acting on the set of pairwise invariant inner products between input vectors, ensuring equivariance under the chosen group without relying on Clebsch–Gordan tensor products. The framework generalizes previous O(d) methods to indefinite orthogonal and symplectic groups, and experiments on stress–strain tensors, path signatures, and sparse recovery demonstrate flexibility and improved inductive bias.

Overall, this is a strong and well-motivated contribution that provides a clean, general, and computationally efficient formalism for building equivariant networks under multiple symmetry groups. The theoretical construction is sound and practically valuable, but the manuscript would benefit from clearer explanations of indexing, permutation handling, and the role of the $q_{t,\sigma,J}$ functions.

**Strengths:**

* Provides a unified theoretical construction of equivariant tensor functions across orthogonal, Lorentz, and symplectic groups.
* Avoids Clebsch–Gordan contractions by using invariant scalar functions and tensor assembly, leading to simpler implementations.
* The mathematical framework (Theorem 1, Corollaries 1–3) is general and connects to isotropic tensor theory.
* Demonstrates practical versatility across physics, geometry, and learning tasks.
* Empirical results show strong improvements over non-equivariant baselines.

**Weaknesses:**

1. **Notation clarity:** The paper should explicitly distinguish permutation indices (e.g., $i,j,k,l$) from “rotational” or group indices (e.g., $a,b,c,d$). Figure 1 blurs this distinction, making it hard to track which quantities are equivariant to which symmetries.

2. **Parameterization of $q_{t,\sigma,J}$:**
   It would be helpful to more clearly explain how the MLPs are used in practice on the collection of inner product of input vectors. I understand that each $q_{t,\sigma,J}$ outputs a scalar coefficient that modulates a tensor built from outer products and isotropic components, but I'm not sure how the MLP handles the permutation indices on the inner product "tensor".

3. **Permutation handling:**
   Related to above, my understanding is that the $q_{t,\sigma,J}$ are not permutation-equivariant functions; they are invariant with respect to the rotational/Lorentz/symplectic group but labeled by specific index tuples $(t,\sigma,J)$.
   In theory, Corollary 1 sums over all permutations $\sigma \in S_{k'}$, but in practice the implementation collapses equivalent $\sigma$ into isotropy classes (Appendix D), so only a small subset of unique tensor structures is instantiated. It would be good to clarify what is a helpful an analytical indexing device vs. how things are practically implemented.

4. **Isotropic tensors:**
   Clarify whether isotropy is defined only with respect to the group action or also includes permutation symmetries (e.g., $a_{ij}=a_{ji}$).
   For rank-4 tensors relevant to the neo-Hookean example, an explicit enumeration of independent isotropic components under partial or full permutation symmetry would help readers connect this to classical elasticity theory.

5. **Related work:**
   It would be good to discuss Cartesian-tensor–based equivariant methods that similarly use inner/outer product constructions to add to the context of your methods:
  * https://www.nature.com/articles/s41467-024-51886-6
  * https://ieeexplore.ieee.org/stamp/stamp.jsp?arnumber=9711441 (which while for a only the rank 1 case, does generalize to higher dimensions)
  * perhaps https://dl.acm.org/doi/10.5555/3666122.3667745

6. **Comparison to irrep-based models:**
   A short discussion on expressivity, i.e., whether the span of inner/outer-product constructions matches the space of irreducible-tensor interactions for O(3), would contextualize how complete the representation is relative to Clebsch–Gordan–based methods.

7. **Minor:** Corollary 1 refers to Figure 3, but this appears to correspond to Figure 1.

**Questions:**

1. How are the $q_{t,\sigma,J}$ implemented in code, e.g. shared across $\sigma$ or instantiated per isotropy class?
2. Does Eq. (11) enforce any permutation equivariance among input vectors, or is equivariance only with respect to the underlying group?
3. For the neo-Hookean example, what precise symmetry assumptions on the rank-4 tensor (e.g., $ijkl=klij$) are encoded?
4. How does the proposed construction’s expressive span compare to that of CG-based O(3) models?

---

> ### Author Response · Authors · 2025-11-25
> **Rebuttal Part 1**
>
> > Notation clarity: The paper should explicitly distinguish permutation indices (e.g., i,j,k,l) from “rotational” or group indices (e.g.,  a,b,c,d). Figure 1 blurs this distinction, making it hard to track which quantities are equivariant to which symmetries.
>
> We have updated the notation in Figure 1 to match what appears in the statement of Corollary 1. In this figure, the $\langle v_i, v_j\rangle$ that are the arguments of each coefficient $q$ are represented as the Gram matrix, the dot products of the input vectors. We have explicitly written out the multi-indices $J$ appearing in $q_{t,\sigma,J}$, as well as the values of $t$ and the permutation $\sigma$ being applied. The caption for Figure 1 also explains the notation being used.
>
> > Parameterization of q_t,\sigma,J: It would be helpful to more clearly explain how the MLPs are used in practice on the collection of inner product of input vectors. I understand that each q_t,\sigma,J outputs a scalar coefficient that modulates a tensor built from outer products and isotropic components, but I’m not sure how the MLP handles the permutation indices on the inner product tensor.
>
> The model we consider here is not permutation invariant with respect to the input vectors. It considers tuples of input tensors with a specified order, and if the inputs are reordered the output can change. Therefore the MLP does not need to consider permutations of the inner product tensors.
>
> If one wants to make this architecture permutation invariant, one can do so by modifying the formulation in Corollary 1. To do so we need to identify the set of indices $J$ and $J’$ if they have the same number of elements $(N)$ and their indices are isomorphic as partitions of $[N]$. For example $J= (1, 1, 3)$ and $J’=(2,3,3)$ should be identified. Further, $\sigma$ and $\sigma’$ such that $\sigma’ \sigma^{-1}$ fixes the last $2t$ components, we need to identify $q_{t,\sigma, J}$ and $q_{t,\sigma’,J}$.
>
> Now we can set the $q_{t,\sigma, J} = q_{t,\sigma’, J’}$ for all $J\sim J’$ and $\sigma \sim \sigma’ (=: q_{t,\bar\sigma, \bar J})$ so the function is independent of the order of the inputs. Moreover all the functions $q_{t,\sigma, \bar J}$ need to be invariant with respect to permutations of rows and columns of the matrix of inner products in the input. This can be implemented using a graph neural, or a model like [1].
>
> > Permutation handling: Related to above, my understanding is that the q_t,\sigma,J are not permutation-equivariant functions; they are invariant with respect to the rotational/Lorentz/symplectic group but labeled by specific index tuples (t,\sigma,J). In theory, Corollary 1 sums over all permutations sigma in S_k’, but in practice the implementation collapses equivalent sigma into isotropy classes (Appendix D), so only a small subset of unique tensor structures is instantiated. It would be good to clarify what is a helpful an analytical indexing devices vs. how things are practically implemented.
>
> Thank you for your question. $S_k’$ includes the permutations of the input tensors (so we consider $v1\otimes v2$ as well as $v2\otimes v1$) and the permutations of the isotropic tensor components. The isotropic tensors have symmetries themselves so the set of permutations we need to consider is effectively much smaller. We discuss this in Appendix D. In practice we implemented them using equation (106). Our code works for any order tensor output. We have added a small note after Corollary 1.
>
> > Isotropic tensors: Clarify whether isotropy is defined only with respect to the group action or also includes permutation symmetries (e.g., a_ij = a_ji). For rank-4 tensors relevant to the neo-Hookean example, an explicit enumeration of independent isotropic components under partial or full permutation symmetry would help readers connect this to classical elasticity theory.
>
> The isotropy is not with respect to permutations of the indices of the tensor, only with respect to the Lie group, $O(d)$. If we were using Theorem 1, we would need to write down the $O(d)$-isotropic 4-tensors (which are not symmetric). However, we use Corollary 2 which uses a different parameterization that does not require the 4-tensors. We clarified in the paper that the permutation equivariance of Corollary 2 is of the function of eigenvalues.
>
> [1] A Galois theorem for machine learning: Functions on symmetric matrices and point clouds via lightweight invariant features. Ben Blum-Smith, Ningyuan Huang, Marco Cuturi, Soledad Villar

---

> > ### Author Response · Authors · 2025-11-25
> > **Rebuttal Part 2**
> >
> > > Related work: It would be good to discuss Cartesian-tensor–based equivariant methods that similarly use inner/outer product constructions to add to the context of your methods
> >
> > Thank you for identifying these references. HotPP and Vector Neurons are both similar in that they use tensor products and contractions to build tensor networks. However, our model is able to exploit particular input types to build more efficient models in certain cases. We have added this discussion to the related works. TensorNet uses spherical tensors, so we have added it as a reference in that section.
> >
> > > Comparison to irrep-based models: A short discussion on expressivity, i.e., whether the span of inner/outer-product constructions matches the space of irreducible-tensor interactions for O(3), would contextualize how complete the representation is relative to Clebsch–Gordan–based methods.
> >
> > This is a great question. The Clebsch–Gordan methods parameterize equivariant tensor functions using representation theory. To do so they decompose tensors in the irreducible representations and use Schur’s lemma to parameterize the equivariant linear maps. If the orders of the intermediate tensor layers are large enough then the CG models can parameterize polynomial equivariant functions of arbitrary degree (and are universal in a Stone-Weierstrass sense).
> >
> > Our method also parameterizes equivariant tensor polynomials of arbitrary degree, but does so using invariant theory results instead of irreducible representations. Basically, we give a parameterization for the invariant and equivariant functions that does not require the computation of the Clebsch–Gordan coefficients.
> >
> > The Clebsch–Gordan–based methods in E3NN, ESCNN, and Domina et al. implement those for SO(d) and O(d) for d=2,3, whereas our method applies to other groups as well. We remark that those methods are more memory efficient than our general formulation in Theorems 1 and 2, but they are comparable to our Corollaries 1 and 3 (which require the inputs to be vectors but are applicable to O(d), the Lorentz group, the symplectic group).
> >
> > In summary, the computational and approximation power should be equivalent, however, the parameterization is different and the mathematical techniques used to arrive at the parameterization are also different.
> > We added a comment in the related work section.
> >
> > > Corollary 1 refers to Figure 3, but this appears to correspond to Figure 1.
> >
> > Fixed, thank you.
> >
> > > How are the q_t,\sigma,J implemented in code, e.g. shared across sigma or instantiated per isotropy class?
> >
> > All the $q_{t,\sigma,J}$ are implemented as a single MLP shared across $t$, $\sigma$, and $J$. Their inputs are the diagonal and upper triangular elements of the gram matrix of input vectors, and they have an output coefficient for each $q_{t,\sigma,J}$. I have added a note in Section 5 to clarify this.
> >
> > > Does Eq. (11) enforce any permutation equivariance among input vectors, or is equivariance only with respect to the underlying group?
> >
> > Equivariance is only enforced with respect to the underlying group. Further invariance for permutation of vector entries in Corollary 1 can be introduced by identifying the $q_{t,\sigma,J}$ appropriately (see answer above).
> >
> > > For the neo-Hookean example, what precise symmetry assumptions on the rank-4 tensor (e.g., ijkl=klij) are encoded?
> >
> > In this example we learn S (a symmetric 2-tensor) as an O(d)-equivariant function of C (a symmetric 2-tensor). If we were using the formulation from Theorem 1 we would need to implement the corresponding isotropic 4-tensors you are asking about. However, we are using Corollary 2 that says that in this case we can implement the O(d)-equivariant functions as permutation equivariant functions of the eigenvalues of the input 2-tensor. So we use the model from Maron et al [1] instead.
> >
> > > How does the proposed construction’s expressive span compare to that of CG-based O(3) models?
> >
> > The expressive power is theoretically the same in the limit. In the CG-based models, the expressive power is constrained by the order of the intermediate tensor layers, which translates into the degree of the spherical harmonics used in the parameterization of the model. The analogous quantity in our model from Corollary 1 is the degree of the q_t functions (assuming they are polynomials). One may argue that it is easier to implement expressive models using Corollary 1 than using the CG-based models, because the q_t functions can be any non-linear functions without breaking the equivariance, whereas implementing non-linearities in CG-based models is slightly more tricky.
> >
> > [1] Haggai Maron, Heli Ben-Hamu, Nadav Shamir, and Yaron Lipman. Invariant and equivariant graph networks. In 7th International Conference on Learning Representations, ICLR 2019, New Orleans, LA, USA, May 6-9, 2019. OpenReview.net, 2019.

---

> > > ### Comment · Reviewer_jDLE · 2025-11-26
> > >
> > > Thank you for the detailed responses to my questions and updates to the notational clarity of the paper. My concerns have been addressed, and I have updated my score accordingly.

---

### Official Review · Reviewer_A7mP · 2025-10-31

**Soundness:** 4
**Presentation:** 3
**Contribution:** 3
**Rating:** 8
**Confidence:** 4

**Summary:**

This paper studies the representation of equivariant mappings between tensors to tensors under various symmetry groups. In particular, they characterize equivariant polynomials for general rank input and output tensors under $O(d)$, $O(s, d-s)$ and $Sp(d)$ symmetries. In the case of vectors and symmetric matrices (ie, rank 1 and rank 2 tensors) under $O(d)$ symmetry, their general characterizations specialize to those known in the literature (eg. Villar, et al. 2021). They derive an extension of their results to algebraic groups. Finally, they report empirical results using their characterization compared to existing models on predicting stress-strain relationships, path signature prediction and sparse vector estimation.

**Strengths:**

I am a big fan of this work. The paper is very well-written, and generalizes several existing results. While the most general constructions are not practically implementable, the authors derive specializations in specific practical settings which are practical to implement.
I think the field will benefit from having these general characterizations in one place.

**Weaknesses:**

The empirical comparisons could be strengthened (see comments below).

**Questions:**

* Do you think there is a way to additionally incorporate permutation equivariance (eg. when dealing with sets of vectors, not sequences) into your characterizations?
* Do you have thoughts on parametrizing $SO(d)$-equivariant polynomials, such as those considered in Villar, et al. 2021 for vectors (where now cross-products come into play)?
* Do you have thoughts on how one can predict $O(d)$-isotropic rank-$k$ tensors?
* Can you add an experiment with existing E3NN baselines such as NequIP for the $O(d)$ equivariance experiments? For example, you could take a look at the e3tools repository: https://github.com/prescient-design/e3tools/blob/main/examples/models/conv.py for an existing PyTorch implementation. My understanding is that those models are performing pairwise tensor products but are explicitly permutation equivariant as well.
* Do you perform data augmentation for your MLP baseline, to approximately obtain G-equivariance? If not, can you add those results?

---

> ### Author Response · Authors · 2025-11-25
>
> > Do you think there is a way to additionally incorporate permutation equivariance (eg. when dealing with sets of vectors, not sequences) into your characterizations?
>
> The functions we consider here take $n$ tensors and output one tensor. In order to consider permutation equivariance, we need to consider functions that output $n$ tensors. Villar et al. 2021 [1] proposes one way to do it (Proposition 11) which should be applicable to our formulation as well. This was implemented in Gemnet [2] as well (Theorem 3).
>
> In case you are referring to permutation invariance, Corollary 1 can be modified to be permutation invariant. To do so we need to identify the set of indices $J$ and $J’$ if they have the same number of elements $(N)$ and their indices are isomorphic as partitions of $[N]$. For example $J= (1, 1, 3)$ and $J’=(2,3,3)$ should be identified. Further, $\sigma$ and $\sigma’$ such that $\sigma’ \sigma^{-1}$ fixes the last $2t$ components, we need to identify $q_{t,\sigma, J}$ and $q_{t,\sigma’,J}$.
>
> Now we can set the $q_{t,\sigma, J} = q_{t,\sigma’, J’}$ for all $J\sim J’$ and $\sigma \sim \sigma’ (=: q_{t,\bar\sigma, \bar J})$ so the function is independent of the order of the inputs. Moreover all the functions $q_{t,\sigma, \bar J}$ need to be invariant with respect to permutations of rows and columns of the matrix of inner products in the input. This can be implemented using a graph neural network, or a model like [3].
>
> > Do you have thoughts on parametrizing SO(d)-equivariant polynomials such as those considered in Villar, et al. 2021 for vectors (where now cross-products come into play)?
>
> We can parametrize those using Theorem 5 in the appendix (since $SO(d)$ is compact it satisfies the hypothesis of Theorem 5). This means that Theorem 1’s generalization to $SO(d)$ is straightforward. This requires using the $SO(d)$ isotropic tensors, which we describe in equation (98) in Appendix C.
>
> If we also want to obtain an analogue of Corollary 1 for $SO(d)$, it is possible but more complicated. It doesn’t reduce to a simple expression, due to the many possible contractions of vectors with the Levi-Civita tensor.
>
> > Do you have thoughts on how one can predict $O(d)$-isotropic rank-$k$ tensors?
>
> Lemma 3 in Appendix C parameterizes $O(d)$-isotropic rank-$k$ tensors. In the positive parity case, they are just linear combinations of powers of the Kronecker delta with permuted indexes. In the negative parity case, it is the same except each summand also includes exactly one Levi-Civita tensor. If the $O(d)$-isotropic order $k$ tensor is a function of some tensor inputs, then the scalars of the linear combinations can depend on the input tensors as $O(d)$-invariant scalar functions.
>
> > Can you add an experiment with existing E3NN baselines such as NequIP for the equivariance experiments? For example, you could take a look at the e3tools repository: https://github.com/prescient-design/e3tools/blob/main/examples/models/conv.py for an existing PyTorch implementation. My understanding is that those models are performing pairwise tensor products but are explicitly permutation equivariant as well.
>
> Thank you for this suggestion. We are working on comparing against E3NN but have run into some technical issues in the code implementation.
>
> > Do you perform data augmentation for your MLP baseline, to approximately obtain G-equivariance? If not, can you add those results?
>
> Thank you, this is a good idea. The existing experiments do not perform data augmentation for the MLP baseline, but we are running those experiments now.
>
> [1] Soledad Villar, David W Hogg, Kate Storey-Fisher, Weichi Yao, and Ben Blum-Smith. Scalars are universal: Equivariant machine learning, structured like classical physics. Advances in Neural Information Processing Systems, 34:28848–28863, 2021.
>
> [2] Johannes Gasteiger, Florian Becker, and Stephan G¨unnemann. Gemnet: Universal directional graph neural networks for molecules, 2024.
>
> [3] A Galois theorem for machine learning: Functions on symmetric matrices and point clouds via lightweight invariant features. Ben Blum-Smith, Ningyuan Huang, Marco Cuturi, Soledad Villar

---

> > ### Comment · Reviewer_A7mP · 2025-11-26
> >
> > Thank you for your response. Looking forward to seeing the results of your new experiments.

---

> ### Author Response · Authors · 2025-12-03
> **MLP with augmented data**
>
> We have added experiments with the baseline MLP trained on an augmented dataset. For each data point in the original training dataset, we sample four elements uniformly from O(d) and use them to rotate the input and output tensors. Thus the resulting training dataset is four times the size of the original. We have run these additional experiments for the materials experiments and the path signature experiments in O(d). The augmented model does better than the baseline MLP, but worse than the equivariant methods.

---

### Official Review · Reviewer_RRnU · 2025-11-01

**Soundness:** 3
**Presentation:** 3
**Contribution:** 3
**Rating:** 6
**Confidence:** 3

**Summary:**

The paper provides a general mathematical characterization and construction of tensor-to-tensor functions equivariant to orthogonal, Lorentz, and symplectic groups. It gives explicit parameterizations, extends invariant-theory results, and demonstrates the approach in three domains (materials science, path signatures, sparse vector estimation), with consistent improvements over non-equivariant baselines.

**Strengths:**

**Originality**

Extending equivariance theory to Lorentz and symplectic groups in a unified tensor framework is new and meaningful.

**Quality**

The theoretical development seems mathematically rigorous, consistent, and well-supported by references in invariant theory.

Empirical results, though synthetic or semi-synthetic, convincingly show the advantage of symmetry-aware architectures across different domains.

**Clarity**

Despite mathematical density, definitions and proofs are clearly stated, and appendices support reproducibility.

**Significance**

The proposed general framework could influence how equivariance is implemented for high-order tensor data—a major area of emerging interest in scientific ML.

Potentially impacts ML applications in materials modeling, physics simulation, and tensor decomposition problems.

Provides a foundation for future architectures extending GNN and geometric deep learning paradigms to more general symmetry groups.

**Weaknesses:**

All data are synthetic or toy (the stress–strain task uses the Garanger et al. (2024) dataset, which if I understood correctly is still a simulated material data), no large-scale, real-world or high-dimensional experiments.

No complexity or scalability discussion, crucial for practical adoption.

Comparative evaluation is narrow. Competes mostly with MLPs and one prior symmetry-enforcing method, no comparison to modern equivariant neural architectures (E3NN, LieConv, TFN). Without these, it’s hard to judge whether improvements are due to the new theory or simply to enforcing symmetry at all.

**Questions:**

How does the proposed method scale computationally compared to established equivariant architectures (e.g., E3NN)?

Can you demonstrate results on a real dataset involving Lorentz or symplectic symmetry (e.g., physical simulation or robotics)?

Is there a formal or empirical ablation isolating the gain from the new isotropic-tensor formulation vs. standard equivariant parameterizations?

How large are the MLPs used for $q_{t,σ,J}$ (remark 1)? Are they a computational bottleneck?

---

> ### Author Response · Authors · 2025-11-25
>
> > No complexity or scalability discussion, crucial for practical adoption.
>
> We now briefly discuss computational complexity after Corollary 1. Note that an intrinsic limitation of the application of this result is that the computation of the equivariant function in Corollary 1 becomes intractable when k’ becomes large. However, for applications, small values of k’ such as k’ =1,2,3,4 are of practical interest.
>
> > Comparative evaluation is narrow. Competes mostly with MLPs and one prior symmetry-enforcing method, no comparison to modern equivariant neural architectures (E3NN, LieConv, TFN). Without these, it’s hard to judge whether improvements are due to the new theory or simply to enforcing symmetry at all.
>
> > How does the proposed method scale computationally compared to established equivariant architectures (e.g., E3NN)?
>
> > Is there a formal or empirical ablation isolating the gain from the new isotropic-tensor formulation vs. standard equivariant parameterizations?
>
> Thank you for this suggestion. We are working on comparing against E3NN but have run into some technical issues in the code implementation. For the materials science experiments, our method compares favorably with the Tensor Field Equivariant Neural Network (TFENN) of Garanger et al (2024), as shown in Table 1.
>
> > Can you demonstrate results on a real dataset involving Lorentz or symplectic symmetry (e.g., physical simulation or robotics)?
>
> Thank you for this suggestion. Do you have any real tensor dataset in mind for the Lorentz or symplectic symmetries?
>
> > How large are the MLPs used for $q_{t,\sigma,J}$ (remark 1)? Are they a computational bottleneck?
>
> The parameter counts and MLP architectures are described in the appendix. Section H.2 for the stress tensor example, Section I.2 for the path signature example, and Section J.7 for the sparse vector recovery. The models we consider are relatively small and the number of parameters is not a computational bottleneck. Most of our models have about 50k parameters and the largest model we consider has 1.3M parameters.

---

> ### Author Response · Authors · 2025-12-03
> **MLP with augmented data**
>
> We have added experiments with the baseline MLP trained on an augmented dataset. For each data point in the original training dataset, we sample four elements uniformly from O(d) and use them to rotate the input and output tensors. Thus the resulting training dataset is four times the size of the original. We have run these additional experiments for the materials experiments and the path signature experiments in O(d). The augmented model does better than the baseline MLP, but worse than the equivariant methods.

---

### Author Response · Authors · 2025-12-03
**Summary of discussion**

We appreciate the reviewer's positive feedback on our paper. This is a quick summary of the discussion.

**Strengths:**
The reviewers valued the paper's clarity, mathematical rigor, and applicability of the proposed methodology to implement equivariant machine learning models on tensors.
Some positive comments were:
(RRnU) "Extending equivariance theory to Lorentz and symplectic groups in a unified tensor framework is new and meaningful."
(A7mP) "I am a big fan of this work."
(jDLE) "Overall, this is a strong and well-motivated contribution that provides a clean, general, and computationally efficient formalism for building equivariant networks under multiple symmetry groups."
(ikq1) "The results are clearly significant and deserve top billing at ICLR."

**Weaknesses:**
The main weaknesses that were brought up by the reviewers related to two points.
(1) Reviewer (jDLE), who gave us the lowest score, was mainly concerned about how permutations of the inputs were considered in our model. We clarified that our model does not consider permutations of the tensor inputs (inputs can represent different types of data, and we describe general models that are not necessarily permutation invariant). We explained how to extend the model to make it permutation invariant using recent results in the literature. And we added a remark in the manuscript explaining this. The reviewer was satisfied with this answer and increased their score. Note that even though reviewer jDLE gave us the lowest score (4), their original review is very positive, including scores
Soundness: 4: excellent
Presentation: 3: good
Contribution: 4: excellent

(2) Reviewers RRnU and A7mP requested extra numerical experiments, including comparison with data augmentation-based methods and comparisons with E3NN. We added results of augmentation-based methods (which performed as expected, better than the non-equivariant methods but worse than the strictly equivariant methods). However, we weren't able to finish the experiments comparing with E3NN during the rebuttal period, unfortunately. The E3NN code doesn't straightforwardly apply to our numerical experiments, and when we adapted the E3NN code to our problem, the performance we obtained was significantly worse than we expected, so we suspect we didn't do it correctly. We are not comfortable reporting these results at this point. However, we believe this experimental comparison is not fundamentally needed to justify the value of the contributions in this paper.

Thank you all reviewers for your positive and constructive feedback.

---

### Meta-Review · Area_Chair_iLCy · 2025-12-28

**Summary:**

This submission develops a general and rigorous characterization of tensor-to-tensor equivariant functions under several classical Lie groups, including the orthogonal, Lorentz, and symplectic groups, and derives practical parameterizations that avoid explicit Clebsch–Gordan constructions. The framework is further demonstrated across three application domains.

The paper was reviewed by four reviewers, whose assessments after the rebuttal range from marginal accept to strong accept. Two reviewers expressed strong enthusiasm for both the theoretical depth of the work and the clarity of its presentation. One reviewer was broadly positive while noting concerns regarding empirical breadth and the scope of experimental comparisons. One reviewer was initially marginal but indicated an improved assessment following the rebuttal.

**Reviewer Concerns:**

The main concerns centered on the empirical evaluation rather than the theory, including limited comparisons to modern equivariant models, reliance on synthetic or semi-synthetic datasets, and the absence of large-scale benchmarks. No reviewer questioned the correctness of the theoretical results, which were widely praised for their rigor and unifying perspective. The authors provided a thorough and technically strong rebuttal, clarifying implementation details, strengthening experiments and related work, and addressing most concerns, leading at least one reviewer to update their score upward.

**Reviewer Scores:**

Among the four reviewers, final evaluations ranged from marginal accept to strong accept, with two reviewers strongly endorsing acceptance, and the initially borderline reviewer revising their stance after rebuttal. The score distribution reflects broad support with limited remaining reservations, primarily about experimental scope rather than core contribution.

---

### Decision · Program_Chairs · 2026-01-26

Accept (Poster)